**www.cambridge.org/qrd**

# Pulmonary surfactant and COVID-19: A new synthesis

Barry Ninham[1,2] 🄳, Brandon Reines[1,3] 🄳, Matthew Battye[4] 🄳 and Paul Thomas[1] 🄳

[1]Materials Physics (formerly Department of Applied Mathematics), Research School of Physics, Australian National University, Canberra, ACT 2600, Australia; [2]School of Science, University of New South Wales, Northcott Drive, Campbell, Canberra, ACT 2612, Australia; [3]Department of Biomedical Informatics, University of Pittsburgh School of Medicine, 5607 Baum Blvd, Pittsburgh, PA 15206, USA and [4]Breakthrough Technologies, Deakin, ACT, Australia

## Research Article

**Keywords**
COVID; Lung Surfactant; Pulmonary Surfactant; Gas Exchange; nano-bubbles; physical chemistry

**Authors lists for each chapter**
Chapter 1: Reines*/Ninham/Battye/Thomas
*Corresponding author. E-mail: reinesb@pitt. edu,
Chapter 2: Ninham*/Reines/Battye/Thomas
*Corresponding author. E-mail: barry. ninham@anu.edu.au

Brandon Reines is the principal author of chapter 1, Barry Ninham is the principal author of chapter 2. The author list is ordered alphabetically.

This work should be read with its prequel.

Reines BP, Ninham BW (2019). Structure and function of the endothelial surface layer: unravelling the nanoarchitecture of biological surfaces. *Quarterly Reviews of Biophysics* **52**, e13, 1–11. https://doi.org/ 10.1017/S0033583519000118

## Abstract

**Chapter 1:** COVID-19 pathogenesis poses paradoxes difficult to explain with traditional physiology. For instance, since type II pneumocytes are considered the primary cellular target of SARS-CoV-2; as these produce pulmonary surfactant (PS), the possibility that insufficient PS plays a role in COVID-19 pathogenesis has been raised. However, the opposite of predicted *high* alveolar surface tension is found in many *early* COVID-19 patients: paradoxically normal lung volumes and high compliance occur, with profound hypoxemia. That 'COVID anomaly' was quickly rationalised by invoking traditional vascular mechanisms–mainly because of surprisingly preserved alveolar surface in early hypoxemic cases. However, that quick rejection of alveolar damage only occurred because the actual mechanism of gas exchange has long been presumed to be non-problematic, due to diffusion through the alveolar surface. On the contrary, we provide physical chemical evidence that *gas exchange occurs by an process of expansion and contraction of the three-dimensional structures of PS and its associated proteins.* This view explains anomalous observations from the level of cryo-TEM to whole individuals. It encompasses results from premature infants to the deepest diving seals. Once understood, the COVID anomaly dissolves and is straightforwardly explained as covert viral damage to the 3D structure of PS, with direct treatment implications. As a natural experiment, the SARS-CoV-2 virus itself has helped us to simplify and clarify not only the nature of dyspnea and its relationship to pulmonary compliance, but also the fine detail of the PS including such features as water channels which had heretofore been entirely unexpected.

## Abstract

**Chapter 2:** For a long time, physical, colloid and surface chemistry have not intersected with physiology and cell biology as much as we might have hoped. The reasons are starting to become clear. The discipline of physical chemistry suffered from serious unrecognised omissions that rendered it ineffective. These foundational defects included omission of specific ion molecular forces and hydration effects. The discipline lacked a predictive theory of self-assembly of lipids and proteins. Worse, theory omitted any role for dissolved gases, $O_2$, $N_2$, $CO_2$, and their existence as stable nanobubbles above physiological salt concentration. Recent developments have gone some way to explaining the foam-like lung surfactant structures and function. It delivers $O_2/N_2$ as nanobubbles, and efflux of $CO_2$, and $H_2O$ nanobubbles at the alveolar surface. Knowledge of pulmonary surfactant structure allows an explanation of the mechanism of corona virus entry, and differences in infectivity of different variants. $CO_2$ nanobubbles, resulting from metabolism passing through the molecular frit provided by the glycocalyx of venous tissue, forms the previously unexplained foam which is the endothelial surface layer. $CO_2$ nanobubbles turn out to be lethal to viruses, providing a plausible explanation for the origin of 'Long COVID'. Circulating nanobubbles, stable above physiological 0.17 M salt drive various enzyme-like activities and chemical reactions. Awareness of the microstructure of Pulmonary Surfactant and that nanobubbles of $(O_2/N_2)$ and $CO_2$ are integral to respiratory and circulatory physiology provides new insights to the COVID-19 and other pathogen activity.

## Table of contents

## CHAPTER 1. Introduction: COVID-19 as a natural experiment in physiology: Developments in physical chemistry throw light on its anomalous features

Whether basic physiology enlightens understanding of human pathology or *vice versa* is perhaps the central conundrum of modern biomedical science (Beecher, 1960; Good, 1991; Reines, 1991). In graduate training, most of us are taught that discoveries are made in laboratories and then applied to clinical practise (Comroe, 1977*a*, 1977*b*; 1977*c*). However, pathways of discovery may sometimes begin with observation of clinical anomalies or 'experiments of Nature' which are then investigated by basic scientists, eventually feeding back to the clinic (Good, 1991). In that light, given current controversy over COVID-19 pathogenesis (Chiumello, 2020), and the reality of its most anomalous features (Chiumello, 2020; Gattinoni *et al.*, 2020*a*; Tobin *et al.*, 2020), we must ask: Can traditional physiology explain COVID-19 pathogenesis, or should it force us to completely re-evaluate our most basic scientific assumptions?

In line with the latter possibility, we contend that COVID-19 pathogenesis is quite unprecedented and requires new and revolutionary physical chemical concepts to explain (Hyde *et al.*, 1997; Ninham *et al.*, 2017*a*, 2017*b*). Chapter 2 outlines new developments in physical chemistry and how their application explains the Covid-19 anomalies. It should be noted at the outset that physical chemistry, like biophysics, is quite different in its conceptual orientation from the molecular biology with which physiologists are more familiar. In physical chemistry, it is not just the geometry of macromolecules that determines specific interactions (e.g., 'lock and key' fit), but the physical forces of attraction embracing the totality of physical properties of the interacting particles, and those of the intervening medium that separates them. These properties give rise to highly specific long- and short-range forces (Bostrom *et al.*, 2001; Lo Nostro *et al.*, 2005; Ninham *et al.*, 2016). Under certain circumstances, such as viral invasion, these forces may give rise to actual molecular and/or cellular damage.

In fact, although viral entry through cell receptors and cellular damage are often highlighted, virus-induced damage is just as likely to occur to molecular constituents of the extracellular spaces. Viruses *do* need to enter cells to replicate but may induce more damage extracellularly – and we believe this to be the case for SARS-CoV-2. Although this sort of subtle damage is unfamiliar to biologists, it is likely to play an important pathogenic role for viruses like SARS-CoV-2 that behave in unusual ways *in vivo*.

What is unusual about SARS-CoV-2's *in vivo* behaviour? At least two unusual aspects of SARS-CoV-2 have emerged over the past 2 years: (1) the virus has great difficulty infecting many cell types, including endothelial cells (Goldsmith *et al.*, 2020; Scholkmann and Nicholls, 2020; Ahmetaj-Shala *et al.*, 2021; Bozzani *et al.*, 2021; Sridhar and Nicholls, 2021), and (2) the virus has an unusual capacity to survive in the extracellular spaces (Goh *et al.*, 2020; Cevik *et al.*, 2021). Interestingly, both its difficulties in infecting cells and enhanced survival in extracellular spaces are likely related to an unusual physical attribute of SARS-CoV-2: hardness of its membrane protein 'shell' (Goh *et al.*, 2020). In fact, Goh has shown that the harder outer shell of SARS-CoV-2 has contributed greatly to the pathogenic properties of the actions of this COVID-19 virus in direct comparison with the less resilient and much less pathogenic properties of the original virus SARS-CoV-1 (Goh *et al.*, 2022).

Indeed, SARS-CoV-2 may spend an inordinate period of its *in vivo* life cycle in the extracellular spaces. Its difficulty in infecting cells may explain another COVID anomaly: the surprising weakness of IFN 1 upregulation in body fluids of those infected with SARS-CoV-2 (Lei *et al.*, 2020; Lopez *et al.*, 2020; Ruetsch *et al.*, 2020; Zhang *et al.*, 2020), as IFN 1 is only increased in cells that are successfully virus infected. Many other anomalous pathogenetic features of COVID-19 have forced us to question whether traditional biological concepts are sufficient to explain it (see Table 1).

### 'Happy' or silent hypoxemia: Absence of dyspnea with low blood oxygen on presentation

Absence of dyspnea with hypoxemia in COVID-19 has been often analysed separately from preserved lung compliance with hypoxemia. For instance, Tobin denies the existence of both anomalies: he doubts that COVID-19 patients are genuinely hypoxemic due to inaccuracies in oximeters and fever effects on oxygen dissociation curve; furthermore, his group contends such patients would not be expected to be dyspneic, even if they were profoundly hypoxemic: older patients particularly those who are diabetic are less likely to present with dyspnea anyway (Tobin *et al.*, 2020). Tobin's is a minority viewpoint, and there is fair consensus that, compared to other causes of ARDS, COVID-19 patients often present with remarkably preserved lung mechanics, given the degree of measured hypoxemia (Chiumello, 2020; Coppola *et al.*, 2021; Gattinoni and Marini, 2021).

If, however, we take the standpoint of COVID-19 as a natural experiment, how might our interpretations of these COVID anomalies be altered? To us, the initial presentation of COVID-19 strongly suggests that those two anomalies are pathogenetically and physiologically linked in ways that are not apparent from traditional concepts and vocabulary (as others have commented)

**Table 1.** COVID-19 anomalies: relative to other viral acute respiratory distress

| Anomalous feature | Current *ad hoc* hypotheses | Our premise | Our conclusion |
|---|---|---|---|
| Hypoxemia absent dyspnea on presentation | (1) no true hypoxemia (2) older patients often not dyspneic (Tobin *et al.*, 2020) | Conscious awareness of difficulty breathing correlates strongly with reduction in compliance | Lack of dyspnea at presentation is due to near normalcy of lung mechanics |
| Hypoxemia with relatively normal lung mechanics on presentation | Hypoxemia is due to vascular derangements and not alveolar damage | Pulmonary surfactant is the main mediator of gas exchange | Hypoxemia is due to insufficient pulmonary surfactant due to: (1) direct damage by SARS-CoV-2 and (2) viral damage to surfactant-producing type II pneumocytes |
| Early radiological features lack typical alveolar edema (suggesting generalised damage to alveolar epithelium) | No explanation | SARS-CoV-2 only successfully infects 1:1,000–1:10,000 of the most infectable lung epithelial cells, type II pneumocytes (Sender *et al.*, 2021) | Too few alveolar epithelial cells are damaged to induce production of exudates and edema characteristic of diffuse alveolar damage B and D that are hydrophobic |
| Interferons rarely found increased in lung lavage fluid from COVID-19 patients (Ruetsch *et al.*, 2020) | SARS-CoV-2 must short circuit usual cellular antiviral mechanisms | SARS-CoV-2 only successfully infects 1:1,000–1:10,000 of the most infectable lung epithelial cells, type II pneumocytes | Low or absent interferons are due to low number of cells successfully infected by SARS-CoV-2 to trigger normal interferon responses (Acharya *et al.*, 2020) |
| Unusually high concentrations of SARS-CoV-2 in saliva and mucus but not blood | Not explained | Relatively 'hard' membrane protein shell of SARS-CoV-2 helps explain low viral load in tissues, but higher survival in body fluids other than blood (Goh *et al.*, 2020) | SARS-CoV-2 is relatively resistant to proteolytic enzymes in saliva in mucus. But the virus is easily cleaved by the easily hydrolysed oxidants released by the $CO_2$ nanobubble foam in the venous micro vessels (Reines and Ninham, 2019), as shown for antimicrobial action of $CO_2$ generally (Sanchis *et al.*, 2019). In other words, the $CO_2$ nanobubbles are destroying the virus in the blood. |
| Microthrombi in alveolar capillaries 9X more frequent in COVID-19 than flu (Ackermann *et al.*, 2020) | Platelet hyperactivity secondary to inflammation ('cytokine storm') | SARS-CoV-2 damages pulmonary surfactant first and penetrates deeper through gas exchange units including alveolar-capillary endothelium | Localised virus-induced damage to alveolar capillary endothelium creates microenvironmental conditions favourable to microthrombus formation |
| Alveolar capillary microthrombi almost always in capillaries or arterioles, and not venous microvessels (Ackermann *et al.*, 2020) | Not explained | SARS-CoV-2 is easily hydrolysed by oxidants released by the $CO_2$ nanobubble foam in the venous microvessels (Reines and Ninham, 2019), as shown for antimicrobial action of $CO_2$ bubbles (Sanchis *et al.*, 2019) | SARS-CoV-2 is able to survive in capillaries and arterioles long enough to cause endothelial damage, but unable to do so in venous micro vessels (Reines and Ninham, 2019; Sanchis *et al.*, 2019) |

(Beachey, 2020). In defining what constitutes 'dyspnea', respiratory physiologists have tended to distinguish it as the person's (conscious) perception of actual lung pathology, but have left some ambiguity to the term, in that 'dyspnea' can also occur while the person is asleep (unconscious). One lesson COVID-19 seems to be trying to teach is that 'dyspnea' should be more rigorously defined as 'conscious perception of mechanical difficulty in inflating the lungs on inhalation while awake'. In that light, the absence of dyspnea in COVID-19 on presentation is understandable, simply because there is no pathologic trigger for dyspnea at that point.

Additionally, a further physiological simplification implied by COVID-19 is that chemoreceptors which respond to $O_2$ and $CO_2$ levels by altering ventilation rate and quality through neural mechanisms are below human awareness (i.e., unconscious). So, hypoxemia alone, absent abnormal lung compliance, would not be expected to produce dyspnea. The main remaining question relevant to the pathogenesis of hypoxemia in COVID-19 is: What is the most likely cause of the early hypoxemia? Do we have any good models of human disease characterised by hypoxemia with normal lung compliance?

## Neonatal respiratory distress model: First clues that pulmonary surfactant mainly functions to facilitate gas exchange

In the midst of the COVID-19 pandemic, Motoyama's group published the revolutionary finding that premature infants suffering from neonatal respiratory distress syndrome (NRDS) do <u>not</u> suffer poor lung compliance compared to two different control groups (Koumbourlis and Motoyama, 2020), contrary to 60 years of doctrine to the contrary (Comroe, 1977c). They, therefore, argued that, given hypoxemia in the face of normal compliance in such premature infants, the best model for COVID-19 is NRDS itself. This extraordinary conclusion led us to revisit the many studies of pulmonary surfactant therapy for NRDS,

which were performed in an attempt to improve compliance of the infants' lungs by lowering surface tension of alveolar surfaces. That entire undertaking was based on the belief that the main function of pulmonary surfactant is to lower alveolar surface tension. It is a fact that premature infants have underdeveloped lungs with insufficient surfactant production (Comroe, 1977*c*). However, although the success of surfactant treatment for preemies has often been construed as the ultimate verification of the traditional surface tension-lowering theory (Hills, 1999), the actual results of the clinical studies flatly contradict that theory, in that: 1) surfactant treatment of NRDS improves oxygenation and Functional Residual Capacity (FRC) long before compliance, and 2) hypoxemic NRDS patients usually have fairly normal compliance to begin with (Milner, 1995; Koumbourlis and Motoyama, 2020).

Although the conventional view of surfactant function has been criticised for many years (Bangham, 1992; Hills, 1999; Dorrington and Young, 2001; Ninham *et al.*, 2017*b*) the success of pulmonary surfactant treatment of respiratory distress syndrome (RDS) in premature infants has often been interpreted as fully verifying the 'one-sided bubble model' of the aqueous hypophase/lower phase of the biphased pulmonary surfactant (AH) of the alveolar surface (Hills, 1999). Not only have the RDS results tended to increase confidence in traditional views of the structure of the alveolar surface, but also the textbook account of its key function: lowering of surface tension.

However, a closer look at the actual results of NRDS therapy reveals a very different story. After administration of surfactant, particularly natural surfactants which include all four surfactant proteins A-D, *oxygenation improved in the treated infants within 15 min, while compliance did not improve until many hours later, as long as 24 h post-treatment* (Milner *et al.*, 1983; Morley, 1984; Bhat *et al.*, 1990; Goldsmith *et al.*, 1991; Pfenninger *et al.*, 1992; Abbasi *et al.*, 1993; Davis *et al.*, 1988). Closely correlated to the oxygenation improvements in NRDS babies was greater functional residual capacity (FRC). Studies showed that natural surfactant treatment increases the FRC by 50–330% within 15 min, precisely the time course during which gas exchange was shown to improve (Goldsmith *et al.*, 1991). Directing attention to many anomalous observations in NRDS, Milner postulated that the early improvement in blood oxygen was 'due to the relatively large volumes of fluid instilled down the endotracheal tube or to the period of manual ventilation by bag using 100% oxygen' (Milner *et al.*, 1984). He found that pulmonary surfactant did not improve measured compliance at all, compared to saline (Milner *et al.*, 1984). He and others suggested that the relief of hypoxemia was due to surfactant enhancing gas exchange indirectly through its effects on V/Q mismatch, intrapulmonary shunting, and related phenomena (Alexander and Milner, 1995). Surfactant may mainly increase the FRC and this would 'tend to reduce intrapulmonary shunting and might improve ventilation/perfusion ratios. It might even diminish extrapulmonary shunts if there was a direct or indirect effect on the pulmonary artery pressure' (Milner, 1995). Milner's *ad hoc* hypotheses attempting to explain how surfactant might operate through traditional vascular physiological and respiratory physical-chemical mechanisms are mirrored in current attempts to explain COVID-19 hypoxemia with preserved lung mechanics in similar fashion, as due to vascular and not alveolar damage (Gattinoni *et al.*, 2020*a*; Lang *et al.*, 2020; Solaimanzadeh, 2020) (although the possibility of early alveolar damage is certainly suggested by damaged diffusing capacity in recovered COVID-19 patients) (Fuschillo *et al.*,

2021). However, this was prior to our current knowledge of the gas exchange function of pulmonary surfactant (Andersson *et al.*, 1999). So the fact that pulmonary-surfactant-treated neonates experienced immediate improvement in oxygenation makes perfect physical-chemical sense. In addition, we believe the reason that improvements in oxygenation in surfactant-treated neonates are temporally related to improved FRC is because the residuum is comprised of $CO_2$ gas nanobubbles, which provide a persistent foam, which is not completely eliminated on expiration.

Although those authors noting the delay in expected improvement in compliance did not claim that surfactant has no role in surface-tension-lowering and improved compliance, they did ask 'How does exogenous surfactant really work?' (Milner, 1993, 1995) It is certainly true that, in many early publications about NRDS, the lungs of premature infants are often described as 'stiff', requiring high pressures to inflate (Avery and Mead, 1959; Gribetz *et al.*, 1959). Laboratory studies of surface tension of bits of minced lung taken from preemies at autopsy using Wilhelmy balance and other older methods suggested that surface tension was indeed higher in preemies than in normal infants (Avery and Mead, 1959). However, heightened surface tension in post-mortem tissues was inappropriately presumed to translate into high lung compliance in actual patients. It is now clear that global compliance is largely unrelated to surface tension in individual alveoli (Perlman *et al.*, 2011). Premature lungs are underdeveloped in many ways including deficiency of surfactant. It is therefore interesting that surfactant was able to effect improved oxygenation, although alveoli are not fully developed in preemies, and are really alveolar buds. This, and other anatomical and phylogenetic considerations, as outlined by Perez-Gil, suggests that pulmonary surfactant is able to enhance gas exchange as long as it occurs in the airways (e.g., tracheoles of birds, and perhaps distal airways in diving seals, as we explore below) (Olmeda *et al.*, 2010). The resistance of premature lungs to inflation is multifactorial, and was always demonstrable globally as inadequate functional residual capacity (FRC), and not higher compliance (Alexander and Milner, 1995). Improved oxygenation with surfactant treatment likely enhanced the general growth and metabolism of lungs in premature infants ('kick started') within 24 h of treatment, and this led to the delay in ability to expand the lungs of about a day (Milner, 1996).

Although surfactant treatment of NRDS babies usually facilitates lung expansion and FRC, whether surfactant played a role in normalising lung compliance was never clear – mainly because lung compliance had rarely been investigated in RDS babies. So, it was speculative that pulmonary compliance is higher than normal in RDS. This was first tested in a controlled fashion by Motoyama's group at the University of Pittsburgh, although published only as an abstract in 1990 (Koumbourlis *et al.*, 1990), and recently in more complete form in response to COVID-19 (Koumbourlis and Motoyama, 2020). Twelve premature neonates who were mechanically ventilated because of respiratory distress syndrome (RDS) were compared with 13 term infants with acute respiratory distress due to meconium aspiration syndrome (MAS) requiring extracorporeal membrane oxygenation. For normal controls, 10 term newborn infants who had been briefly intubated for procedures under anaesthesia, but had normal lungs were compared to other two groups. Although both NRDS and MAS infants' deflation flow-volume curves showed evidence of high airway conductance, the NRDS infants had both lung volume and compliance near that of normal controls (Koumbourlis and Motoyama, 2020).

## Gas exchange and tissue storage is mediated principally by nanobubbles and lipids, and only secondarily by specialised globins

Traditional respiratory physiology holds that (1) $O_2$ and $CO_2$ simply diffuse through the alveolar and capillary linings, (2) lipid-rich pulmonary surfactant has no role in gas exchange, and (3) $O_2$ is stored solely in specialised globins–hemoglobin in blood and myoglobin in muscle – in birds and mammals.

Notwithstanding continued faith in those three assumptions, a very wide range of physical-chemical observations *in vitro* and molecular whole-animal and even natural historical studies contradicts them (Larsson *et al.*, 1999, 2002; Spragg *et al.*, 2004; Meir *et al.*, 2009; Olmeda *et al.*, 2010; Jue *et al.*, 2016; Nguyen and Perlman, 2018). Lack of communication between molecular biologists and biologically-oriented physical chemists has slowed progress in understanding respiratory physiology. But recognition of new physical-chemical observations is paramount, including the repeated demonstration that $O_2$ and other gases can be held in tiny 3–60 nm 'nanobubbles' which are quite stable at physiological salt concentration (Craig *et al.*, 1993*b*; Bunkin *et al.*, 2011; Alheshibri *et al.*, 2016; Ninham *et al.*, 2016; Reines and Ninham, 2019). Such nanobubbles very likely form a parallel system of $O_2$ storage and release, with hemoglobin and myoglobin (see Box 1).

Although nanobubbles are still essentially unknown to molecular biologists, the question of pulmonary surfactant function has been vigorously debated among molecular thinkers and physical chemists for many decades (Bangham, 1992; Scarpelli and Hills, 2000; Ninham *et al.*, 2017*b*). A very brief collaboration between physical chemists and paediatric pulmonary physiologists in the 1950s and 1960s led to the current view of pulmonary surfactant function as principally surface tension lowering (Comroe, 1977*c*). But, since then, the growth of physical-chemical knowledge about lipid self-assembly generally and pulmonary surfactant, in particular, has been explosive (Ninham *et al.*, 2017*a*). Although almost completely unknown to biologists, a physical-chemical revolution has occurred in understanding of cell surfaces and biological surfaces in genera. Although not familiar to biologists or even physical chemists, there have been considerable developments in awareness of the subtlety of membrane structures and biological surfaces generally, particularly in the ubiquity of non-Euclidean geometries (Hyde *et al.*, 1997; Larsson *et al.*, 1999; Almsherqi *et al.*, 2006; Chong and Deng, 2012; Larsson and Larsson, 2014; Deng and Angelova, 2021). This has also led molecular biologists to begin to re-evaluate the structure and function of pulmonary surfactant.

Although the notion that pulmonary surfactant exists as a monolayer at the air–water interface had been criticised by physical chemists for many years, because of the impossibility that such a monolayer could maintain the very low surface tension required by classical theory (Bangham, 1992; Dorrington and Young, 2001), it was not until experimental demonstration that surfactant exists as a multilayered liquid crystalline phase *in vivo* that the monolayer theory was clearly refuted by Larsson's group (Andersson *et al.*, 1999; Larsson and Larsson, 2014; Ninham *et al.*, 2017*b*). This led other investigators to explore possible 3D structures for surfactant, and also different functions. The first experimental *in vitro* demonstration that intact pulmonary surfactant actually does accelerate $O_2$ transport compared to a pure water phase or water phase saturated with purely lipid membranes was first published in 2010 (Olmeda *et al.*, 2010). Although Perez-Gil argues that pulmonary surfactant likely has an important role in gas exchange due to its purported capacity to accelerate oxygen diffusion (Olmeda *et al.*, 2010), we reject the notion of $O_2$ 'diffusion' in favour of the model of opening and closing of the 3D cubic structure of pulmonary surfactant shown in Fig. 1*d*, explained further in the following chapter.

As shown in Fig. 1*d*, this remarkably open tetragonal structure is what pulmonary surfactant (PS) looks like when in its most ordered and contracted state *in vivo* (Larsson *et al.*, 1999), *when it contains nanobubbles of $O_2/N_2$ air.* How these nanobubbles are delivered to the alveolar capillaries is unclear, but it is possible that the entire sheet of PS is frictionless enough to slide through to the capillary wall, where the $O_2/N_2$ cargo might be deposited quite close to the capillary surface (as captured in Jastrow's TEM image Fig. 3*d*, a position which has yet to be verified in Cryo-TEM). The front, sides and back of the cubes consist of bilayers of dipalmitoyl-phosphatidyl choline (DPPC), while the peg-like structures at the corners are the surfactant proteins, principally SP-A, as explained in the second chapter. Protein A in particular has been shown to induce peaks and valleys in sheets of 'tubular myelin' *in vitro* and likely plays a similar role in creating three dimensions out of 2 *in vivo* (Perez-Gil, 2008). The proteins likely bind the lipid planes together at the corners in a 'nuts and bolts' or Lego-like contraption. Surfactant proteins B and C are small and extremely hydrophobic, and likely embedded inside the lipid tails of the bilayers of phosphatidyl-glycerol (PG) in the corners (see Fig. 3, and sections 'The real lung surfactant – Structure and machinery', 'Summary', and 'Lung pathologies and lipids: biophysical aspects'). The disassembly of PS involves a loosening/undoing of the nuts and bolts of the corners of the cubes; at full PS expansion, $CO_2$ and $H_2O$ are adsorbed as nanobubbles (Alheshibri *et al.*, 2016), and eventually exhaled (Box 2 and second chapter).

As a visual aid to understanding, one can imagine the Mathematica-calculated 3D structure in Fig. 1*d* ( as occurring in bilayer 'strings' or 'threads' on top of one another (lining the alveolar surface on inspiration). This multilayered structure might be likened to a knitted sweater of lipid bilayers held together by protein pegs at its corners; if one were to take a cross section through this 'knitted sweater' of 3D pulmonary surfactant threads, one would see a structure something like the one in Fig. 1*d*. On PS expansion, as the Lego-like protein 'pegs' fall off of the corners of the structure, the single 'thread' of pulmonary surfactant is effectively pulled out to a

---

**Box 1: Nanobubbles: a parallel system of $O_2$ storage and delivery in blood**

It had long been universally assumed that, where gas is involved in physiology, it must occur in a molecular state. For instance, oxygen was presumed to occur only as molecular oxygen or $O_2$. A good deal of recent evidence, however, points to the existence of states of aggregation of dissolved gas in the form of nanobubbles of approximately 20–60 nm in size. In physiological saline, such nanobubbles are stable (Bunkin *et al.*, 2011; Alheshibri *et al.*, 2016; Yurchenko *et al.*, 2016; Reines and Ninham, 2019). The evidence is connected to a simple phenomenon that involves bubble–bubble interactions (Craig *et al.*, 1993*a*). Nanobubbles will fuse with one another in solutions where salts occur outside of physiological concentrations. However, above 0.17 molar (M) salt concentration, gas nanobubbles will not fuse and are quite stable. Although the phenomenon is well documented and reliable, its mechanism is not understood: indeed, the stability of nanobubbles at physiological salt concentration is completely inexplicable by classical physical chemistry and colloid science. In blood, nonetheless, $O_2$ must be delivered to red blood cells by partitioning from the nanobubbles to the large available adsorbent in haemoglobin, a process which may be enhanced by increased hydrostatic pressure. In biological evolution, nanobubbles must have preceded specialised globins for $O_2$ storage and release by millennia, but the two systems subsequently co-evolved in parallel. See next Chapter 2 for a more complete discussion of nanobubbles.

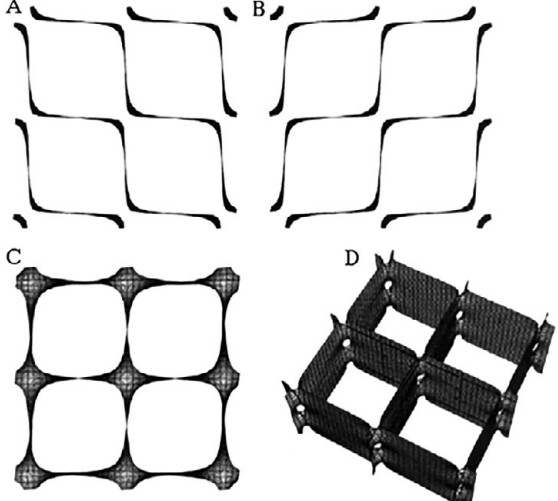

**Fig. 1.** Two-dimensional (2D) cryo-TEM models of actual pulmonary surfactant *ex vivo* and calculated 3D structure on full molecular 'inspiration'. (*a–c*) are sections of pulmonary surfactant (PS) *ex vivo* revealed by cryo-TEM studies of freshly opened alveolar surfaces of rabbit lungs, by transfer of the surface layer to the grid and immediately freezing the structure without ice crystal formation. The imaged pattern is seen in two adjacent cross-sections (*a* and *b*), and then these cross-sections overlap (*c*). Seeing this cubic lattice-like appearance came as a welcome surprise to Larsson *et al.* (1999), as it had precisely the dimensions of the lattice-like structure often seen in traditional EM known as 'tubular myelin' (TM). The finding that the cubic dimensions (and interbilayer distance) are approximately 40 nm in the PS found by cryo-TEM and what had been known as TM strongly suggested that they were one in the same structure, albeit slightly different microanatomic positions. This observation implied that TM is not merely a highly-ordered storage depot for PS lipids, but the $O_2$-containing contracted state of PS at the alveolar surface, and perhaps subjacent to it (although the possibility exists that traditional EM processing methods artefactually 'knock' the PS at the surface to a deeper position below the alveolar surface than it normally occupies, as conceded by Jastrow himself, personal communication with BR, 10/12/2021). Image (*d*) is the structure calculated using Mathematica software program and adopting the nodal surface approximation, showing the postulated non-Euclidean 3D structure of pulmonary surfactant on full inspiration known as 'crossed layers of parallels' or 'CLP' (Reproduced from Andersson *et al.*, 1999; Larsson and Larsson, 2014), obtaining permission of Springer Copyright 1999 and 2002. The corners of the cubes are occupied by surfactant protein A (SP-A), which holds the corners together in Lego-like fashion. The expanded and contracted states of PS are explained in more detail in Box 2, and in the second chapter.

stack of single lamellar lipid bilayers (as in the top of Fig. 10 in Chapter 2). The original thick 2D sweater can expand prodigiously in length. In its stretched-out configuration, as a long strand of bilayers, it adsorbs $CO_2$ nanobubbles and water which are then exhaled. The lung surfactant proteins released in the expanded state await

---

**Box 2: Molecular breathing: Gas exchange via phase changes of pulmonary surfactant (PS) 3D structure**

Respiration is generally thought of macro-anatomically as involving expansion and contraction of the thoracic cavity, where $O_2$ and $N_2$ are breathed in the air on inhalation, and $CO_2$ and $H_2O$ are breathed out on exhalation. The movements of the gases through tissue are generally treated as non-problematic, as they are thought to move simply by diffusion. However, many discoveries in physical chemistry cast doubt on that simplistic view. It appears that, in fact, pulmonary surfactant (PS) has a 3D structure, which itself expands and contracts: actively imbibing $O_2 \cdot N_2$ as nanobubbles on contraction, and releasing $CO_2/H_2O$ foamed nanobubbles on its full expansion.

In addition, it is important for physiologists to note that this 'molecular breathing' on the nanoscale is quite different from true anatomic breathing on the macroscale, and the interconnections between nano-scale molecular and macro-scale anatomic breathing will take time to work out.

---

reassembly and reincorporation into the folded bilayer that incorporates $O_2$ and $N_2$, as it collapses down again to its compact form, and then expands, and so on. PS is as close as we have seen to a perpetual motion machine.

## Widening the view: Molecular studies of pulmonary surfactant (PS) and respiration in birds and mammals generally

Regardless of the specific mechanism, it is apparent that there is now good experimental *in vitro* (Olmeda *et al.*, 2010), cryo-TEM, clinical (mainly from RDS), and natural historical evidence in favour of the gas exchange function of pulmonary surfactant. Perez-Gil questions whether the currently accepted paradigm of pulmonary surfactant and respiration is accurate and general, as, 'for instance, in birds, pulmonary surfactant is associated with the tracheoles, the gas exchange moiety of the lungs, and not that much with the air sacs, the changing volume of which is actually responsible for mobilising air through the tubular avian lungs (Olmeda *et al.*, 2010).' We agree with Perez-Gil that there is need for a complete re-evaluation of the function of pulmonary surfactant in many species. What is implied in this new view is that lipids and lipid-protein complexes are likely the primary means of gas exchange and storage (Koopman and Westgate, 2012), with globins providing a secondary means of controlling terrestrial oxygen delivery (Jue *et al.*, 2016). By far the most incisive 'natural experiment' in this regard are the adaptations of marine mammals to extended deep dives.

Indeed, from a biological standpoint, the most striking contradiction to the traditional view of surfactant function is the fact that the deepest diving pinnipeds have surfactant *with little or no surface activity* (Spragg *et al.*, 2004). The mechanical demands on deep diving mammalian lungs are very similar to those on a newborn infant, but even more challenging: the pinniped lung must completely collapse during the deep dive, and then fully expand at the surface to breathe air, and this process must be repeated over and over during the lifetime of the seal.

Astonishingly, however, given the still pervasive view that surfactant lowers surface tension, seals that dive deepest and for longest time periods have surfactant which actually *raises* surface tension of water (Spragg *et al.*, 2004). The northern elephant seal (*Mirounga angustirostris*) routinely dives to depths of 400–800 m for 10–30 min, staying at the surface for only 2 min. How elephant seals are able to maintain essentially aerobic metabolism during their deep dives concurrent with severe hypoxemia constitutes an outstanding anomaly. In routine dives, $PvO_2$ and $PaO_2$ values reached 2–10 and 10–23 mmHg, respectively, corresponding to $SO_2$ of 1–26% and 2 contents of .3 (venous) and 2.7 ml 02/dl blood (arterial $O_2$ content depletes by 91% in arteries and 100% in veins during the dive (Meir *et al.*, 2009).

How seals maintain aerobic metabolism has mainly been attributed to increased capacity for oxygen storage in hemoglobin in blood and myoglobin in muscle, by virtue of possibly increased concentrations of myoglobin (Mirceta *et al.*, 2013). But the rates of oxygen depletion noted are very difficult to reconcile with maintenance of aerobic metabolism in deep-diving seals and other marine mammals.

Is it possible that seals switch to a completely novel mechanism of oxygen storage and delivery to tissues during the deepest dives? We believe so. And we postulate that excess production of pulmonary surfactant during dives is key to this alternative

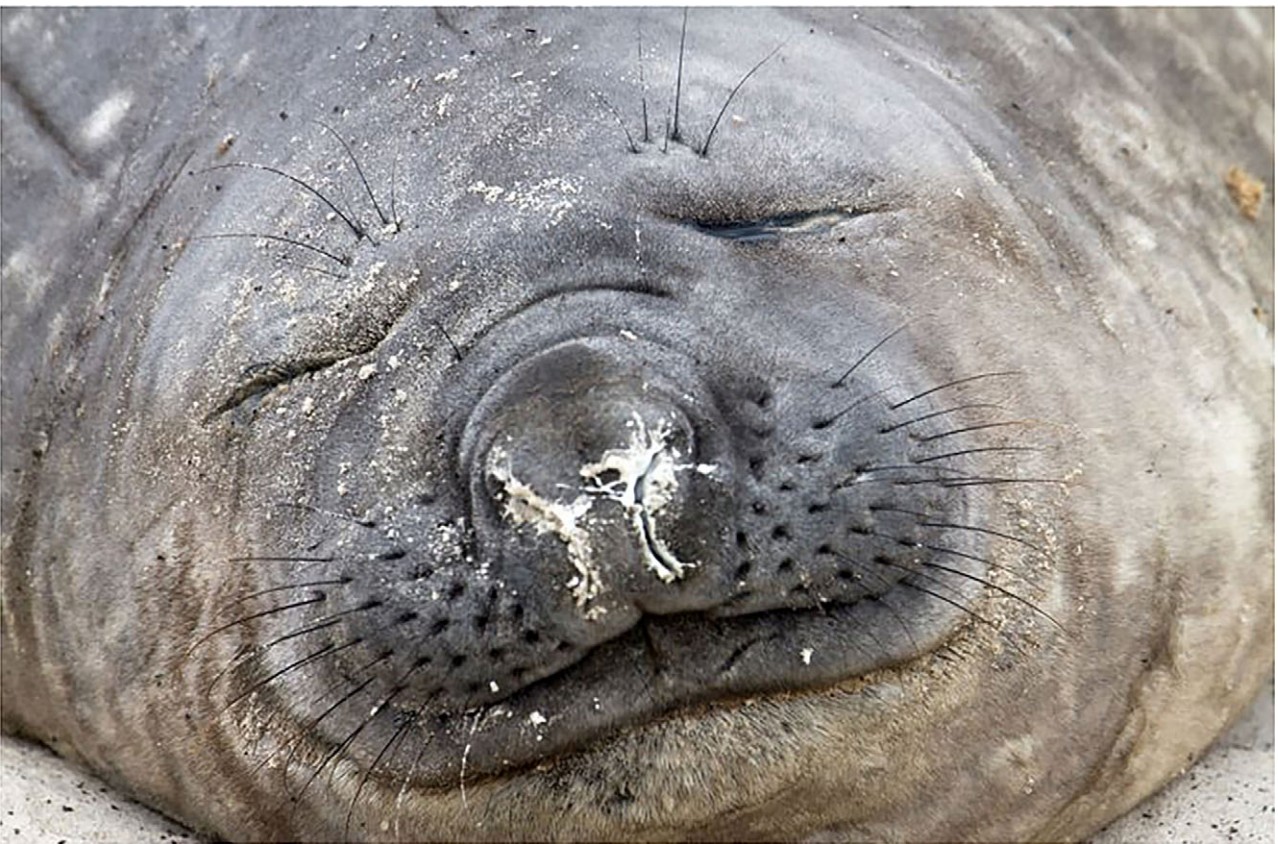

**Fig. 2.** Copious production of pulmonary surfactant in Elephant Seals. This is a photograph of an elephant seal sleeping on the beach. Note that the whitish material collecting around the external nares is not mucus, but pulmonary surfactant. Weddell seals are also known to cough or sneeze up excess pulmonary surfactant when they surface. This fits with our hypothesis that deep-diving seals switch to a completely hydrostatic system for delivering oxygen to tissues, where $O_2$ is stored in the surfactant as nanobubbles during the dive and diffuses to all tissues at depth (as documented in 1959 in swine experiments where erythrocytes were depleted, and oxygen delivered under hyperbaric conditions). It has been documented *in vitro* that type II pneumocytes of seals can produce surfactant at high pressures of deep dives, and likely produce it abundantly.

mechanism. It has been shown that deep-diving seals not only have the capacity to produce surfactant during dives (Miller *et al.*, 2004), but, on surfacing, actually cough or sneeze out large volumes of it (Miller *et al.*, 2004). It has been documented that the alveolar contents including surfactant are squeezed into the distal airways during deep dives. At depth, we postulate that oxygen is stored in nanobubbles which are held as a foam in the tetragonal 3D structure of pulmonary surfactant; and this $O_2$ diffuses out to the tissues due to high hydrostatic pressure. It came as a surprise to us that, although the occurrence of $O_2$-containing nanobubbles has yet to be proven *in vivo* in human beings, precisely the mechanism of tissue oxygenation we postulate here was proven as long ago as 1959 in pigs depleted of erythrocytes and exposed to hyperbaric oxygen (!) (Boerema *et al.*, 1959).

Although the structure of pulmonary surfactant of deep-diving seals has a few minor differences from terrestrial mammal surfactant, the seal surfactant should still have some capacity to lower, rather than raise, surface tension. Seal biologists have not presented any physical-chemical mechanisms by which surfactant would lose its purported surface-tension-lowering activity in favour of 'an anti-adhesive function', as has been argued (Spragg *et al.*, 2004). It is certainly possible that the pulmonary surfactant of elephant seals does have some 'anti-adhesive' action, as does terrestrial pulmonary surfactant, as its 3D structure is that of a foam, which would tend to keep alveolar surfaces from direct contact (much as

the nanobubble foam in the endothelial surface layer (ESL) prevents erythrocytes from directly contacting the blood vessel wall; Reines and Ninham, 2019).

## Rush to attribute early COVID-19 hypoxemia to vascular mechanisms was premature, due to incomplete understanding of gas exchange

Beginning with Gattinoni's first reports on early COVID-19 silent hypoxemia, because little in the way of alveolar damage could be detected radiologically or otherwise, explanations have centred on vascular and not alveolar aspects of gas exchange (Gattinoni *et al.*, 2020a, 2020b). Vascular damage of some kind is presumed to underlie early hypoxemia in COVID-19 (Gattinoni and Marini, 2021). However, three new analyses call into question those results highlighting vascular mechanisms to explain the early COVID anomaly (DuBrock and Krowka, 2020; Herrmann *et al.*, 2020; Cherian *et al.*, 2021). Herrmann *et al.* (2020) explain that anomalies in the pathogenetic data led them to construct a mathematical model which would more carefully test the idea of V/Q mismatch and absent or poor hypoxic pulmonary vasoconstriction. They wrote the early stages of COVID-19 appear to be unique and poorly understood, manifesting in the lung as peripheral lesions characterised by ground-glass opacification on computed tomography. Curiously,

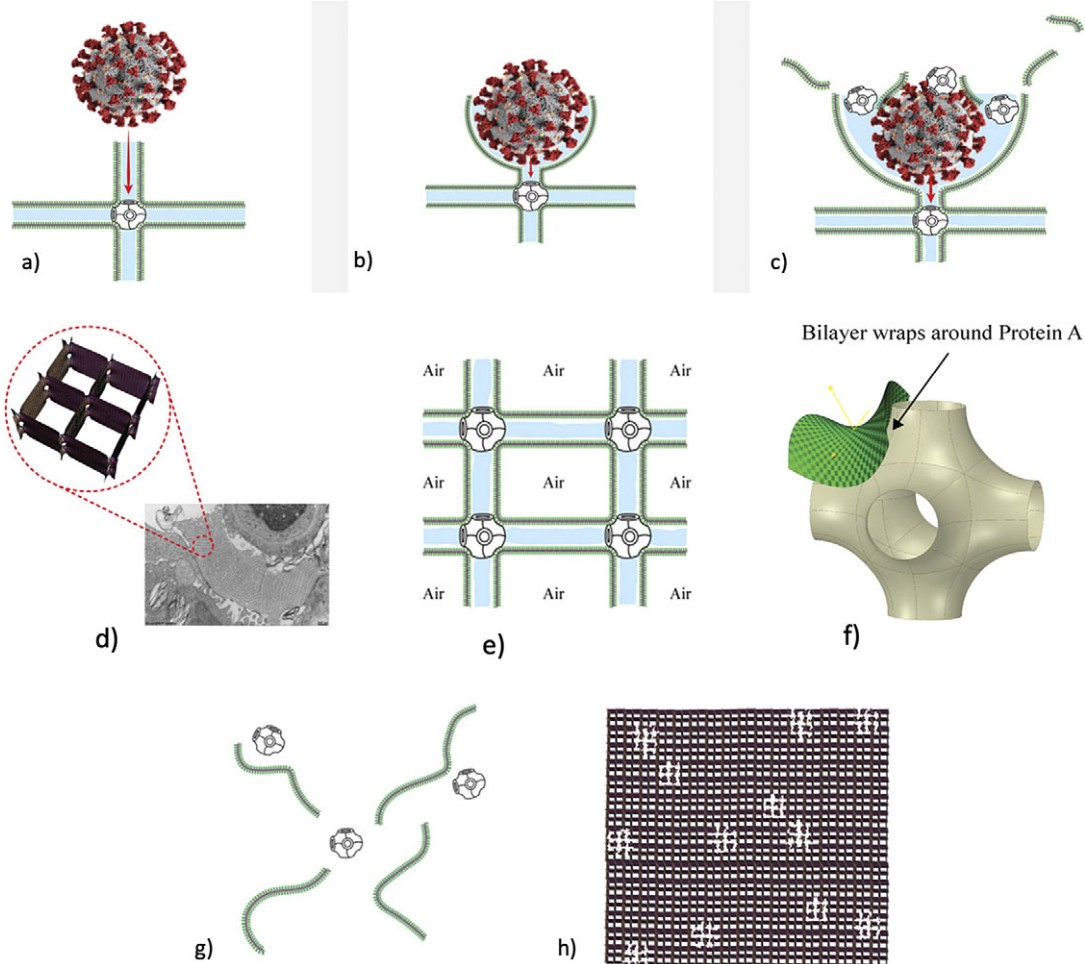

**Fig. 3.** How SARS-CoV-2 disrupts the 3D structure of pulmonary surfactant by breaking open its corners. (*a*) Analysis of the crisscross pattern often seen in electron micrographs of the alveolar surface reveals a 3D structure which is pulmonary surfactant (PS), shown in its full open configuration on inspiration. (*b*) A top-down view of the 3D structure of PS shows within its corners the hexagonal surfactant protein A (SP-A) shown in white against blue water channels, which are formed inside the polar head groups of the lipid bilayer of DPCC which comprises the main phospholipid (90%) forming the walls of the 3D structure of PS. (*c*) A more detailed analysis of an individual corner of PS reveals an enlarged SP-A with a single green covering of phosphatidyl glycerol (PG) which comprises 10% of the phospholipid of PS, the hydrophilicity of which allows it to have hydration compatibility with the hydrophilic surface of SP-A. (*d–f*) SARS-CoV-2 is shown entering a water channel of PS and moving towards a SP-A-containing corner, which it ultimately disrupts, freeing SP-A into circulation (*g*), and destroying the overall 3D structure of PS (*h*).

the fraction of lung affected in this way is often surprisingly low given the severity of the associated hypoxia and estimated shunt fractions (average 50%). If one assumes that ground-glass opacification represents lung that is nonventilated, these CT studies imply abnormally high ratios of shunt fraction to nonaerated lung fraction of 3.0 for COVID-19 compared to 1.3 for ARDS Herrmann et al., 2020. Their model finds that truly vast increases in diameter of pulmonary arterioles would be necessary to explain such a vast decrease in vascular resistance, opening of pulmonary vessels, and delivery of blood to non-aerated lung tissue, concluding only that 'Approximating vascular resistance using the Hagen-Poiseuille equation, this change in resistance corresponds to an increase in vascular diameter of 26 to 35%. Whether this degree of vasodilation is physiologically plausible is uncertain Herrmann et al., 2020.

Reynolds *et al.* (2020) had used automated transcranial Doppler (TCD) ultrasound to define the prevalence of intracardiac or intrapulmonary shunting in patients with COVID-19. With this method, saline microbubbles are injected into a central or peripheral venous catheter and TCD is used to detect and quantify microbubbles that appear in the cerebral circulation. Normally,

the microbubbles, whose diameter exceeds the pulmonary capillaries (15 μm), are trapped in the pulmonary circulation. In patients with extreme intrapulmonary vasodilatation or abnormal arteriovenous communications, however, the bubbles transit through the pulmonary circulation and can be visualised downstream in the middle cerebral artery (as detected by TCD).

Reynolds and colleagues found that the majority (15/18, 83%) of patients with COVID-19 had detectable microbubbles in the cerebral circulation by TCD. However, although Reynolds *et al.* interpret their results using hepatopulmonary syndrome as a model, in HS, dilation of pulmonary arterioles and capillaries can be as wide as 500 μm. The results from TCD imply that, in COVID-19, the pulmonary vessels are somewhat wider than 15 μm, but this is just an assumption. Cherian *et al.* (2020) dismiss the HS model out of hand, suggesting that compensatory opening of pulmonary arteriovenous shunts is a much more likely mechanism. Dubrock and Krowka likewise question the HS model, because 'despite severe hypoxemia in HPS due to intrapulmonary vascular dilatation, the response to 100% inspired oxygen can sometimes result in remarkably high $PaO_2$ values (500–600 mm Hg), no doubt reflecting the

**Table 2.** COVID19 in comparison to other interstitial lung diseases

| Name of ILD | COVID similarities | COVID differences | Reference |
|---|---|---|---|
| Idiopathic Pulmonary Fibrosis | Reduced gas exchange, dry cough, hypoxemia, fatigue Progressive but much slower | Much more gradual hypoxemia, reduced lung volumes gradually increasing reduction with time. Continually, progressive no remit some COVID-19 pts do have remission+ improvement. Unresponsive to steroids Pulmonary symptoms only | https://foundation.chestnet.org/lung-health-a-z/pulmonary-fibrosis/?Item=Symptoms Chest Foundation |
| Sarcoidosis | Affects a few organs, especially heart as well as lungs, sometimes improved with steroids | Usually, a recoverable disease with treatment compared to severe COVID-19 if not long survivability time frame comparably. Small nodules Less hypoxemia | https://www.hopkinsmedicine.org/health/conditions-and-diseases/pulmonary-sarcoidosis Hopkins university |
| Autoimmune ILDS such as sclerosis, rheumatoid lung | Reduced gas exchange, dry cough, hypoxemia, steroids sometimes help +improve | Less pronounced hypoxemia than severe COVID-19, much slower disease progression | https://columbiasurgery.org/conditions-and-treatments/autoimmune-interstitial-lung-disease Columbia surgery |
| Hypersensitivity Pneumonitis or Extrinsic Allergic alveolitis, e.g., Farmer's Lung or Bird Fanciers' Lung | Fast dangerous reduction in gas exchange, caused by breathing alien proteins, Hypoxemia, cough, malaise, fever. Responsive to steroids | Fast reduction in lung volumes as well as gas exchange noticeable breathlessness. Allergen cause such as foreign proteins or hay spores not virus related. | https://uihc.org/health-topics/hypersensitivity-pneumonitis Iowa Hospitals/clinics |
| Industrial ILDs Pneumoconiosis Asbestosis, Fibre Glass ILDs | Dry cough, Breathlessness, hypoxemia Progressive but slower | Unresponsive to steroids progressive but slower irreversible but symptoms eased by supplemental $O_2$ if clinically needed with improvement in blood $O_2$ levels | https://www.hopkinsmedicine.org/health/conditions-and-diseases/occupational-lung-diseases John Hopkins University |
| Drug induced ILDs, e.g., Amiodarone | Dry cough reduced gas exchange some fibrosis Responsive to steroids | More gradual hypoxemia, reversible if drug medication withdrawn | https://www.ncbi.nlm.nih.gov/pmc/articles/PMC2687560/ National Institute of Health UK |
| Infectious ILDS usually pneumonia or 'normal' ARDS-Acute Respiratory Disease Syndrome | Hypoxemia, Fever, breathlessness, fatigue, confusion | Reduced Lung volumes improved usually with antibiotics supplemental $O_2$ improves hypoxemia Productive cough with mucus often present | https://www.mayoclinic.org/diseases-conditions/pneumonia/symptoms-causes/syc-20354204 Mayo clinic |

lack of associated alveolar damage as seen in ARDS or COVID-19 pneumonia' (emphasis added) (DuBrock and Krowka, 2020). The latter statement is a rare, if tacit, recognition that there is very likely some form of alveolar damage in COVID-19.

## Discussion and conclusions: A 'consilience of inductions' implicates covert viral damage to 3D structure of pulmonary surfactant in COVID-19

Discovery rarely occurs due to overwhelming evidence within a single body of fact favouring a new viewpoint. More often, discovery occurs when suggestive evidence from a variety of bodies of fact converges on a common theme or conclusion, a process known as 'consilience of inductions' (Joseph, 2016). In fact, COVID-19 itself, as a natural experiment in physiology, is one of many such bodies of fact, which strongly suggest that not only is pulmonary surfactant (PS) principally involved in gas exchange but also when selectively damaged by SARS-CoV-2, defective PS produces the very unusual clinical syndrome of hypoxemia with normal lung mechanics – the COVID anomaly so often misattributed to defective vascular mechanisms (Gattinoni et al., 2020b). We review above the other bodies of fact implicating PS in gas exchange including: (1) cryo-TEM images of alveolar surface layer contradicting the assumption that PS exists as a monolayer and revealing a non-Euclidean 3D structure known as crossed layers of parallel (CLP) (Andersson et al., 1999; Larsson et al., 1999, 2002; Larsson and Larsson, 2014), (2) the chronology of re-oxygenation of premature infants when treated with PS clearly shows that oxygenation improves in minutes, while compliance, if abnormal at all, improves slowly over 24 h (Milner, 1995, 1996), (3) PS in deepest diving seals does NOT lower surface tension of water at all, and often raises it (Spragg et al., 2004; Meir et al., 2009), (4) animal experiments in which PS is insufflated via endotracheal tube shows that PS often raises surface tension of alveoli (Nguyen and Perlman, 2018), (5) when adult patients with ARDS are treated with PS, oxygenation often improves rapidly, but surface tension lowering may occur only to a minimal degree (Markart et al., 2007), and (6) early trials of PS in advanced COVID-19 have produced promising results (Bhatt et al., 2021).

Given the evidence for a gas exchange function of PS outlined above (and in the paper that follows), we believe that the most likely explanation for the severe hypoxemia and normal compliance in COVID-19 patients on presentation is that (1) *SARS-CoV-2 directly damages the delicate structure of pulmonary surfactant*, and perhaps also to a lesser extent, (2) SARS-CoV-2 damages type II pneumocytes and thereby decreases production of PS. The first

mechanism is graphically depicted in Fig. 3. Of course, viral damage to PS by respiratory viruses is not unusual, and has been documented since 1967 (Ashbaugh *et al.*, 1967). What IS unusual about SARS-CoV-2 is not that it damages PS, but that it causes so little damage to the alveolar pneumocytes and alveolar surface generally, with only 1/10,000 of type II pneumocytes likely productively infected by SARS-CoV-2 (Sender *et al.*, 2021). *So, it is the selectivity of the damage induced by SARS-CoV-2 in disrupting the 3D structure of PS – unachievable in any laboratory experiment– that makes it unique, and produces the anomalous clinical features of COVID-19.*

As shown in Fig. 3, we postulate that SARS-CoV-2 directly damages the corners of the 3D pulmonary surfactant structure which abrogates its gas exchange function.

We have produced an animation showing how SARS-CoV-2 damages the overall structure of PS which is explained in much greater detail in the second chapter: https://barryninham.com/COVIDanimate.

Although SARS-CoV-2 causes little direct damage to the alveolar lining generally, at least until quite late in only the most advanced cases (Coppola *et al.*, 2021), its specificity in most COVID-10 cases is what makes it unique: the virus damages key components of gas exchange units including PS, and the endothelial lining mainly of the alveolar capillaries and venous microvessels (Ackermann *et al.*, 2020). In addition, although endothelial damage often occurs in advanced COVID-19, most of the generalised endotheliopathy seems to be due to secondary systemic inflammation (or 'cytokine storm'). These considerations strongly suggest that COVID-19 is primarily an interstitial lung disease (ILD) with covert viral damage mainly to PS and to the gas exchange units of the lung (see Tables 1 and 2). Table 2 is that COVID-19 itself could be classified as an Interstitial Lung Disease ILD and all ILDs are primarily lung diseases with reduced or poor Gas exchange particularly diminishing Oxygen uptake causing hypoxemia. However, the really significant difference is that severe Respiratory COVID-19 Disease causes very severe hypoxemia in an incredibly short time period. Many other ILDs can cause severe hypoxemia but do so over a much-prolonged time period.

Therefore, treatment of advanced COVID-19 should be aimed at replacing defective and/or depleted PS with natural or artificial PS preparations which contain all four proteins SP-A-D (Busani *et al.*, 2020; Mirastschijski *et al.*, 2020; Schousboe *et al.*, 2020; Bhatt *et al.*, 2021; Kitaoka *et al.*, 2021; Veldhuizen *et al.*, 2021). Drugs that enhance PS production by type II pneumocytes should also be utilised (Ansarin *et al.*, 2020; Kumar, 2020; Wang *et al.*, 2020; Fu *et al.*, 2021; Tolouian *et al.*, 2021).

In the following chapter and analysis, more complete documentation of the physical-chemical evidence that PS is involved with gas exchange, and how SARS-CoV-2 damages PS structure will be presented.

## CHAPTER 2: Pulmonary surfactant self assembly: Biophysical aspects of structure, function and gas exchange: Implications for COVID-19

### Introduction: New concepts

We begin with a list of concepts in physical chemistry necessary to provide insights into pulmonary surfactant (PS) structure and function. Most of these are quite new and not part of the classical canon.

1.  An extraordinary phenomenon of bubble–bubble fusion inhibition occurs above 0.17 M salt. It is precisely the same effective salt concentration in blood. The same occurs for nanobubbles. The phenomenon is unexplained and occurs for 1 class of salts while another class of salts bubbles fuse on collision. Universal combining rules, reminiscent of the periodic table characterise it. Similar phenomena occur for different isomers of sugar and for amino acids. At the same critical effective salt concentration stable nanobubbles form.
2.  Oxygen and nitrogen are compartmentalised by a foam formed with the pulmonary surfactant (PS). This provides the structure that ensures $O_2/N_2$ are delivered as nanobubbles. The bubble–bubble inhibition phenonium ensures these nanobubbles are stable in circulation.
3.  In a similar way, $CO_2$ from metabolism produces stable nanobubbles that form the protective foam which is the endothelial surface layer of the venous system. $CO_2$ nanobubbles circulate in the blood and are expelled through a different expanded form of PS.
4.  $CO_2$ nanobubbles are potent in destroying viruses and other pathogens (Garrido *et al.*, 2016; 2018; Sanchis *et al.*, 2019; Garrido and Jin, 2020).
5.  Nanobubbles are most probably integral to enzymatic and other chemical reactions.
6.  Our intuition derived from physical chemistry as it applies to biology also misses specific ion (Hofmeister) effects absent from and crucial to all aspects of theory from pH, buffers and more.
7.  Our intuition also ignores the fact that self-assembled phases of lipids and proteins are often bicontinuous structures.

See Craig *et al.* (1993*b*), Hyde *et al.* (1997), Ninham and Yaminsky (1997), Ninham (2006, 2017, 2019), Ninham and Lo Nostro, 2010, 2020), Ninham *et al.* (2016, 2017*a*, 2017*b*), Garrido *et al.* (2018), Reines and Ninham (2019), Sanchis *et al.* (2019), Lo Nostro and Ninham (2020), Hopfer *et al.* (2021), and Ball *et al.* (2022).

### Bubble-bubble fusion in electrolytes

Bubble–bubble fusion in electrolytes is inhibited above a narrow effective ionic strength centred at 0.17 M salt for a range of cation-anion pairs. For other ion pairs, there is no effect of salt concentration on bubble coalescence. *This concentration is exactly that of the blood.* Strict rules govern the phenomenon. That is, it is ion pair specific. Some pairs, fo example, ($Na^+$ $Cl^-$) exhibit the phenomenon of bubble–bubble fusion inhibition; others ($Na^+$ $ClO_4^-$) do not. Seawater is foamy; freshwater is not. This has been *inexplicable* by classical physical chemistry for over 50 years and is ignored.

The same phenomenon occurs for solutions of different isomers of sugars. Some inhibit fusion. Others do not, as for salts. These observations will be seen to be fundamental.

**Technical.** The concentration of salts in the blood (mainly $Na^+$, $K^+$, $Ca^{2+}$, $Cl^-$) is around 0.15 M. The difference between this and 0.175 M is made up by contributions in the blood to effective ionic strength (the Debye length) by small concentrations of multi-charged proteins (Kékicheff and Ninham, 1990; Nylander *et al.*, 1994). The critical concentration scales with the electrolyte strength

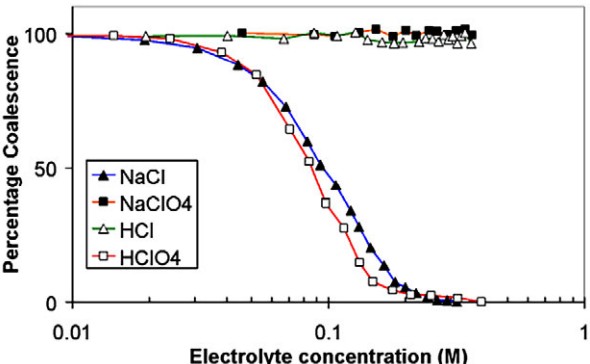

**Fig. 4.** Schematic of bubble–bubble interaction experiment with addition of salts and sugars (see Craig *et al.*, 1993*a*, 1993*b*; Nylander *et al.*, 1994; Kékicheff and Ninham, 1990; Henry and Craig, 2009; Craig and Henry, 2010).

(inverse Debye length). All (multivalent mixtures) then fall on the same curve.

It is a log plot, so the coalescence inhibition is spread over a decade of concentration. For some salts no effect occurs. With different isomers of sugars physiological implications are certain but as yet unexplored. The present-day salt concentration of the ocean is 35 g/l, mainly NaCl, four times the critical 0.17 M. The reduction of salt concentration, after an ice age, below this level would have caused massive evolutionary extinctions, probably in the Permian era.

## Classical electrolyte theory and electrochemistry

The classical theories that underpin physical, colloid and surface chemistry focus on electrostatic interactions between ions, and ions and surfaces. The theory is used to interpret pH, buffers, membrane, zeta potentials, conductivity, electrophoresis, interfacial tensions, ion transport, conduction of the nervous impulse (rafts, Hodgkin Huxley equations).

These theories were developed from the early twentieth century before quantum mechanics. But quantum mechanics is necessary to explain crucial specific ion (Hofmeister) effects. The omission means that the meaning of measurements based on classical theories are seriously flawed at physiological salt concentrations (Hyde *et al.*, 1997; Ninham and Yaminsky, 1997; Ninham, 2006, 2017, 2019; Ninham *et al.*, 2016, 2017*a*, 2017*b*; Ninham and Lo Nostro, 2010; Lo Nostro and Ninham, 2020; Ninham and Lo Nostro, 2020). For example, pH and buffers change in unpredictable ways with background solutes like sugars and proteins and salt concentration and salt kind (Ninham *et al.*, 1997; Boström *et al.*, 2003; Salis *et al.*, 2006, 2020).

When the missing quantum forces are included, a better predictive framework emerges.

But even that misses another essential factor. This is the role of dissolved gas.

**Colloid Science**. Theories of particle interactions, of macromolecules and protein structure by computer simulation do include quantum (van der Waals, dispersion) forces. But they commit a fatal error in treating these quantum forces incorrectly. Specific ion (Hofmeister) effects are excluded. The result is that the interpretation of 'hydration' becomes an artefact of erroneous theory.

While classical theories based on two-body molecular interactions are invalid in condensed media like water, the actual forces (Lifshitz theory) involve and take care of many-body interactions (Hyde *et al.*, 1997; Ninham and Yaminsky, 1997; Ninham, 2006, 2017, 2019; Ninham *et al.*, 2016, 2017*a*, 2017*b*; Ninham and Lo Nostro, 2010; Lo Nostro and Ninham, 2020; Ninham and Lo Nostro, 2020).

## Dissolved atmospheric gas

The influence of dissolved atmospheric gas has been ignored so far in theorical descriptions and in simulations of soft matter and protein structure.

*This is a catastrophe*, since if dissolved gas is removed, then (Ninham *et al.*, 2016; Ninham, 2017, 2019; Lo Nostro and Ninham, 2020; Ninham and Lo Nostro, 2020; Ninham and Pashley, 2020; Taseidifar *et al.*, 2020):

- *'hydrophobic interactions' disappear,*
- *emulsions become stable,*
- *colloidal particle forces change dramatically in magnitude and specific ion effects emerge,*
- *chemical relations like emulsion polymerisation do not proceed,*
- *enzymatic catalysis stops,*
- *cavitation stops,*
- *'hydration' changes.*

Protein structure simulation also ignores dissolved gas.

## Nanobubbles

In parallel with the neglect of effects of dissolved gas in classical theory, the existence and stability of 'nanobubbles' has never been explained and has been a matter of debate. It is confusing because most of the debate in the literature is concerned with microbubbles with radii of the order of 100–250 nm, not nanobubbles. Our concern is with nanobubbles proper, with size from say 3–40 nm (Ninham *et al.*, 2016; Ninham and Lo Nostro, 2020).

Microbubbles nucleated at surfaces are typically about 10 nm in height and hundreds of nanometres horizontally. The arguments about existence and stability are irrelevant in biology as they are stabilised against fusion by background salts ~0.17 M (recall section 'Bubble–bubble fusion in electrolytes'; Bubbles do not fuse above 0.17 M) and are further stabilised by adsorbed proteins and lipids.

The existence of nanobubbles continues to ignore 'theory', as does the bubble–bubble fusion inhibition phenomenon of section 'Bubble–bubble fusion in electrolytes'. Long lived nanobubbles in suspensions of erythrocytes in blood are reported by Bunkin *et al.* (2011).

The issue of nanobubbles is complicated and contentious (Bunkin *et al.*, 2011; Alheshibri *et al.*, 2016; Ninham *et al.*, 2016; Alheshibri and Craig, 2019*a*; Ninham and Lo Nostro, 2020) and indeed central to us. We list a substantial number of references for completeness (Vinogradova *et al.*, 1995; Bunkin *et al.*, 1996, 1997, 1999, 2010, 2011; Rao, 2010; Horn *et al.*, 2011; Alheshibri *et al.*, 2016; Yurchenko *et al.*, 2016; Alheshibri and Craig, 2018, 2019*a*, 2019*b*; Alheshibri *et al.*, 2019).

In that literature the term 'nanobubbles' embraces nano, micro, macro bubbles nucleated at surfaces or by impurities or with energy

input by flow, shaking, ultrasonics and cavitation (e.g., by propellers; Ninham and Pashley, 2020; Taseidifar *et al.*, 2020).

Many theoretical attempts have been made to account for them, but none really work. The reasons are that macroscopic physics does not apply to nanoscopic objects, and the theories ignore the effects of salt that stabilises them.

The story is described in below sections. Nanobubbles do exist in the same way as surfactant micelles exist and are central to chemical and enzyme reactions.

### *Concerning nanobubbles in the blood; A deviation*

'One of the most famous studies 'life without blood' was published by Boerema *et al.* (1959), who showed that he could keep swine alive using hyperbaric oxygen therapy (HBOT) despite haemoglobin levels that would not normally be compatible with life (see also Chiumello, 2020; Beachey, 2020). After draining all their red blood cells, a plasma or plasma-like solution was used as volume replacement, which was hyper oxygenated with hyperbaric oxygen therapy. At the end of the treatment, they were reinfused with blood and recovery was uneventful. The use of HBOT has continued to be researched' (Credit Wikipedia).

### *A useful role for nitrogen*

We know regardless of 'theory' that $O_2/N_2$ are delivered to capillaries via nanobubbles in the first instance because the TEM figures (Fig. 11) show the gas is contained in a structure with tetragonal symmetry.

The gas is adsorbed from the atmosphere and packed as nanobubbles in 'boxes' of planar bilayer lipids DPPC and PG lipids. Other supporting molecules SP-B and C and cholesterol assist the containerisation (see section 'Giant vesicle and other lipid phases: Consequences for lung pathologies' and Figs 13*h* and 14). Another large protein SP-A, a hexamer, joins the boxes together at their corners to form a regular array. Once the boxes reach their delivery site, the SP-A proteins are removed by binding to a particular cell enzyme, ACE11 (SP-D, a small hydrophilic protein also has several roles as discussed below).

The plug is pulled, by removing SP-A, and the gas is released to another medium. This has not just the role of stabilising salt above 0.17 M, but a range of all manner of proteins, lipids and other surface-active agents. These will cover the nanobubble surfaces and allow them to circulate in the bloodstream in the same way the $CO_2$ nanobubbles formed by the glycocalyx, a molecular frit lining the veins circulate.

With the $O_2/N_2$ nanobubbles, oxygen is twice as soluble as nitrogen. Usually, most of the oxygen will be transported to the haemoglobin – there is a chemical driving force to do this as haemoglobin binds $O_2$ very strongly.

If there is a barrier put up against this accelerated diffusion, for example, if the arterial surface is roughened by disease, short or long term, less oxygen will get across to the haemoglobin. It will then be retained in the nanobubbles of $N_2/O_2$.

So, there is a backup source of oxygen, regardless of whether the haemoglobin/myoglobin is there or not, as for the pigs of Boerema's experiment (Boerema *et al.*, 1959).

The natural $N_2/O_2$ nanobubbles from breathing the atmosphere are the required backup container-buffer for regulated release of oxygen.

No nitrogen, no oxygen regulation. One can make these artificial extra circulating sources of oxygen for athletes who cheat with fluoro carbons which bolsters the case.

It also fits with the fact that pure oxygen for more than an hour is poisonous and kills – if the haemoglobin is saturated and there are no nitrogen bubbles to act as a reservoir for oxygen, too much flooding the system will accelerate reactivity of too many enzymes.

These notions appear to provide an active and useful role for nitrogen not previously recognised.

### *Measurement of oxygen and carbon dioxide*

Standard techniques to measure gas content in clinical practise take no account of the existence of nanobubbles of these gases in the blood. This may vary considerably depending on patient condition. This might well give rise to inadvertent misinterpretation.

For the reasons previously outlined and evident from the structures formed of lung surfactant lipids, Oxygen is delivered by lung surfactant in the form of nanobubbles, ~40 nm. It is not delivered as molecular oxygen and nitrogen. $CO_2$ is expelled as nanobubbles on expiration. Nanobubbles appear to exhibit the same critical salt dependence against fusion as do microbubbles.

Only repair of pulmonary surfactant by either artificial (steroid) or alternatively natural depletion of the virus plus natural lung repair will have directly improved natural oxygen uptake sufficiently.

## Carbon dioxide bubbles destroy viruses and bacteria: $CO_2$ a saint not sinner

Warm $CO_2$ bubbles are extremely efficient in killing viruses and other pathogens, especially in salt water above the critical 0.17 M salt; so efficient that they are being used in piggery feedlots to sterilise wastewater (Adams, 1948; Garrido *et al.*, 2016, 2018; Sanchis *et al.*, 2019; El-Betany *et al.*, 2020; Garrido and Jin, 2020). Free radicals produced by cavitation that produces nanobubbles seem to be the culprit. These effects have been known but ignored since 1948 (Adams, 1948).

Egyptian mummy preservation against pathogens used natron, that is, sodium bicarbonate. Proteins adhering to pots are removed by surface nanobubbles of sodium carbonate which cuts phosphate bonds.

### *Interplay between the endothelial surface layer, COVID and sterilisation by $CO_2$*

#### Physiology of the endothelial surface layer and glycocalyces (Reines and Ninham, 2019; El-Betany et al., 2020)

The glycocalyx of predominately sulphated glycosylated polymers that is ubiquitous and lines the veins of all mammals (chondroitin and heparin sulphate and sodium hyaluronate). It is about 50 nm thick. On top of that a much larger 'exclusion zone' sits between the glycocalyx and the passing blood flow. The exclusion zone is about 1 micron thick, 20 times thicker and repels blood cells, T-cells, bacteria and lipoproteins.

In mammals it comprises sparce thin conducting fibres separated by $CO_2$ nanobubbles in a foam. This is produced by passage of molecular $CO_2$ from metabolism through the molecular frit that is the glycocalyx. The nanobubbles are constantly swept away by the passing blood stream and out via the lungs. Free radicals in the foam destroys viruses like COVID-19 (see references correspond to

section 'Chemical reactions, enzyme reactions, and more nanobubbles').

This major organ shares much with and is analogous to the fuel cell polymer Nafion, which provides hydrogen ions and hydrogen. The permeable bicontinuous membrane of Nafion and its very large 'exclusion zone' is, if we like, very much analogous to the ESL-glycocalyx. But Nafion is written on a larger canvas because of the very high hydrophobicity of its Teflon backbone, as compared to that of the biological analogue which is the glycocalyx. The internal permeating conducting sulphated water channels are similar. The conductivity and connectivity of the whole body glycocalyx structure of veins is probably the origin of and provides respectability to acupuncture. Similarly, the exothelial surface layer of fish again mirrors the same nanostructures and must be the source of the very high voltage electrical discharges used by 'electric' eels and fish.

Unlike lipids, the various states of self-assembled microstructure formed by the glycocalyx, and other polymers are less predictable but being necessarily permeable and bicontinuous. They are known to form nanochannels passing through the cell membrane connecting the inside and outside, as revealed by some elegant work of Arkill *et al.* (2011). *Parallels with Nafion point to its role in setting pH, which is also previously unknown.*

### The Endothelial Surface Layer

The ESL is a micron thick layer that sits on top of the glycocalyx of all venous tissue. Its structure and function were unknown until 2 years ago and appears void of matter (Reines and Ninham, 2019).

It comprises a dynamic foam of nanobubbles separated by sparce strands of polymers that form the protective glycocalyx (Arkill *et al.*, 2011; Reines and Ninham, 2019). These polymers are mostly hydrated heparin and chondroitin sulphate and sodium hyaluronate. $CO_2$, produced via metabolism, passes through the porous medium, a molecular frit which is the glycocalyx, and assembles into nanobubbles of $CO_2$. Red cells, T-cells, bacteria, LDLs are all repelled by the ESL. The nanobubbles of $CO_2$ of the thick ESL foam are constantly produced sloughed off to join the bloodstream and are replenished.

Those in the bloodstream make their way back to the lungs and exit with water vapour via the lung surfactant.

A consequence of this fact is that covid viruses exiting venous tissue via the glycocalyx must run the gauntlet of attack by $CO_2$ nanobubbles of the ESL that produce fragments of the virus.

### Long COVID

*Several experiments (Adams, 1948; Arkill et al., 2011; Garrido et al., 2016; 2018; Reines and Ninham, 2019; Sanchis et al., 2019; Garrido and Jin, 2020; Queisser et al., 2021) suggest that simultaneous destruction of the ESL combined with fragments of the virus coat proteins destroyed by $CO_2$ nanobubbles are the sources of Long COVID.*

The implications are several:
The virus will have very short shrift from children, whose generally greater physical activity compared with adults, generates sufficient $CO_2$ to ensure a healthy, thick ESL damaging to the virus. Consequently, children will also be more likely to suffer from long covid due to extra ESL damage and fragments of viral coat. The nanobubbles are of the same dimensions as those of oxygen/nitrogen formed in the lung surfactant, of the order of 30–40 nm. Older people have damaged ESLs and have low metabolic rates, with

lower $CO_2$ production. Consequently, this avenue of protection is absent for older patients.

Some very remarkable experiments of Oberleithner (see Reines and Ninham, 2019, section 'high' versus 'low' salt effects on the ESL) have shown that collapse of the ESL due nanobubble fusion (and embolism) will occur if salt concentration drops below the (effective) critical 0.17 M concentration. There are some serious implications of this too. Supply of saline to ill patients is via solution that contains only NaCl salt. Apart from stress on kidneys, this will, via the Oberleithner effect, damage and remove the protection of the ESL.

### Nanobubbles of $O_2/N_2$. further remarks

Awareness of the existence of and the stability of nanobubbles with or without absorbed salt and protein or lipid molecules above the critical (physiological) concentration is very new (Kim *et al.*, 2001; Bunkin *et al.*, 2011; Reines and Ninham, 2019). A consequence is that oxygen, nitrogen and $CO_2$ can all circulate as nanobubbles. They are a source of free radicals and mimic specific enzymatic activity (Karaman *et al.*, 1996; Gudkov *et al.*, 2019; Lee *et al.*, 2019; Bunkin *et al.*, 2020; Fang *et al.* 2020).

Delivery of $O_2/N_2$ to blood cells from lung surfactant is probably via lipid coated nanobubbles rather than molecular oxygen. These might fuse with red cell membranes. The more likely event is that partial transfer (partitioning) of oxygen from adsorbed nanobubbles to haemoglobin in red cells is via chemisorption and that both haemoglobin and free nanobubbles of oxygen (are available for metabolism). Nitrogen does not bind to haemoglobin strongly.

The nanobubbles often are highly reactive and carry out many pseudo-enzymatic reactions. Typical is the arginine paradox (Taylor *et al.*, 1995). See also effects of hydrophobic cavitation (Ninham *et al.*, 2016; Lo Nostro and Ninham, 2020; Ninham and Lo Nostro, 2020; Taseidifar *et al.*, 2020). The universality of nanobubbles in liquids and their probable intimate involvement in all reactions, chemical and enzymatic and crystallisation is a very new development in physical chemistry (Ninham *et al.*, 2016; Ninham and Lo Nostro, 2020; Taseidifar *et al.*, 2020).

*The self-assembly of dissolved gas molecules into nanostructures mirrors the same association processes that underlies the self-assembly of surfactants and lipids into micelles and bilayers. With gas there appears to be a critical nanobubble concentration of salt at which nanobubbles form. Longer lived entities like micelles from surfactants (Ninham et al., 2016, 2017a, 2017b; Ninham and Lo Nostro, 2020) They are a foriori stabilised by adsorption of proteins and other constituents of the biological milieu.*

*The low solubility of $O_2$ and $N_2$ is 11.92 mL/kg and 6.352 at $20°C$ from moist air at 1 atm total pressure, respectively. So, delivery via nanobubbles is an effective gas transfer mechanism that molecular models cannot accommodate).*

**A further comment**.
The fog attending the hidden role of dissolved gas and its organisation lifts a little if we consider the following. The effect of impurities on tensile strength is well understood in solid state physics (explained quantitively by a theory of Griffiths). Dissolved gas is the reason the tensile strength of water is a hundred times less than that estimated by molecular bonds alone. This effect can be exploited practically to prevent propeller cavitation (Ninham and Pashley, 2020; Taseidifar *et al.*, 2020).

## Problem: Restriction enzyme action

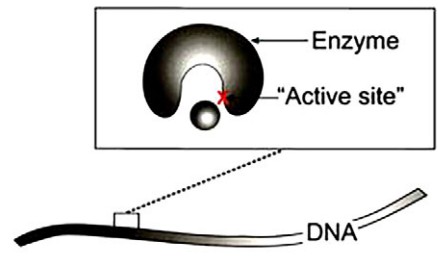

Cuts DNA at precise palandromic sequence

**Fig. 5.** Cartoon of the problem of restriction enzymes reproduced (Kim *et al.*, 2001).

### Chemical reactions, enzyme reactions, and more nanobubbles

The literature classifies restriction enzymes into four or five types. Except for type II, all others are assumed to require some 'helper' molecules like ATP.

How enzymes catalyse reactions is still an open question, particularly where the required substantial energy to cut a phosphate bond comes from. If type II does not need ATP, its energy source must lie elsewhere. Such a mechanism has been identified.

Here, for example is some data on EcoR1, an enzyme that cuts linear DNA (Kim *et al.*, 2001). It requires $Mg^{2+}$. This is necessary to set the curvature and 'hydrophobicity' of the active site, a cartoon of which follows. The enzyme can work also with other divalent ions $Mn^{2+}$, $Ni^{2+}$ and $Ca^{2+}$.

This flexibility suggests that a physical mechanism is involved (Fig. 5).

The cutting efficiency of the enzyme as a function of different salts and buffers are shown in (Fig. 6). This phosphate and cacodylate buffers are supposed to be at pH 7 and not affected by salt. They do, for reasons above (The buffer anion adsorbs at the enzyme surface and alters its 'hydrophobicity').

Consider the results in the graphs that show per cent DNA cutting on the Y-axis versus salt concentration on X-axis.

The buffers (4 mM) are phosphate and cacodylate at pH 7.5. The Hofmeister series reverses with change in buffer: a Cation variation b Anion variation. The results are inexplicable with classical theory of pH and buffers. Note peak cutting efficiency at 0.17 M salt, the same ionic strength as for blood.

One figure plots the cutting efficiency as a function of varying anion. The other varies cation. Each figure compares different buffers. Efficiency peaks at the magic 0.17 M for bubble–bubble fusion inhibition, and where we know from the ESL studies that nanobubbles are stable. Not shown are unpublished experiments by BWN which found that on addition of vitamin C, a known free radical scavenger, the enzyme ceases cutting.

The inference is that the source of the energy is hydrophobic cavitation. This is well known experimentally from many surface forces measurements between hydrophobic molecularly smooth surfaces. Spontaneous cavitation occurs below about 50 angstroms separation. This physical property provides a means whereby nature harnesses all the weak van der Waals forces cooperatively to produce the required energy.

It depends on availability of dissolved gas to form nanobubbles. Anyway, the peak at 0.17 M gives the game away. For other enzymes it would be silly to pass over and ignore such a mechanism for the source of the necessary energy to drive the catalysis. It seems reasonable to postulate that ATP may be involved in preparing the enzyme's particular DNA surface-demethylation and other grooming prior to execution.

Now as regards to hormones regulating and stimulating enzyme activity, Feng *et al.* (2019) provides a hint. They explain how ATP and other factors induce KINKs in the DNA target to expose hydrophobic target sites (see also Pingoud and Jeltsch, 2001).

With hormones, it may be reasonable to presume that the specific DNA hormone pairing surface puts the now dressed DNA into a better substrate form for more efficient cutting.

This mechanism will operate globally and has been ignored.

### The language of shape. Self-assembly of lipids

A major conceptual lock lies in neglect of the role of surfactants and lipids. Without cell membranes, organelles like mitochondria, and lung surfactants would not exist. The role of lipids has been ignored in favour of proteins. The omission is almost as catastrophic as the omission from classical theory of dissolved gas.

An up-to-date account of the situation is given by Hyde *et al.* (1997)and Ninham *et al.* (2017*a*, 2017*b*). We use the words surfactants to include single chained and double chained lipid amphiphiles.

Several remarks may be useful. The theory of self-assembly of amphiphilic molecules has its roots in statistical mechanics. It predicts, via simple molecular parameters extracted from knowledge of molecular forces, size, and shape, what kinds of aggregates (phases) form.

It is equivalent to thermodynamics used to describe ordinary liquids, gases and solids characterised by pressure, temperature, and density. But here these are replaced by renormalised variables that are more useful. These variables are local *curvature* at the head group-hydrocarbon tail interface, set by intramolecular forces; and *global packing constraints* set by interaggregate forces; and temperature (Hyde *et al.*, 1997; Ninham *et al.*, 2017*a*, 2017*b*).

The interfacial curvature, or surfactant parameter v/al increases with increased chain length l, or volume v, or head group area, a. The solution of surfactants transits from aggregates that are small spherical micelles which grow to cylinders, to single walled vesicles to multi walled vesicles (liposomes).

An alternative progression is from vesicles to bicontinuous aggregates of cubic symmetry depending on another characteristic parameter, hydrocarbon chain 'stiffness'.

It allows one to explain a range of biological phenomena from detergency, local and general anaesthesia, missing parts of transmission of the nervous impulse, induced immunosuppression of T cells and more (Ninham *et al.*, 2017*a*, 2017*b*). Its extension to include oils or cholesterol that adsorb into the lipid tails allows one to predict the microstructure of microemulsions. These multi-component biomimetic equilibrium systems can have variations of conductivity of 10 orders of magnitude and 5 or more orders for viscosity (Mitchell and Ninham, 1981; Zemb *et al.*, 1987; Knackstedt and Ninham, 1994; Knackstedt and Ninham, 1995; Hyde *et al.*, 1997; Ninham and Lo Nostro, 2010; Evans *et al.*, 1986; Ninham *et al.*, 1987; Allen *et al.*, 1987).

Once this language of shape is understood, with the parallel subtle language of molecular forces, a unified and systematic world view on matters like drug action and how to change self-assembled

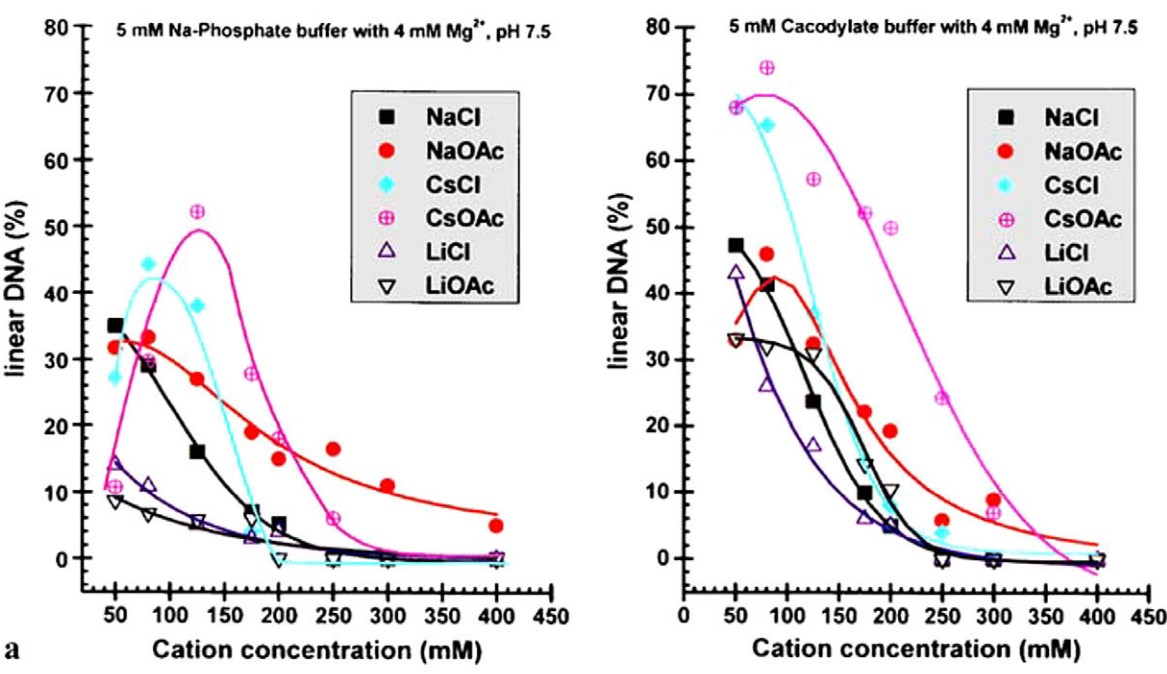

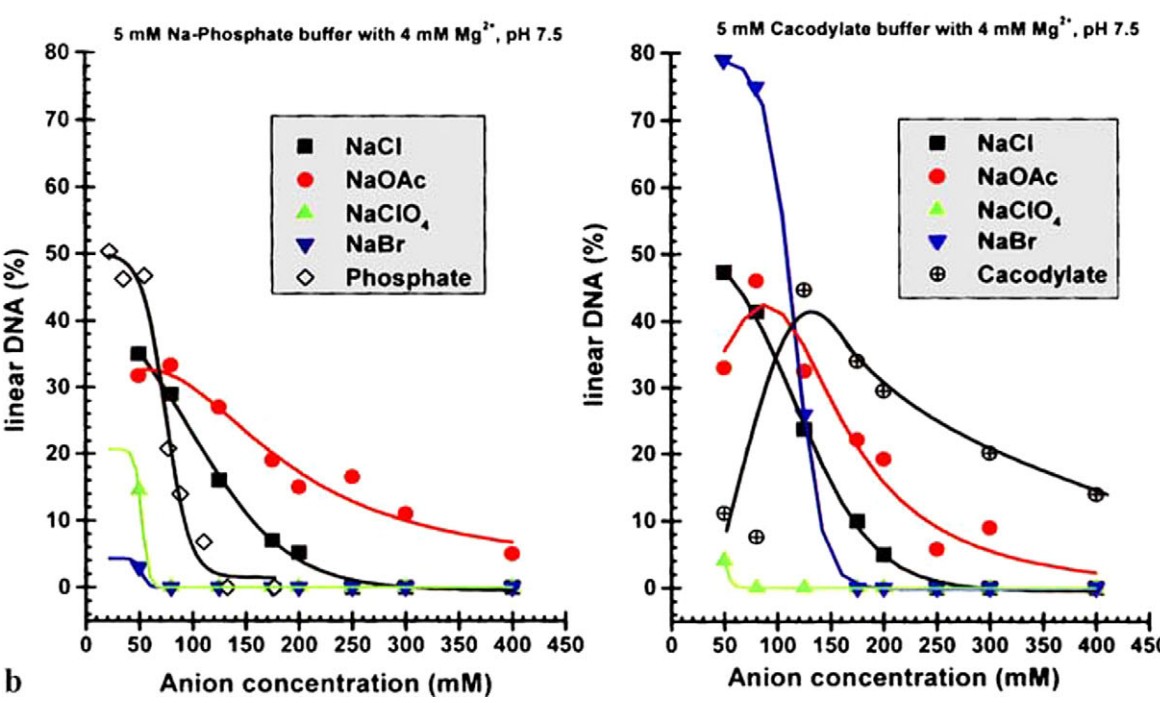

**Fig. 6.** Efficiency of a standard restriction enzyme in cutting DNA as a function of salt and salt type.

structures comes into sight. What seems almost magical changes in properties becomes obvious.

### Lung surfactant self assembly: The monolayer myth

The same principles of self-organisation hold for lung surfactants, but with some differences. These have to do with typical length scales.

There is another conceptual lock that prohibits progress in making sense of lung surfactant function that we now discuss. The major lipids are double chained di-palmitoyl phosphatidyl choline (DPPC) 90%,and phosphatidyl glycerol (PG).

As an aside, we remark that bio membranes contain mixtures of many lipids normally. The presence of only two, as for lung surfact-ants, and especially DPPC and PG is unique for cell membranes that

usually have many components lipid and proteins. The reasons are many, as described below. But it is also necessary so that the containers of nanobubbles of gas to be delivered via the lungs do not leak!

There is an enduring notion that at the air water interface the lipids DPPC and PG must present a monolayer. This was supposed to lower the interfacial tension sufficiently to enable breathing.

This is not so. It was shown to be energetically impossible by Bangham many years ago (Bangham *et al.*, 1979; Bangham, 1992, 1995).

The myth derives from earlier *primitive theories* of self-assembly. Monolayers in equilibrium do exist in equilibrium with micelles, for soluble single chained surfactants. But not always. For insoluble double chained surfactants or lipids phospholipids like the DPPC and PG and cholesterol of lung surfactant, the story is different. The hydrocarbon tails are (C16) (Bunkin *et al.*, 2011). The double chains are necessary to form bilayers. The chain length must be C16 to maintain chain fluidity in the required temperature range. Surface layers of such a system in water or electrolyte must be a few bilayers, in equilibrium with single (Bangham, 1992; Ninham *et al.*, 2017a, 2017b) or few walled vesicles, typically about 200 nm diameter, or usually multiwalled vesicles. The thickness of the bilayers is about 2–3 nm and similarly for the water layers.

For a liposome, a multiwalled vesicle, the radii of the interior bilayers decrease as one proceeds to the interior. There comes a point when, as shown in Fig. 7, that the outside chains can stretch no more, and the inside chains can no longer be compressed further. *Note that a curved bilayer in a vesicle or liposome is necessarily asymmetric.* The tension can be relieved by transiting to a different packing arrangement where the average curvature of the interface of a bilayer remains at zero, that of a planar bilayer, but the product of curvatures, the Gaussian curvature varies continuously over the surface. These simple principles, which flow from packing constraints only, come into play also for lung surfactant structure, both on inspiration and expiration, for which there are different structures. For a single bilayer, the constraint can most easily be visualised and demystified in the picture (Fig. 8) – a Pringle, a potato chip forms due to the same kind of constraints.

The water channels between the red and blue opposite sides of lipid bilayer faces are about 15 angstroms thick only. These so-called cubic phases (cubic symmetry) form naturally and can transit from one form to another with little or no energy (Fig. 9).

Such organelles like mitochondria are ubiquitous inside biological cells (see Hyde *et al.*, 1997, *The Language of Shape for 700 + images*) and bicontinuous for obvious reasons.

As bulk phases they are very viscous. Alternatively, and germane to the present situation, when additional components are added or physio chemical conditions that change head group and hydrocarbon tail forces, the interior of the multilayered lamellar phases collapses to a water filled interior. This situation probably occurs with pneumonia.

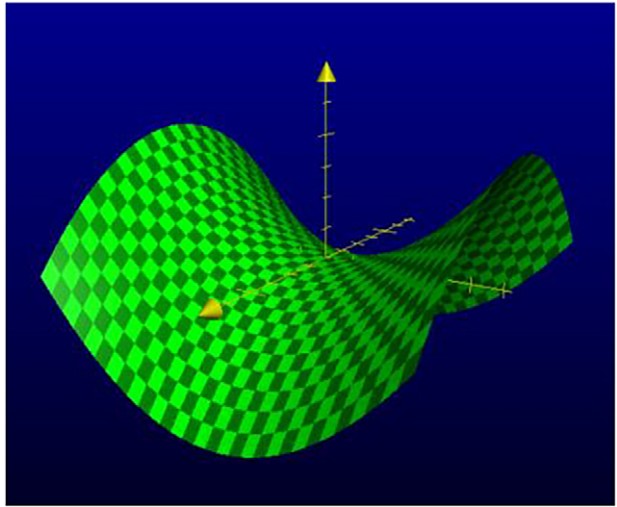

**Fig. 8.** With multilayered liposomes, the interior collapses to a bicontinuous phase of cubic symmetry with separated channels. Mitochondria are typical examples of these ubiquitous cell organelles.

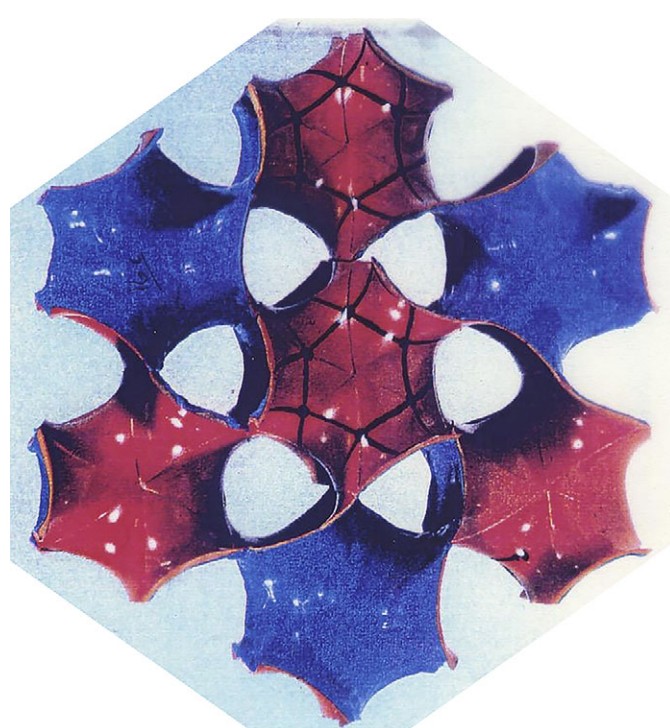

**Fig. 9.** Image of cubic phase of phospholipids courtesy of Stephen Hyde.

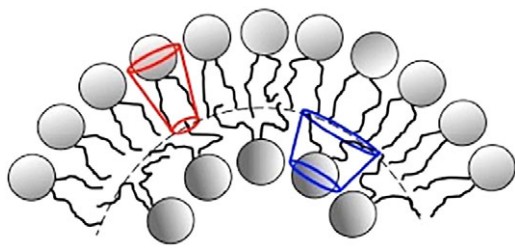

**Fig. 7.** Asymmetry of the curved bilayer of a vesicle.

These things emerge naturally and easily and quantitatively from theory. They depend on curvature at the lipid water interface determined by molecular forces, together with global packing constraints. The latter determine the proportion of each phase that exists, as a function of component ratios and temperature.

But for normal conditions of operation of lungs, matters are more complicated due to the need to containerise oxygen and nitrogen. For the double chained lipids of the lung surfactant, the surface phase is not a hydrophobic monolayer. This has been verified by neutron scattering (Follows *et al.*, 2007). It looks more like the Figure 10 below (from a paper visualising the air water

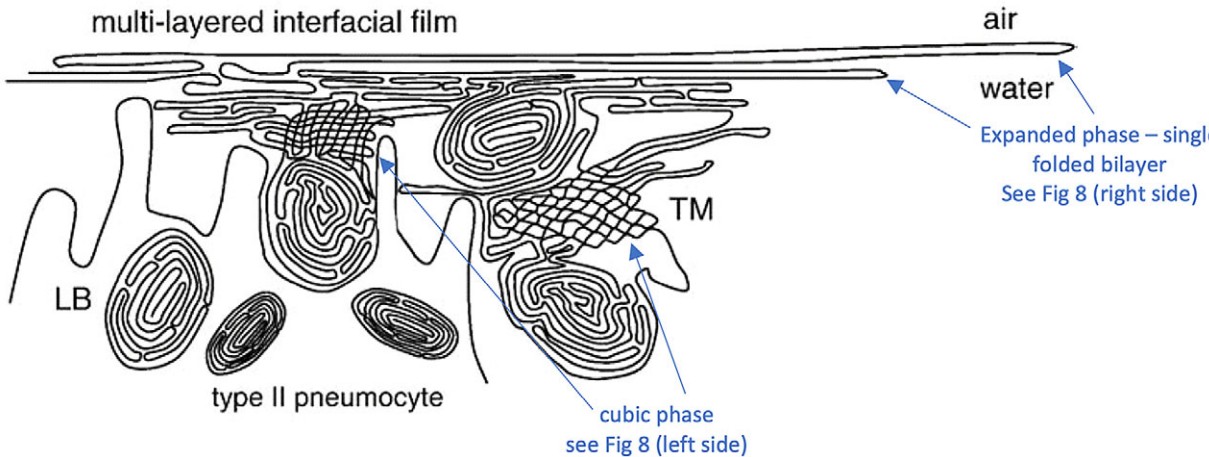

**Fig. 10.** Schematic of the 2 different lung surfactant phases (expiration, inhalation) from Perez-Gil (2008).

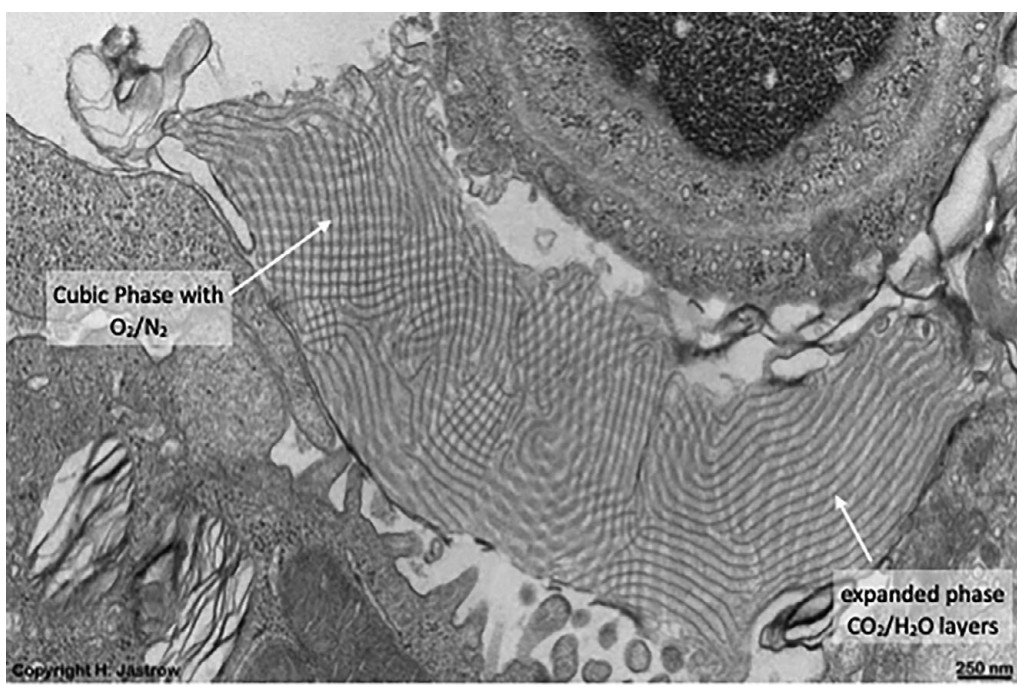

**Fig. 11.** TEM of lung surfactant surrounding capillary. Note cubes containing $O_2/N_2$ (cf. Fig. 12). The sides of the cubes are in fact two facing bilayers separated by channels of water (Courtesy of Holger Jastrow, http://www.drjastrow.de/Demo/Lung.jpg; tubular myelin (rat)).

interface of the lung surfactant from the pioneering work of Perez-Gil (2008)), Fig. 10.

The difference between this conceptualisation and the real lung surfactant configuration at the air water interface is this: The 10% fraction of lipid PG is associated and separated into regularly spaced microaggregates within the dominant DPPC bilayer (see *Figs 13b* and *13c*. It is these softer sub-regions that curve to form the air containing nanocontainers).

### Giant vesicle and other lipid phases: Consequences for lung pathologies

That this could represent the natural state of the lung surfactant bilayer at the air water interface was convincingly demonstrated by

Gershfeld (1989*a*, 1989*b*) on model double chained lipids over 30 years ago. It was so astonishing a result, against the zeitgeist, that it was as ignored, as was the existence of nanobubbles.

Gershfeld did careful Langmuir–Blodgett trough work on the surface phase of some double chained (membrane forming) lipids. He found that instead of a hydrophobic monolayer of the lipid at the air-water interface in equilibrium with a subphase in the bulk water beneath of multiwalled vesicles (liposomes) something different was going on. Above a critical temperature, the surface layer is still a bilayer. But the subphase exploded into giant single walled vesicles of the size of biological cells. The critical temperature is the temperature at which cell membranes from the animal exists. Further, in several other works he was able to link several pathologies to precisely this

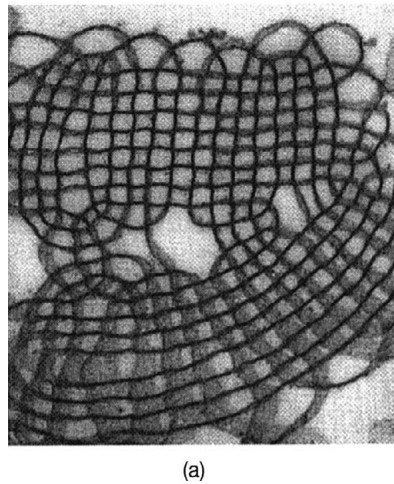 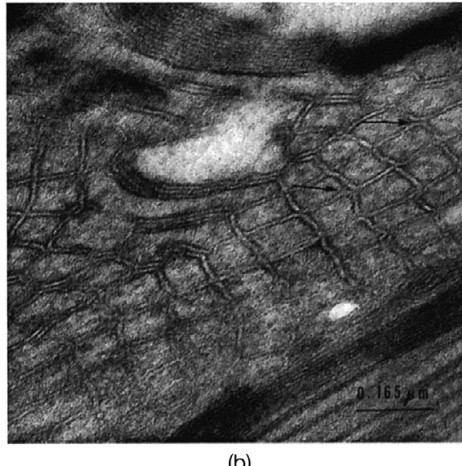

(a)                                                                                           (b)

**Fig. 12.** (*a*) TEM of foetal rat lung (Young *et al.*, 1992; Sanderson and Vatter, 1997) and (*b*) TEM of rat lung (Sanderson and Vatter, 1997; Andersson *et al.*,1999) showing tubular myelin in cubic phase of lung surfactant consistent with Fig. 10 (lower) and Fig. 11 (left side) also found in Andersson *et al.* (1999).

phenomenon. Gershfeld's work was ignored! It was confirmed much later by neutron scattering by Follows *et al.* (2007).

This apparently universal critical phase behaviour of lipids. Biological cell membranes could not exist without it. Such giant single vesicles folded with a spacing of 50 nm are visible in Figs. 10 and 11 and are the expansion phase of the lung surfactant DPPC and PG mixture of lipids, sans SPs.

From that point of view our nanostructured lipids assisted by the complication of SP, are, at least in the expansion phase, the same giant vesicles.

It is reasonable then to speculate what happens if a patient gets too cold, dropping by only a few degrees; and especially if the lung surfactant is congested with particles. The critical giant vesicle state of a freely expanded bilayer at the interface will collapse and deposit on the solid surfaces of the particles, be it mineral – fibreglass or asbestos or tobacco smoke.

If it forms multiwalled vesicles they will have water layer spacing between the bilayers of 2 nm, not 50 nm. So, a lot of lung surfactant would be taken out of the system and is not available for gas delivery. At the same time there will be a real surface tension, making it difficult to breathe; and there will be large pools of water around for bacteria to go swimming about and places to adsorb on and reproduce. Where removal of excess water is necessary, the Heliox process is a proven candidate, while salt particles, that nucleate nanobubbles, have potential (Ninham and Pashley, 2020; Reuben and Harris, 2004).

In another conception depending on the physical characteristics of the particles in the lung, the unfolding lipid can deposit and form not liposomes, but actual cubic phases of the lipid as in Fig. 9. These hold the same amount of water as the liposomes but are extremely viscous. They are ubiquitous in cells in biology (Hyde *et al.*, 1997; Ninham *et al.*, 2017*a*, 2017*b*; Almsherqi, 2021; Angelova *et al.*, 2021; Deng and Angelova, 2021; Zhuo *et al.*, 2021; Ball *et al.*, 2022).

Other states of the lipids are possible. One is a particle of a one or two layers of a large bilayer with inside a collapsed multilayer (Ninham *et al.*, 2017*a*, 2017*b*). All these would lead to different pathological states. One cannot consider them as speculative. They exist as surely as the sun rises daily. It would be straightforward to make a phase diagram with DPC and 10% PG with different particle contaminants. They are the only components as the lung surfactant is expanding.

There must be a correlation between different lung diseases and the phase behaviour of the lipids in the presence of different contaminants or scarring. For the predominant double-chained bilayer forming lipid of lung surfactants, di palmitoyl phosphatidyl choline (DPPC) the expanded phase of lung surfactant looks schematically like the top section of Fig. 10. A single lipid layer is at the air–water interface and folds into contiguous multiple layers beneath. The spacing is more like 50 nm, much greater than the 3 nm spacing of liposomes. See also the actual image of Jastrow in Fig. 11.

A few recent surface tension measurements on a mix of real lung surfactant reconfirm that the myth is unsustainable (Nguyen and Perlman, 2018). Like the denial of nanobubbles, the surface tension myth has inhibited progress and will continue to do so. Similar conceptual locks are legion. Another example is the unexpected structure of brain lipid assemblies (Alfredsson *et al.*, 2021). Yet another omission in understanding self-assembly is the ubiquitous nature of supra self-assembly (Hyde *et al.*, 1997; Ninham and Lo Nostro, 2010; Ninham *et al.*, 2017*b*) We note that the lifetime of a short-chained monomer surfactant molecule like Sodium decyl sulphate is about $10^{-8}$ s, that of the whole micelle (aggregation number ~50), that of the whole micelle $10^{-5}$ s.

For the insoluble DPPC, the time for a lipid to flip from one side of a bilayer to another is about 3 months.

And finally, no self-respecting **hydrophilic** virus is going to adsorb from the air through a **hydrophobic** monolayer.

### Lung surfactant D and other pathologies

The biochemical/immunological roles of SP-A and D have been much studied. An excellent review of SP-D is given by Sorensen (2018). Proteins A and D are very hydrophilic. Protein D is very small, and its function was unknown until recently. However, it is speculated that loss or degeneration of this small hydrophilic protein is the cause of irreversible pulmonary fibrosis IPF which is progressive and has a non-steroid response unlike COVID or any other Lung $O_2$ uptake disease which damages the large protein A in pulmonary surfactant structure. Protein A is very large on the scale of membrane thickness. The protein component forms only 10% of the whole lung surfactant matrix. The 10% PG lipid component will phase separate into rafts (see Fig. 13*b,c*) to assist the curvature required at the corners of the intersecting planes of DPPC. But it is

not enough. The very hydrophobic proteins SP-B and SP-C are necessary to do the job (see Fig. 13*f*).

Apart from recognition of their roles as collectins, no collective biophysical aspects of their place in the scheme of things comparable with that of the lipids have been identified. Diseases like irreversible pulmonary fibrosis are maddeningly suggestive of such a role for SP-D (Greene *et al.*, n.d.). To clothe such a notion with definiteness, we can recognise that SP-D does exhibit cooperative behaviour and self-assembles into aggregates just as do lipids, but different kinds of aggregates.

The small monomer has a helical part corresponding to the lipid's hydrocarbon tails. But unlike hydrocarbon tails the helical part is rigid It self-assembles into a familiar trimeric hydrophilic aggregate. But there is a whole hierarchy of more complicated larger structural forms available to it depending on solution conditions (e.g., $Ca^{2+}$ concentration).

The trimeric form is very interesting. The word collectin already recognises that SP-D and A adsorb onto the surfaces of foreign cells and the rough surface of colloidal particles of silica, tobacco smoke or fibre glass. Bound together further by polymers of the glycocalyces of cells to which they bind, the resulting large aggregates form a mucous destined for early exit.

But in some circumstances, the adsorbed hydrophilic trimer can form extended aggregates at the surface. These aggregates can associate with the hydrophilic head groups of the minor component of the lung surfactant, phosphatidyl glycerol. Depletion of this lipid means then the gas delivery nanocontainers cannot form. The surface association of SP-D with PG on contaminant particles would build with time, progressively reducing breathing capacity and simultaneously removing SP-D from its homeostatic normal restorative role.

This (biophysical) scenario is likely to be relevant to irreversible pulmonary fibrosis once the lung surfactant structure we have described is recognised. Such effects could also be involved in the association of cancer cells into tumours with their own supply of local oxygen due to the destruction of the lung surfactant nanofoam organisation.

It is of much interest that oligomerization of SP-D is dependent upon cysteine residues at positions 15 and 20 within the N-terminal tail region. Cysteine is highly selective in binding of divalent metal ions (Ninham and Pashley, 2020). So, the oligomerisation can sometimes be reversed – and diseases like IPF or rheumatoid arthritis – by trace elements like selenium or zinc. While still only anecdotal for humans, it is well known that trace elements are essential to grazing animal health in poor soils.

There are several trials in progress on covid and selenium connections accessible on the Web.

### The real lung surfactant – Structure and machinery

The extraordinary transmission electron microscopic (TEM) image of Holger Jastrow lung surfactant in action captures the heart of the matter which we now rehearse in detail.

The first to discover the structure in 1999 were Marcus and Kare Larsson *et al.*, and the author BWN (Larsson *et al.*, 1999). It has been ignored ever since. Scarpelli (1998) recognised that lung surfactant formed a foam and some related consequences.

The parallel bilayers of 'tubular myelin' (expanded phase) are separated by water layers and $CO_2$ nanobubbles. These layers are about 40–50 nm, *thus at least 10 times thicker than water layers between phospholipid bilayers in liposomes* determined by van der

Waals forces. They must be filled with a foam of $CO_2$, and probably contain disorganised surfactant proteins A, B C, and D. It is evidently a reassembled single bilayer (tubular myelin) ready to spread out at the air lung surfactant on inspiration.

The demands made on DPPC and PG to organise themselves to deliver air in quantity – reversibly – and reorganise to expel $CO_2$, can only be met if a foam of connected nanobubbles of air is formed, as the detailed Cryo-TEM image. The anointed assistants are lung proteins A, B, C and D.

These images are almost identical with the first Cryo-TEM images of Larsson *et al.* (1999). This definitive breakthrough more than three decades ago was ignored until today.

*Several points.* No other lipid membrane is comprised exclusively of 90% of a single plane-forming lipid. It is necessary for reproducibility. The planar head groups of DPPC have dimethyl ammonium terminal groups. The methyl groups are known from NMR to bind two water molecules of hydration per methyl group, so tightly that anhydrous quaternary ammonium surfactants dehydrate phosphorous pentoxide, the most favoured drying agent!

This is why mono species of planar phosphatidyl choline lipids are potent protectants against bacteria. *The nonaqueous tightly packed surface is peculiar and almost hydrophobic.*

*The argument against a DPPC monolayer membrane forming a surface layer and accommodation for air nanobubbles falls to the ground again.*

### The machinery and mechanism of lung surfactant in detail

The size of the cubes is about 40 nm maximum; The thickness of bicontinuous aqueous phase is sufficient to accommodate the joining SP-A that holds the cubes together. It can be seen in Fig. 13e and the PEREZ-GILL picture in Fig. 10 that on reaching the white clear region of the Jastrow image how the SP-A binds to cells ACE-2 and pneumonocytes.

In doing so, it releases the nanobubbles and the cubes unravel.

To recapitulate this topologically very complex problem: The very hydrophilic six-pronged joining lego-like protein A has a key job to do. It looks topologically like Fig. 14 (surface of cubic lipid layer). That is, its job is to join eight adjacent $O_2/N_2$ cubes.

The curvature problem facing the six intersecting DPPC bilayers (8 cube faces) is solved with the assistance of hydrophobic proteins B and C; and the lipid PG and cholesterol to facilitate compatible lipid chain packing. The walls of our tetragonal foam phase are a sandwich of double bilayers with a water filling.

The structure is now bicontinuous in water (electrolyte) connecting directly to the blood. It provides an aqueous entrée for the hydrophilic virus to its receptor ACE-2, creating havoc on the way. Not least of that havoc is the removal of the small hydrophobic lung surfactants B and C.

This seems to explain why it is that small steroids like dexamethasone and linoleic acid can act as temporary palliatives.

### Virus versus lung surfactant

As the virus is pulled through, it adsorbs the curved corner regions of the lung surfactant and is disguised as a hermit crab and gives it a free entry. This destroys a local region of the ordered phase as depicted in Fig. 15d as the PG is absorbed onto the viral coat. This appears to be observed in studies of the viral coat that contain lipids (Deng and Angelova, 2021).

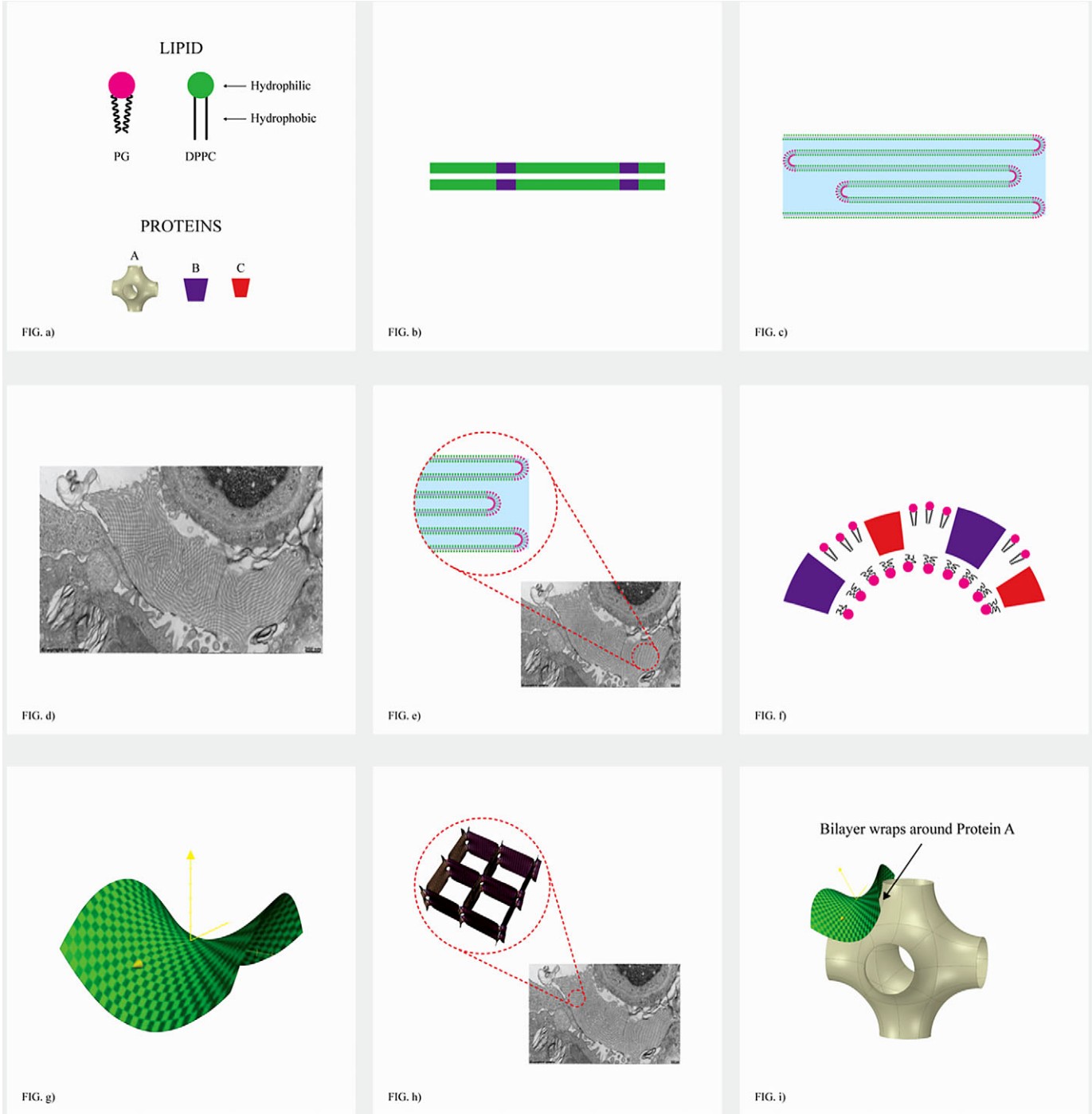

**Fig. 13.** The nuts and bolts of lung surfactant. SP-D is small, but self assembles into complex roles (see section 'Lung surfactant D and other pathologies'). (*a*) Nuts and bolts: Membrane forming lipids (double chained) (90%) DPPC [di palmitoyl (C16) 2 diphosphotidyl choline] and 10% PG [phosphatidyl glycerol], plus cholesterol (not depicted)-membrane stiffener, and four lung surfactant proteins, SP-A large and SP-D small are very hydrophilic. SP Proteins B and C are very hydrophobic. SP-D self assembles into complex structures and has a complex role described in section 'Lung surfactant D and other pathologies'. (*b*) Hydrophobic association of lipids produces membranes coloured green (DPPC) and red (PG). lLipids separate into rigid planar regions (DPPC) that repel viruses and bacteria, and softer PG regions. Such a phase separations occur in any mixture and are crucial to formation of nanostructures required. (*c*) Expanded lung surfactant is a single extended folded bilayer (giant vesicle) after expulsion of $CO_2$/water nanobubbles (and virus). (Tubular myelin) Aqueous thicknesses are about 50 nm (thicknesses of usual liposomes of membrane lipids is only 3 nm). (*d*) Actual lung surfactant showing coexistence of both expanded (Fig. 11 RHS) and (Fig. 11 LHS) (tetragonal cubic) phase nano-compartments of $O_2$/$N_2$. (*e*) Expanded lung surfactant after delivery of gas nanobubbles. (*f*) The curved bilayers at the corners of the cubic gas containers. (*f,g*) are formed of PG (hydrophilic head group) and the hydrophobic SP-B and C that allow necessary (green) saddle-shaped joining region of (*i*) made by the pseudo 'lego' SP-A. (*h*) Image shows how the bilayers of the expanded lung surfactant phase folds down to form connected compartments of a phase of cubic symmetry containing $O_2$/$N_2$ nanobubbles. (*i*) Schematic of the *topology* of the large hydrophilic SP-A, showing six arms to join corners of stacks of cubes containing gas.

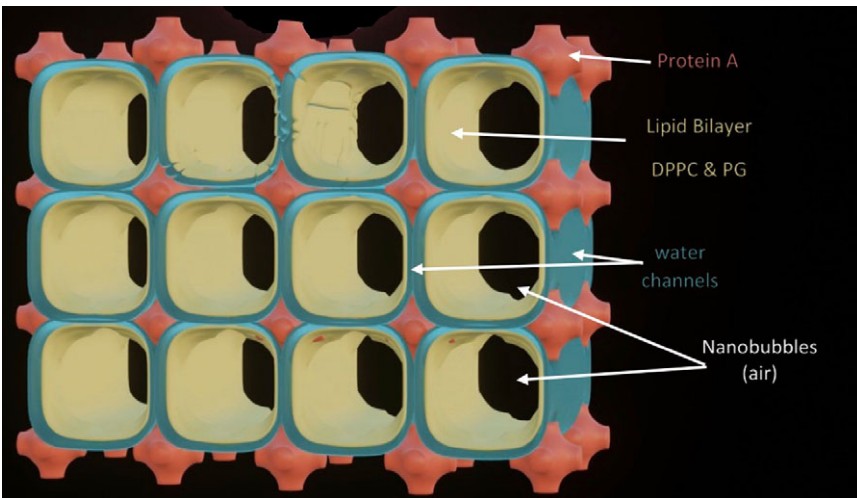

**Fig. 14.** The double bilayer water-sandwich model.

a)

b)

c)

d)

**Fig. 15.** (*a*)–(*d*) The virus opening up of the water channels. The virus is the size of 2–3 compartments ~1100 nm. This releases SP-A.

While the protein spikes have been removed from the above to reduce graphic complexity, they are likely to contribute significantly to the removal of lipids and the related proteins, especially PG and proteins B and C contained in the corner regions. A short animation depicting this process may be found here https://barry ninham.com/COVIDanimate.

## Effectiveness of viral variants

It may also explain the very different infection rates of viral mutations like the present Delta variety.

A small mutation that involves a change in hydration of viral coast proteins should be easy to accomplish. This can change the interaction (adhesion) of the virus protein coat surface to the lung surfactant PG rich curved regions, and quite dramatically. The mysterious 'furin cleavage' site of SARS-CoV-02 which some suggests COVID-19 may not have a natural origin, may be such an example that explains the extra high infectivity of the Delta variety.

*'this site is created by an insertion of four amino acids (proline, arginine, arginine, alanine - PRRA) directly at the usual coronavirus spike protein cleavage site in the junction separating the two parts of the spike protein. This insertion of 12 nucleotides of RNA is found only in this strain, whereas the S1-S2 region of the spike protein is highly conserved in most coronavirus strains'* Taken from *What really happened in Wuhan, Sharri Markson Harper Collins Publishers, page169ff.*

This apparently small change could be the difference between covid and other corona viruses, affecting its facility in affecting the binding of viral coat and the PG curved lipid region. A typical example of such an increase in adhesion is provided by a comparison of direct force measurements between bilayers of quaternary ammonium double-chained surfactants. One has a dimethyl ammonium head group, the other a methyl ethyl hydroxide ammonium.

*The differences are apparently trivial. The adhesion forces differ by an order of magnitude* (Boerema *et al.*, 1959; Gattinoni *et al.*, 2020b). The forces are physical, not chemical and are reversible, metastable not equilibrium states. This seems to make the nanobubble delivery comprehensible, leaving a reservoir of lung surfactant protein. The lung surfactant expansion and unravelling phase taking along $CO_2$ nanobubbles and water will be different not least because bicarbonate ion binds to the DPPC head group.

Any virus that has survived the gauntlet of destruction in passage through the ESL will receive further shrift in its leaving via the lung surfactant (Johnson *et al.*, 2020; Peacock *et al.*, 2021; Wu and Zhao, 2021; Zou *et al.*, 2021). We can take these speculations further by considering the paper in Reines and Ninham (2019).

The 'arginine paradox 'is a long-standing major puzzle of biochemistry.

Essentially, as there described: 'From a classical biochemical perspective, nitric oxide (NO) is produced enzymatically from the amino acid arginine acting as a substrate for nitric oxide synthase (NOS) enzymes located in or on endothelial cells. However, most measurements *in vitro* and *in vivo* contradict the classical view: the production and biological activity of NO is not determined by cellular arginine at all, but by the amount of extracellular arginine.

The 'arginine paradox' is the fact that despite intracellular physiological concentration of arginine being several hundred micromoles per litre, far exceeding the $\sim$5 μM KM of eNOS, the acute provision of exogenous arginine still increases NO production.

While a variety of explanations have been put forward none have resolved the issue'.

The ubiquity of nitric oxide production is probably due to nanobubbles of $N_2/O_2$ that adsorb at hydrophobic sites which catalyses nitric oxide production by the same process as the restriction enzyme action outlined in section 'Chemical reactions, enzyme reactions, and more nanobubbles'. A specific illustration is excess nitric oxide production in asthma, which provides a clue as to possible interventions.

*The cavity at which $N_2/O_2$ nanobubbles form is similar to that of enzymes, about 3 nm, see sections 'Chemical reactions, enzyme reactions, and more nanobubbles' and 'Lung pathologies and lipids: biophysical aspects', whereas nanobubbles for gas delivery in the PS at around 40 nm are not reactive.*

With the furin cleavage site, we have arginine in spades. We have calcium which binds to arginine available in the lung surfactant. We have available a plentitude of $N_2/O_2$ nanobubbles in the lung surfactant structure that the virus accesses on inspiration.

Bearing in mind the mechanism of enzyme action in section 'Chemical reactions, enzyme reactions, and more nanobubbles', it is not too far a stretch to conceive that the conjunction of highly attractive forces between spike protein terminal glycerol moiety and the phosphatidyl glycerol lung surfactant lipid and the arginine-rich furin site would produce a plentitude of nitric oxide and free radicals to damage the PG rich region of the lung surfactant.

Hence the palliative properties of oxygen sans nitrogen.

That would suggest a variety of approaches to handle the problem of infection better.

There is no doubt that the furin cleavage is of immense importance to the real infective and replicating powers of the SARS-CoV-2 virus.

## Possible repair mechanisms – Heliox and nitric oxide

The Shutterstock torn sweater image below might be a reasonable representation of what goes on, at least to make a point.

Note the large compartments of oxygen/nitrogen that are no longer available for delivery. And related to hypoxemia. The virus in its passage through the lung surfactant adsorbs the curved lipid regions of adjoining cubes and opens up the water channels. *So, the large damaged air-filled region left behind (Fig. 15d) also contains water.*

This explains a clinical problem known for over 70 years – the use of helium gas mixed with Oxygen can be a treatment for respiratory problems as an enabler of better oxygen uptake (Reuben and Harris, 2004)

Helium gas is insoluble and gas bubbles take up three times as much water vapour as ordinary gases.

So ventilation with helium/oxygen gas will be a very efficient non-toxic method of 'drying out' the covid damaged lung surfactant to enable repair with palliative agents like dexamethasone to replace lost SP-B and C proteins.

Helium gas bubbles have recently been used very successfully for sterilising water and desalination, exploiting the same property of helium gas (Wei *et al.*, 2020).

Of note, and the same connection, but a different process of defence is the intriguing paper (Safaee Fakhr *et al.*, 2021).

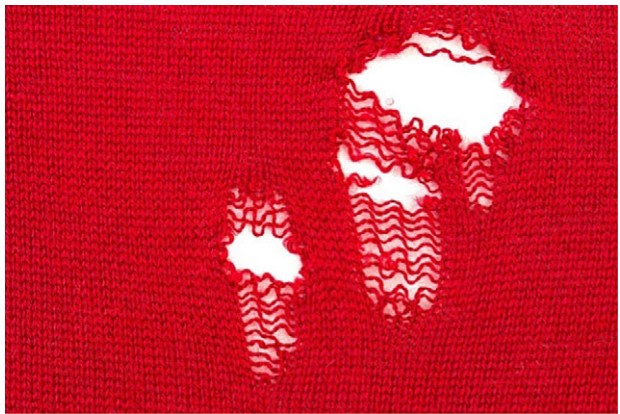

**Fig. 16.** Torn sweater is analogous to the damaged PS structure preventing single bilayer to structure phase change.

It is intriguing in the light of section 'A useful role for nitrogen'. Nitrogen adsorbed from nanobubbles of $N_2/O_2$ may renucleate at small-scale hydrophobic sites with oxygen and be catalysed to nitric oxide for metabolic processes. Given the universality of enzymatic catalysis by hydrophobic cavitation revealed in Kim et al. (2001), Pingoud and Jeltsch (2001), and Feng *et al.* (2019), it seems a probable expectation.

The unravelling torn sweater image (courtesy of Shutterstock) shows how, in one dimension, a single strand of wool can fold up and then refold to give complex domains reminiscent of gas-filled nanobubble containers (Fig. 16). How the actual reversible topological convolutions required are achieved is the province of masters of Origami. The sweater is supposed to represent a 2D cross section, a 3D structure with the 'strings' being the ends of bilayers. The analogy is the best we could do.

## Summary

### Molecular breathing: Gas exchange via changes of structure in pulmonary surfactant phases

The lattice-like nanostructure structure of pulmonary surfactant called 'tubular myelin' deep below the alveolar surface has been known to respiratory physiologists for a very long time. It had been thought of as just a highly ordered storage depot for PS lipids. So, in a theoretical vacuum, there developed the standard monolayer, low surface tension concepts that have dominated the field until now. The hiatus reached attributed gas exchange to molecular diffusion but did not explain it.

This is understandable. A systematic theory of self-assembly of lipids, surfactants, and electrolytes, the phases they form, and their nanostructures had to wait until the 1970s. It was incomplete. A wider, correct theory, that considered crucially important bicontinuous, so-called cubic phases and vesicles and liposomes in a predictive theory had to wait another 20 years. More complexity arose when a further component, oil was added. Microemulsions with their ultralow interfacial tension and astonishingly complex nanostructured phases emerged. All such literature ignored the role of dissolved gas as an active participant in forming nanostructures.

Pulmonary surfactant is a far more complex mixture DPPC and PG lipids, cholesterol, calcium in abundance and electrolytes, and 4 different hydrophobic and hydrophilic proteins. And of course, the participation of oxygen and nitrogen in one phase –realisation, carbon dioxide and excess water in another.

Gas exchange via nanobubbles that formed the alternating phases with its numerous consequences, could not have been predicted, except by taking many beautiful TEM and Cryo-TEM images at face value. The Zeitgeist was against us.

### Lung pathologies and lipids: Biophysical aspects

We have brought several new concepts to respiratory physiology.

Taken together they offer a qualitatively new picture of the two major phases of pulmonary surfactant that drive gas exchange. That occurs via nanobubbles of $O_2/N_2$ for one phase and for the other, $CO_2$, not as diffusion of molecular gas. The consequences are far reaching. The model appears to capture some of the essence of corona (spike) virus infections generally, their variability in sensitivity and, for COVID-19 particularly, its unique form of lung damage.

If our claims are valid, they ought to provide an entrée into understanding lung pathologies.

To that exercise, we now proceed, briefly, and defer a more detailed demonstration to a succeeding paper.

There are roughly two kinds of lung pathologies.

Chronic Obstructive Pulmonary Disease, COPD comprises the subset: Emphysema, Bronchitis, and possibly Asthmatic components. The subset disease components may overlap.

Interstitial Lung Disease, ILD, is a wider class that includes a number of individual pulmonary diseases Asbestosis, Pneumoconiosis, and many Fibrotic Lung disease types. What the ILD class share is direct disease and damage to the actual alveolar membrane which is by far the main factor in the disease process here.

The most striking feature of pulmonary surfactant as compared with all other cells is their constitution as predominately 90% DPPC, and 10% phosphatidyl glycerol. In all other cell membranes, the lipid component is very varied to allow endocytosis, exocytosis protein packing and compatibility, channels for ion transport and other functionalities.

With the alveoli, the 10% phase separates within the bilayer on the inhalation stage to allow gas nano compartment formation, assisted by PS proteins that together allow the complicated structure. (A consequence of the PG association within the membrane is vulnerability to and extreme variation in sensitivity to corona virus spike attacks.)

On exhaling, the nanofoam disappears with gas delivery and the lipid is left free of surfactant proteins A and D to expand to a giant folded bilayer phase (Gershfeld, 1989*a*, 1989*b*; Follows *et al.*, 2007; Fig. 10) Individual bilayers of this expanded phase separated by a large distance, about 50 nm.

### Enter colloidal particles

Confronted by colloidal particles which have large relatively flat surface areas, the bilayer forms new phases. It adsorbs and forms extensive multilayers. The intrabilayer spacing is now typical of ordinary liposomes, about 3 nm. (Whether this phase forms or not depends on molecular forces between bilayer and colloid.)

The particular spacing of layers and adsorption depends on the colloidal particle associated with the disease (anionic for silicosis, cationic for asbestosis and fibre glass related disease, hydrophobic and mixed-surface properties for tobacco smoke and other minerals. This 'precipitation' of lipid leaves large volumes of the lung space filled with water (electrolyte) and unavailable for $O_2/N_2$ uptake and accumulates $CO_2$. The large aqueous volumes allow entry by bacteria, T-cells etc., that precipitate the alveolic wars and leave a mess, pus -that further inhibits repair.

The particular phase – lung disease – that form is also temperature dependent. All lipids have a so-called Krafft temperature. At and around this temperature the lipid hydrocarbon chains become solid, and the lipids crystallise into multiple solid layers. For pure DPPC say 35 degrees, but lowered because of the 10% mix of PG and cholesterol.

But with PS the predominance of DPPC a single lipid. This crystallisation collapse is similar to that which is induced by the surface of solid particles like asbestos or fibre glass. Again, we have large volumes of water freed to allow entry of large objects like pneumonia bacteria.

### Cationic versus anionic versus hydrophobic colloids

Here several factors can contribute to differences between COPD and ILD diseases. One of these can be seen if we recall that both single and double-chained cationic surfactants in everyday use and cationic polymers, are potent immunosuppressants at very low concentrations (Ashman and Ninham, 1985; Ashman *et al.*, 1986; Ninham *et al.*, 2017*a*, 2017*b*). At higher, but still low concentration, they disrupt cell membranes. These effects in industry are very well known as is the mechanism of immunosuppression of T-cells (Ninham *et al.*, 2017*a*, 2017*b*). For cationic colloids like asbestos or fibre glass, cationic polymers are released for the surface and act as immunosuppressants besides destroying DPPC bilayers.

(Vaccines or other drugs that use double-chained cationic surfactant (lipids) vesicles as delivery agents, for example, RNA to T-cells, will adsorb the surfactants and effect immunosuppression by switching off the major histocompatibility complex (Ninham *et al.*, 2017*a*, 2017*b*). The residence time of these surfactants non-biodegradable surfactants in the body is about 6 months maximum (known and measured *in vivo*!). This may have consequences for vaccines.)

For anionic particles like silica, no such effect is expected, in line with differences between COPD and ILD diseases.

The coated anionic particles can form a slime or sludge, a well-known colloidal state that forms with so-called 'fines' in the mining industry. Large lakes of fines pose huge environmental problems as they are very viscous. Such sludges are probably characteristic of emphysema, with the best treatment Heliox (section 'The real lung surfactant – Structure and machinery', and Ziaee M, Taseidifar M, Pashley RM and Ninham BW (2020) Efficient dewatering of slimes and sludges with a bubble column evaporator. *Substantia* **4** (2 Suppl.), 89–94. https://doi.org/10.36253/Substantia-841.

For hydrophobic colloids as in cigarette smoke, and others with small hydrophobic heterogeneities, hydrophobic cavitation can occur with $O_2/N_2$ to produce active NO radicals that cause T-cell damage. The mechanism is as accounted for in section 'Chemical reactions, enzyme reactions, and more nanobubbles'. (Such effects will equally damage viruses.)

### Cystic fibrosis

Another example of the interrelation of new concepts for understanding lung pathologies is that of cystic fibrosis, a genetic disease accompanied by excessive sweating. The clue lies in the Mayo Clinic recommendation for much salt is needed for replacement in adolescents −3 tsps of salt per day, with 1 tsp = 2.3 gm.

Here is how.

The salt concentration (NaCl) in the blood is say 0.15 M. With 5 l of blood, say 0.75 moles of sodium chloride (Molecular Weight 58) per adolescent or adult. That is say 44 g.

3 teaspoons are $\times$ (2.3) = 7 gm is required to replace salt lost through sweat. Without this supplement, blood salt concentration has dropped to 0.13 M. Now recall the Oberthleiter experiments on the Endothelial Surface layer (see Reines and Ninham, 2019, section 'high' versus 'low' salt effects on the ESL).

He found if he dropped sodium salt from 0.14 to 0.13, the ESL was destroyed. In our language, the $CO_2$ nanobubbles no longer exist as stable entities because the effective salt strength drops below the critical O.17 M for bubble–bubble fusion stability. As a result, significantly less $CO_2$ nanobubbles now circulate in the blood stream and out through the expanded lung surfactant phase. It is these nanobubbles that destroy pathogens, viruses, and bacteria, in the experiments of Adrian Garrido-Sanchez. This explains why the cystic fibrosis patients are so susceptible to infections (Garrido *et al.*, 2018; Reines and Ninham, 2019; Sanchis *et al.*, 2019).

A second part of the salt loss story is the effect it has on the delivery of $O_2$ and $O_2/N_2$ nanobubbles as they become less stable. The alternative oxygen delivery system via nanobubbles – including the brain, complementary to the haemoglobin storage system, is less effective. Further, partial fusion of some nanobubbles will produce shortness of breath and bends-like symptoms.

A third part has to do with the change of relative ratios of components of salts in the plasma. Na/K/Ca/Mg/Cl/Phosphate. This affects binding particularly of calcium and potassium which is very important for matters like metabolism, in particular glucose, and T-cell recognition. It would seem a matter of extreme importance that treatment of patients should involve replacement of lost salt with balanced sea salt solution that contains all necessary ingredients, not just sodium chloride.

This is the same universal dietary story for all. In a normal western diet, even with no salt intake, the food we ingest has about 10 g of sodium chloride. This misses all the required trace elements like selenium for enzymatic activity besides the more obvious lack of K, Ca, Mg and Ca, which places heavy burdens on the kidneys. The trace element deficiency for cystic fibrosis patients due to replacement salt that lacks them is likely to be an existential matter, easily rectified.

### Nicotine

Another example of interconnected concepts at play occurs with tobacco smoking beyond that described above occurs with nicotine. It is a small molecule, $C_6 H_{12} N_2$, about the same size as octane and hydrofluorane, $C_6 F_{12}$.

Hydrofluorane is a standard anaesthetic gas, which is very hydrophobic. It partitions in excess into the hydrophobic (interior) region of the axon membrane. It changes the local curvature at the axon bilayer surface and prohibits the phase change from planar (lamella) to a mesh phase with holes that allow ion transport. This switches off conduction of the nervous impulse (Ninham *et al.*, 2017*a*, 2017*b*). After a time, the hydrofluorane leaks out and things go back to normal.

With hydrocarbons like petrol, say octane, the gas goes into the hydrocarbon region of the membrane. But it is impossible to get out, so the sniffer has a high, but stays high and brain damaged. With nicotine much the same happens as for octane, but it leaks out after a while, so you lose your 'high'.

### Denouement

A century ago, D'Arcy Thompson in his famous book *On Growth and Form* (Cambridge University Press, 1916–1968) tells us that the founders of the cell theory of biology, and of physiology believed

that their disciplines would be strengthened and enlarged by making use of the chemistry, and of the physics of the age. They did so. But puzzlingly, the gap between the biological and physical sciences worsened.

We have attempted to explain how this came about. The foundations were flawed. They were missing dissolved gas and its capacity to organise into nanobubbles, their reactivity and dependence on salt. It is a startling jump to realise that gas exchange is via nanobubbles, and the consequences of that.

The foundations were missing, in their treatment of molecular forces. They were missing the extraordinary behaviour and specificity of salts and bubble fusion. They were missing the role of $CO_2$; and even the existence of the endothelial surface layer integral to the exchange of gas, and lethal to COVID-19. And they missed the fact that the Language of Shape for lipids involved bicontinuous geometries. While we do not have the whole story, once these things are recognised much falls into place.

## Glossary

**Compliance** is a measure of the expandability of the lungs – how easily they can be inflated. It can be measured clinically by direct observation of the patient, how easily they talk and breath, and more precisely with a variety of machines including mechanical ventilators. Although the clinical phenomenon of compliance is often assumed to be directly related to the physical surface tension of individual alveoli, there is no direct correlation between the two phenomena.

**Dyspnea** is sometimes defined behaviorally as 'difficulty breathing', which can occur in either conscious or unconscious states (i.e., sleeping). We prefer to restrict the term dyspnea to 'conscious perception of mechanical difficulty in inflating the lungs on inhalation while awake'. Dyspnea is not different from 'air hunger' or 'shortness of breath'. It refers to strictly clinical pathological phenomena and is therefore psychologically and physiologically very complex.

**Pulmonary shunt** occurs when the alveoli fill with fluid. This causes parts of the lung to be unventilated (i.e., not oxygenated), although still perfused with blood. A pulmonary shunt denotes passage of deoxygenated blood from the right side of the heart to the left, without gas exchange occurring in the pulmonary capillaries.

**Pulmonary surfactant (PS)** is the complex of lipids and proteins which occur outside the glycocalyx of alveolar epithelial cells. The designation 'surfactant' is only accurate chemically to the extent that the complex is comprised of amphiphilic lipid, mainly dipalmitoylphosphatidylcholine (DPPC), and functionally whether *in vitro* interventions are able to 'squeeze out' all components of the complex except for a single monolayer of DPPC at the air-water interface. Even in this paradigmatic experimental set up, the surfactants which are most 'squeezable' and have highest surface-tension-lowering activity (artificial) are not those that are most effective clinically (natural) (Zhang *et al.*, 2011). We prefer to define pulmonary surfactant as 'the three-dimensional (3D) complex of lipids and proteins mediating gas exchange at the alveolar surface'.

**Tubular myelin (TM)** traditionally refers to the lattice-like structure often visualised by electron microscopy below the alveolar surface. TM had been presumed to constitute a storage depot unravelling to a monolayer of DPPC which had been thought to occur at the alveolar surface and to lower surface tension to near zero. However, Larsson *et al.* (1999) showed by cryo-TEM, the lattice-like structure definitely occurs precisely at the alveolar surface (where the monolayer of DPPC had been thought to occur), and that it has precisely the inter-bilayer distances long associated with TM (40–50 nm). The presumption that the lattice arrangement derives from 'tubules' was also shown to be incorrect by Larsson et al. (1999), who found that when lattice-like structures appear in EM, they are in reality comprised of 3D basket-like 'cubic phases' (as shown for PS in images in the text). It is unclear, due to artefacts inherent in traditional EM, whether the oft-noted position of TM below the alveolar surface accurately depicts an unusual position of PS or is merely an artefact of tissue processing.

**V/Q mismatch** refers to ventilation/perfusion mismatch and is the most commonly cited mechanism of hypoxemia in COVID and other diseases. This is so because gas exchange is presumed to operate by simple diffusion of $O_2$ in and $CO_2$ out. In this traditional model, the main way of explaining hypoxemia is due to either less delivery of $O_2$ to tissue (i.e., perfusion), or less $O_2$ occurring in alveoli (i.e., ventilation). As poor ventilation is not the only cause of low $O_2$ in alveoli, we would prefer the terminology 'oxygenation/perfusion mismatch'.

**Acknowledgements.** Courtesy of and thanks to Holger Jastrow. Jastrow's library of images of structures in biology is an immense World Heritage treasure trove (http://www.drjastrow.de/Homepage/Holger%20Jastrow.jpg). We wish to thank Vince Craig for helpful feedback on our discussion of nanobubbles as they relate to pulmonary surfactant function. We thank Bengt Norden and Stephen Hyde for substantial advice and improvements to the original manuscript. B.W.N. acknowledges sustained collaboration with Pierandrea Lo Nostro and many colleagues over many decades that contributed to this work. We also wish to thank Christopher Sutherland of Complete X for their design of graphical images of the 3D structure of pulmonary surfactant and the images and video showing how SARS-CoV-2 damages it.

**Open Peer Review.** To view the open peer review materials for this article, please visit http://doi.org/10.1017/qrd.2022.1.

## References

### Chapter 1

**Abbasi S**, **Bhutani VK and Gerdes JS** (1993) Long-term pulmonary consequences of respiratory distress syndrome in preterm infants treated with exogenous surfactant. *The Journal of Pediatrics* **122**, 446–452. https://doi.org/10.1016/s0022-3476(05)83390-0.

**Acharya D**, **Liu G and Gack MU** (2020) Dysregulation of type I interferon responses in COVID-19. *Nature Reviews Immunology* **20**, 397–398. https://doi.org/10.1038/s41577-020-0346-x.

**Ackermann M**, **Verleden SE**, **Kuehnel M**, **Haverich A**, **Welte T**, **Laenger F**, **Vanstapel A**, **Werlein C**, **Stark H**, **Tzankov A**, **Li WW**, **Li VW**, **Mentzer SJ and Jonigk D** (2020) Pulmonary vascular endothelialitis, thrombosis, and angiogenesis in COVID-19. *The New England Journal of Medicine* **383**, 120–128. https://doi.org/10.1056/NEJMoa2015432.

**Ahmetaj-Shala B**, **Peacock P**, **Baillon L**, **Swann OC**, **Gashaw H**, **Barclay WS and Mitchell JA** (2021) Resistance of endothelial cells to SARS-CoV-2 infection in vitro. *bioRxiv*. https://doi.org/10.1101/2020.11.08.372581.

**Alexander J and Milner AD** (1995) Lung volume and pulmonary blood flow measurements following exogenous surfactant. *European Journal of Pediatrics* **154**, 392–397. https://doi.org/10.1007/BF02072113.

**Alheshibri M**, **Qian J**, **Jehannin M and Craig VS** (2016) A history of nanobubbles. *Langmuir* **32**, 11086–11100. https://doi.org/10.1021/acs.langmuir.6b02489.

**Almsherqi ZA**, **Kohlwein SD and Deng Y** (2006) Cubic membranes: A legend beyond the Flatland* of cell membrane organization. *The Journal of Cell Biology* **173**, 839–844. https://doi.org/10.1083/jcb.200603055.

**Andersson S**, **Larsson K**, **Larsson M and Jacob M** (1999) The lung surface structure and respiration. In Andersson S (ed), *Biomathematics: Mathematics of Biostructures and Biodynamics*. Elsevier Science, Chap. 14, pp. 341–362.

**Ansarin K**, **Tolouian R**, **Ardalan M**, **Taghizadieh A**, **Varshochi M**, **Teimouri S**, **Vaezi T**, **Valizadeh H**, **Saleh P**, **Safiri S and Chapman KR** (2020) Effect of bromhexine on clinical outcomes and mortality in COVID-19 patients: A randomized clinical trial. *BioImpacts: BI* **10**, 209–215. https://doi.org/10.34172/bi.2020.27.

**Ashbaugh DG**, **Bigelow DB**, **Petty TL and Levine BE** (1967) Acute respiratory distress in adults. *Lancet* **2**, 319–323. https://doi.org/10.1016/s0140-6736(67)90168-7.

**Avery ME and Mead J** (1959) Surface properties in relation to atelectasis and hyaline membrane disease. *AMA Journal of Diseases of Children* **97**, 517–523. https://doi.org/10.1001/archpedi.1959.02070010519001.

**Bangham AD** (1992) "Surface tension" in the lungs. *Nature* **359**, 110. https://doi.org/10.1038/359110a0.

**Beachey CW** (2020) *PulmCrit: Understanding Happy Hypoxemia Physiology: how COVID taught me to treat pneumococcus.* Farkas J (ed) https://emcrit.org/pulmcrit/happy-hypoxemia-physiology/.

**Beecher HK** (1960) *Disease and the Advancement of Basic Science.* Harvard University Press. Cambridge

**Bhat R, Dziedzic K, Bhutani VK and Vidyasagar D** (1990) Effect of single dose surfactant on pulmonary function. *Critical Care Medicine* **18**, 590–595. https://doi.org/10.1097/00003246-199006000-00002.

**Bhatt RM, Clark HW, Girardis M and Busani S** (2021) Exogenous pulmonary surfactant in COVID-19 ARDS. The similarities to neonatal RDS suggest a new scenario for an 'old' strategy. *BMJ Open Respiratory Research* **8**, e000867. https://doi.org/10.1136/bmjresp-2020-000867.

**Boerema I, Meyne N, Brummelkamp W, Bouma S, Mensch MH, Kamermans F, Stern Hanf M and van Aalderen M** (1959) Life without blood. *Journal of Cardiothoracic Surgery* **13**, 133–146.

**Bostrom M, Williams DR and Ninham BW** (2001) Specific ion effects: Why DLVO theory fails for biology and colloid systems. *Physical Review Letters* **87**, 168103. https://doi.org/10.1103/PhysRevLett.87.168103.

**Bozzani A, Arici V, Tavazzi G, Boschini S, Mojoli F, Bruno R, Sterpetti AV and Ragni F** (2021) Re: "Endothelitis in COVID-19-positive patients after extremity amputation for acute thrombotic events". *Annals of Vascular Surgery* **73**, e6–e7. https://doi.org/10.1016/j.avsg.2021.01.067.

**Bunkin NF, Ninham BW, Ignatiev PS, Kozlov VA, Shkirin AV and Starosvetskij AV** (2011) Long-living nanobubbles of dissolved gas in aqueous solutions of salts and erythrocyte suspensions. *Journal of Biophotonics* **4**, 150–164. https://doi.org/10.1002/jbio.201000093.

**Busani S, Dall'Ara L, Tonelli R, Clini E, Munari E, Venturelli S, Meschiari M, Guaraldi G, Cossarizza A, Ranieri VM and Girardis M** (2020) Surfactant replacement might help recovery of low-compliance lung in severe COVID-19 pneumonia. *Therapeutic Advances in Respiratory Disease* **14**, 1753466620951043. https://doi.org/10.1177/1753466620951043.

**Cevik M, Tate M, Lloyd O, Maraolo AE, Schafers J and Ho A** (2021) SARS-CoV-2, SARS-CoV, and MERS-CoV viral load dynamics, duration of viral shedding, and infectiousness: A systematic review and meta-analysis. *Lancet Microbe* **2**, e13–e22. https://doi.org/10.1016/S2666-5247(20)30172-5.

**Cherian R, Chandra B, Tung ML and Vuylsteke A** (2020) COVID-19 conundrum: Clinical phenotyping based on pathophysiology as a promising approach to guide therapy in a novel illness. *The European Respiratory Journal* **56**, 2002135. https://doi.org/10.1183/13993003.02135-2020.

**Cherian R, Chandra B, Tung ML and Vuylsteke A** (2021) Positive bubble study in severe COVID-19 indicates the development of anatomical intrapulmonary shunts in response to microvascular occlusion. *American Journal of Respiratory and Critical Care Medicine* **203**, 263–265. https://doi.org/10.1164/rccm.202008-3186LE.

**Chiumello D, Busana M, Coppola S, Romitti F, Formenti P, Bonifazi M, Pozzi T, Palumbo, MM, Cressoni M, Herrmann P, Meissner K, Quintel M, Camporota L, Marini JJ and Gattinoni L** (2020) Physiological and quantitative CT-scan characterization of COVID-19 and typical ARDS: A matched cohort study. *Intensive Care Medicine* **46**, 2187–2196. https://doi.org/10.1007/s00134-020-06281-2.

**Chong K and Deng Y** (2012) The three dimensionality of cell membranes: Lamellar to cubic membrane transition as investigated by electron microscopy. *Methods in Cell Biology* **108**, 319–343. https://doi.org/10.1016/B978-0-12-386487-1.00015-8.

**Comroe JH** (1977a) Premature science and immature lungs. Part I. Some premature discoveries. *The American Review of Respiratory Disease* **116**, 127–135. https://doi.org/10.1164/arrd.1977.116.1.127.

**Comroe JH** (1977b) Premature science and immature lungs. Part II: Chemical warfare and the newly born. *The American Review of Respiratory Disease* **116**, 311–323. https://doi.org/10.1164/arrd.1977.116.2.311.

**Comroe JH** (1977c) Premature science and immature lungs. Part III. The attack on immature lungs. *The American Review of Respiratory Disease* **116**, 497–518. https://doi.org/10.1164/arrd.1977.116.3.497.

**Coppola S, Chiumello D, Busana M, Giola E, Palermo P, Pozzi T, Steinberg I, Roli S, Romitti F, Lazzari S, Gattarello S, Palumbo M, Herrmann P, Saager L, Quintel M, Meissner K, Camporota L, Marini JJ, Centanni S and Gattinoni L** (2021) Role of total lung stress on the progression of early COVID-19 pneumonia. *Intensive Care Medicine* **47**, 1130–1139. https://doi.org/10.1007/s00134-021-06519-7.

**Craig V, Ninham B and Pashley R** (1993a) Effect of electrolytes on bubble coalescence. *Nature* **364**, 317.

**Craig VS, Ninham B and Pashley R** (1993b) The effect of electrolytes on bubble coalescence in water. *The Journal of Physical Chemistry* **97**, 10192–10197.

**Davis JM, Veness-Meehan K, Notter RH, Bhutani VK, Kendig JW and Shapiro DL** (1988) Changes in pulmonary mechanics after the administration of surfactant to infants with respiratory distress syndrome. *The New England Journal of Medicine* **319**, 476–479. https://doi.org/10.1056/NEJM198808253190804.

**Deng Y and Angelova A** (2021) Coronavirus-induced host cubic membranes and lipid-related antiviral therapies: A focus on bioactive Plasmalogens. *Frontiers in Cell and Development Biology* **9**, 630242. https://doi.org/10.3389/fcell.2021.630242.

**Dorrington KL and Young JD** (2001) Development of the concept of a liquid pulmonary alveolar lining layer. *British Journal of Anaesthesia* **86**, 614–617. https://doi.org/10.1093/bja/86.5.614.

**DuBrock HM and Krowka MJ** (2020) Bubble trouble in COVID-19. *American Journal of Respiratory and Critical Care Medicine* **202**, 926–928. https://doi.org/10.1164/rccm.202008-3096ED.

**Fu Q, Zheng X, Zhou Y, Tang L, Chen Z and Ni S** (2021) Re-recognizing bromhexine hydrochloride: Pharmaceutical properties and its possible role in treating pediatric COVID-19. *European Journal of Clinical Pharmacology* **77**, 261–263. https://doi.org/10.1007/s00228-020-02971-4.

**Fuschillo S, Ambrosino P, Motta A and Maniscalco M** (2021) COVID-19 and diffusing capacity of the lungs for carbon monoxide: A clinical biomarker in postacute care settings. *Biomarkers in Medicine* **15**, 537–539. https://doi.org/10.2217/bmm-2021-0134.

**Gattinoni L, Chlumello D, Caironi P, Busana M, Romitti F, Brazzi L and Camporota L** (2020a) COVID-10 pneumonia: Different respiratory treatment for different phenotypes? *Intensive Care Medicine* **46**, 1099–1102. https://doi.org/10.1007/s00134-020-06033-2.

**Gattinoni L, Coppola S, Cressoni M, Busana M, Rossi S and Chiumello D** (2020b) COVID-19 does not Lead to a "typical" acute respiratory distress syndrome. *American Journal of Respiratory and Critical Care Medicine* **201**, 1299–1300. https://doi.org/10.1164/rccm.202003-0817LE.

**Gattinoni L and Marini JJ** (2021) Isn't it time to abandon ARDS? The COVID-19 lesson. *Critical Care* **25**, 326. https://doi.org/10.1186/s13054-021-03748-6.

**Goh GK, Dunker AK, Foster JA and Uversky VN** (2020) Shell disorder analysis predicts greater resilience of the SARS-CoV-2 (COVID-19) outside the body and in body fluids. *Microbial Pathogenesis* **144**, 104177. https://doi.org/10.1016/j.micpath.2020.104177.

**Goh GK, Dunker AK, Foster JA and Uversky VNC** (2022) Experimental, and clinical evidence of a specific but peculiar evolutionary nature of (COVID-19) SARS-CoV-2. *Journal of Proteome Research* **21**, 874–890. https://doi.org/10.1021/acs.jproteome.2c00001.

**Goldsmith CS, Miller SE, Martines RB, Bullock HA and Zaki SR** (2020) Electron microscopy of SARS-CoV-2: A challenging task. *Lancet* **395**, e99. https://doi.org/10.1016/S0140-6736(20)31188-0.

**Goldsmith LS, Greenspan JS, Rubenstein SD, Wolfson MR and Shaffer TH** (1991) Immediate improvement in lung volume after exogenous surfactant: Alveolar recruitment versus increased distention. *The Journal of Pediatrics* **119**, 424–428. https://doi.org/10.1016/s0022-3476(05)82057-8.

**Good RA** (1991) Experiments of nature in the development of modern immunology. *Immunology Today* **12**, 283–286. https://doi.org/10.1016/0167-5699(91)90127-F.

**Gribetz I, Frank NR and Avery ME** (1959) Static volume-pressure relations of excised lungs of infants with hyaline membrane disease, newborn and stillborn infants. *The Journal of Clinical Investigation* **38**, 2168–2175. https://doi.org/10.1172/JCI103996.

**Herrmann J, Mori V, Bates JHT and Suki B** (2020) Can Hyperperfusion of nonaerated lung explain COVID-19 hypoxia? *01 June 2020, PREPRINT (Version 1) available at Research Square.* https://doi.org/10.21203/rs.3.rs-32949/v1.

Hills BA (1999) An alternative view of the role(s) of surfactant and the alveolar model. *Journal of Applied Physiology (Bethesda, MD: 1985)* **87**, 1567–1583. https://doi.org/10.1152/jappl.1999.87.5.1567.

Hyde ST, Andersson S, Larsson K, Lidin S, Landh T, Blum Z and Ninham BW (1997) *The Language of Shape: The Role of Curvature in Condensed Matter Physics, Chemistry and Biology.* Elsevier Science.

Joseph KS (2016) A consilience of inductions supports the extended Fetuses-at-risk model. *Paediatric and Perinatal Epidemiology* **30**, 11–17. https://doi.org/10.1111/ppe.12260.

Jue T, Simond G, Wright TJ, Shih L, Chung Y, Sriram R, Kreutzer U and Davis RW (2016) Effect of fatty acid interaction on myoglobin oxygen affinity and triglyceride metabolism. *Journal of Physiology and Biochemistry* **73**, 359–370. https://doi.org/10.1007/s13105-017-0559-z.

Kitaoka H, Kobayashi H, Takimoto T and Kijima T (2021) Proposal of selective wedge instillation of pulmonary surfactant for COVID-19 pneumonia based on computational fluid dynamics simulation. *BMC Pulmonary Medicine* **21**, 62. https://doi.org/10.1186/s12890-021-01435-4.

Koopman HN and Westgate AJ (2012) Solubility of nitrogen in marine mammal blubber depends on its lipid composition. *The Journal of Experimental Biology* **215**, 3856–3863. https://doi.org/10.1242/jeb.074443.

Koumbourlis AC and Motoyama EK (2020) Lung mechanics in COVID-19 resemble respiratory distress syndrome, not acute respiratory distress syndrome: Could surfactant be a treatment? *American Journal of Respiratory and Critical Care Medicine* **202**, 624–626. https://doi.org/10.1164/rccm.202004-1471LE

Koumbourlis AC, Motoyama EK and Mutich RL (1990). Changes in lung function in premature infants with respiratory insufficiency during the first 4 weeks of life. *The American Review of Respiratory Disease* **141**, Abstract 157.

Kumar P (2020) Co-aerosolized pulmonary surfactant and Ambroxol for COVID-19 ARDS intervention: What are we waiting for? *Frontiers in Bioengineering and Biotechnology* **8**, 577172. https://doi.org/10.3389/fbioe.2020.577172.

Lang M, Som A, Mendoza DP, Flores EJ, Reid N, Carey D, Li MD, Witkin A, Rodriguez-Lopez JM, Shepard J-AO and Little BP (2020) Hypoxaemia related to COVID-19: Vascular and perfusion abnormalities on dual-energy CT. *The Lancet Infectious Diseases* **20**, 1365–1366. https://doi.org/10.1016/S1473-3099(20)30367-4.

Larsson M and Larsson K (2014) Periodic minimal surface organizations of the lipid bilayer at the lung surface and in cubic cytomembrane assemblies. *Advances in Colloid and Interface Science* **205**, 68–73. https://doi.org/10.1016/j.cis.2013.07.003.

Larsson M, Larsson K, Andersson S, Kakhar J, Nylander T, Ninham B and Wollmer P (1999) The alveolar surface structure: Transformation from a liposome-like dispersion into a tetragonal CLP bilayer phase. *Journal of Dispersion Science and Technology* **20**, 1–12. DOI: 10.1080/01932699908943775.

Larsson M, Larsson K and Wollmer P (2002) The alveolar surface is lined by a coherent liquid-crystalline phase. *Progress in Colloid and Polymer Science* **120**, 28–34.

Lei X, Dong X, Ma R, Wang W, Xiao X, Tian Z, Wang C, Wang Y, Li L, Ren L, Guo F, Zhao Z, Zhou Z, Xiang Z and Wang J (2020) Activation and evasion of type I interferon responses by SARS-CoV-2. *Nature Communications* **11**, 3810. https://doi.org/10.1038/s41467-020-17665-9.

Lo Nostro P, Ninham BW, Lo Nostro A, Pesavento G, Fratoni L and Baglioni P (2005) Specific ion effects on the growth rates of Staphylococcus aureus and Pseudomonas aeruginosa. *Physical Biology* **2**, 1–7. https://doi.org/10.1088/1478-3967/2/1/001.

Lopez L, Sang PC, Tian Y and Sang Y (2020) Dysregulated interferon response underlying severe COVID-19. *Viruses* **12**, 1433. https://doi.org/10.3390/v12121433.

Markart P, Ruppert C, Wygrecka M, Colaris T, Dahal B, Walmrath D, Harbach H, Wilhelm J, Seeger W, Schmidt R and Guenther A (2007) Patients with ARDS show improvement but not normalisation of alveolar surface activity with surfactant treatment: Putative role of neutral lipids. *Thorax* **62**, 588–594. https://doi.org/10.1136/thx.2006.062398.

Meir JU, Champagne CD, Costa DP, Williams CL and Ponganis PJ (2009) Extreme hypoxemic tolerance and blood oxygen depletion in diving elephant seals. *American Journal of Physiology. Regulatory, Integrative and Comparative Physiology* **297**, R927–R939. https://doi.org/10.1152/ajpregu.00247.2009.

Miller NJ, Daniels CB, Costa DP and Orgeig S (2004) Control of pulmonary surfactant secretion in adult California Sea lions. *Biochemical and Biophysical Research Communications* **313**, 727–732. https://doi.org/10.1016/j.bbrc.2003.12.012.

Milner AD (1993) How does exogenous surfactant work? *Archives of Disease in Childhood* **68**, 253–254. https://doi.org/10.1136/adc.68.3_spec_no.253.

Milner AD (1995) How does exogenous surfactant really work? *European Journal of Pediatrics* **154**, S5–S6. https://doi.org/10.1007/BF02155102

Milner AD (1996) Surfactant and respiratory distress syndrome. *The Turkish Journal of Pediatrics* **38**, 37–43.

Milner AD, Vyas H and Hopkin IE (1983) Effects of artificial surfactant on lung function and blood gases in idiopathic respiratory distress syndrome. *Archives of Disease in Childhood* **58**, 458–460. https://doi.org/10.1136/adc.58.6.458.

Milner AD, Vyas H and Hopkin IE (1984) Effect of exogenous surfactant on total respiratory system compliance. *Archives of Disease in Childhood* **59**, 369–371. https://doi.org/10.1136/adc.59.4.369

Mirastschijski U, Dembinski R and Maedler K (2020) Lung surfactant for pulmonary barrier restoration in patients with COVID-19 pneumonia. *Frontiers in Medicine* **7**, 254. https://doi.org/10.3389/fmed.2020.00254.

Mirceta S, Signore AV, Burns JM, Cossins AR, Campbell KL and Berenbrink M (2013) Evolution of mammalian diving capacity traced by myoglobin net surface charge. *Science* **340**, 1234192. https://doi.org/10.1126/science.1234192.

Morley CJ (1984) Effect of exogenous surfactant on total respiratory system compliance. *Archives of Disease in Childhood* **59**, 798–799. https://doi.org/10.1136/adc.59.8.798.

Nguyen TL and Perlman CE (2018) Tracheal acid or surfactant instillation raises alveolar surface tension. *Journal of Applied Physiology (Bethesda, MD: 1985)* **125**, 1357–1367. https://doi.org/10.1152/japplphysiol.00397.2017.

Ninham BW, Larsson K and Lo Nostro P (2017a) Two sides of the coin. Part 1. Lipid and surfactant self-assembly revisited. *Colloids and Surfaces. B, Biointerfaces* **152**, 326–338. https://doi.org/10.1016/j.colsurfb.2017.01.022.

Ninham BW, Larsson K and Lo Nostro P (2017b) Two sides of the coin. Part 2. Colloid and surface science meets real biointerfaces. *Colloids and Surfaces. B, Biointerfaces* **159**, 394–404. https://doi.org/10.1016/j.colsurfb.2017.07.090.

Ninham BW, Pashley R and Lo Nostro P (2016) Surface forces: Changing concepts and complexity with dissolved gas, bubbles, salt and heat. *Current Opinion in Colloid and Interface Science* **27**, 25–32.

Olmeda B, Villen L, Cruz A, Orellana G and Perez-Gil J (2010) Pulmonary surfactant layers accelerate O(2) diffusion through the air-water interface. *Biochimica et Biophysica Acta* **1798**, 1281–1284. https://doi.org/10.1016/j.bbamem.2010.03.008.

Perez-Gil J (2008) Structure of pulmonary surfactant membranes and films: The role of proteins and lipid-protein interactions. *Biochimica et Biophysica Acta* **1778**, 1676–1695. https://doi.org/10.1016/j.bbamem.2008.05.003.

Perlman CE, Lederer DJ and Bhattacharya J (2011) Micromechanics of alveolar edema. *American Journal of Respiratory Cell and Molecular Biology* **44**, 34–39. https://doi.org/10.1165/rcmb.2009-0005OC.

Pfenninger J, Aebi C, Bachmann D and Wagner BP (1992) Lung mechanics and gas exchange in ventilated preterm infants during treatment of hyaline membrane disease with multiple doses of artificial surfactant (exosurf). *Pediatric Pulmonology* **14**, 10–15. https://doi.org/10.1002/ppul.1950140104.

Reines BP (1991) On the locus of medical discovery. *The Journal of Medicine and Philosophy* **16**, 183–209. https://doi.org/10.1093/jmp/16.2.183.

Reines BP and Ninham BW (2019) Structure and function of the endothelial surface layer: Unraveling the nanoarchitecture of biological surfaces. *Quarterly Reviews of Biophysics* **52**, e13. https://doi.org/10.1017/S0033583519000118.

Reynolds AS, Lee AG, Renz J, DeSantis K, Liang J, Powell CA, Ventetuolo CE and Poor HD (2020) Pulmonary vascular dilatation detected by automated transcranial Doppler in COVID-19 pneumonia. *American Journal of*

*Respiratory and Critical Care Medicine* **202**, 1037–1039. https://doi.org/10.1164/rccm.202006-2219LE.

**Ruetsch C**, **Brglez V**, **Crémoni M**, **Zorzi K**, **Fernandez C**, **Boyer-Suavet S**, **Benzaken S**, **Demonchy E**, **Risso K**, **Courjon J**, **Cua E**, **Ichai C**, **Dellamonica J**, **Passeron T and Seitz-Polski B** (2020) Functional exhaustion of type I and II interferons production in severe COVID-19 patients. *Frontiers in Medicine* **7**, 603961. https://doi.org/10.3389/fmed.2020.603961.

**Sanchis A**, **Pashley R and Ninham B** (2019) Virus and bacteria inactivation by C0$_2$ bubbles in solution. *npj Clean Water* **2**, 4–9.

**Scarpelli EM and Hills BA** (2000) Opposing views on the alveolar surface, alveolar models, and the role of surfactant. *Journal of Applied Physiology (Bethesda, MD: 1985)* **89**, 408–412. https://doi.org/10.1152/jappl.2000.89.2.408.

**Scholkmann F and Nicholls J** (2020) Pulmonary vascular pathology in COVID-19. *The New England Journal of Medicine* **383**, 887–888. https://doi.org/10.1056/NEJMc2022068.

**Schousboe P**, **Wiese L**, **Heiring C**, **Verder H**, **Poorisrisak P**, **Verder P and Nielsen HB** (2020) Assessment of pulmonary surfactant in COVID-19 patients. *Critical Care* **24**, 552. https://doi.org/10.1186/s13054-020-03268-9.

**Sender R**, **Bar-On YM**, **Gleizer S**, **Bernsthein B**, **Flamholz A**, **Phillips R and Milo R** (2021) The total number and mass of SARS-CoV-2 virions. *Proceedings of the National Academy of Sciences of the United States of America* **118**, e2024815118. https://doi.org/10.1073/pnas.2024815118.

**Solaimanzadeh I** (2020) Heterogeneous perfusion in COVID-19 and high altitude pulmonary Edema: A review of 2 cases followed by implications for hypoxic pulmonary vasoconstriction, thrombosis development, ventilation perfusion mismatch and emergence of treatment approaches. *Cureus* **12**, e10230. https://doi.org/10.7759/cureus.10230.

**Spragg RG**, **Ponganis PJ**, **Marsh JJ**, **Rau GA and Bernhard W** (2004) Surfactant from diving aquatic mammals. *Journal of Applied Physiology (Bethesda, MD: 1985)* **96**, 1626–1632. https://doi.org/10.1152/japplphysiol.00898.2003.

**Sridhar S and Nicholls J** (2021) Pathophysiology of infection with SARS-CoV-2-what is known and what remains a mystery. *Respirology* **26**, 652–665. https://doi.org/10.1111/resp.14091.

**Tobin MJ**, **Jubran A and Laghi F** (2020) Misconceptions of pathophysiology of happy hypoxemia and implications for management of COVID-19. *Respiratory Research* **21**, 249. https://doi.org/10.1186/s12931-020-01520-y.

**Tolouian R**, **Mulla ZD**, **Jamaati H**, **Babamahmoodi A**, **Marjani M**, **Eskandari R and Dastan F** (2021) Effect of bromhexine in hospitalized patients with COVID-19. *Journal of Investigative Medicine*. https://doi.org/10.1136/jim-2020-001747.

**Veldhuizen RAW**, **Zuo YY**, **Petersen NO**, **Lewis JF and Possmayer F** (2021) The COVID-19 pandemic: A target for surfactant therapy? *Expert Review of Respiratory Medicine* **15**, 597–608. https://doi.org/10.1080/17476348.2021.1865809.

**Wang Y**, **Zhang Y**, **Chen X**, **Xue K**, **Zhang T and Ren X** (2020) Evaluating the efficacy and safety of bromhexine hydrochloride tablets in treating pediatric COVID-19: A protocol for meta-analysis and systematic review. *Medicine (Baltimore)* **99**, e22114. https://doi.org/10.1097/MD.0000000000022114.

**Yurchenko SO**, **Shkirin AV**, **Ninham BW**, **Sychev AA**, **Babenko VA**, **Penkov NV**, **Kryuchkov NP and Bunkin NF** (2016) Ion-specific and thermal effects in the stabilization of the gas nanobubble phase in bulk aqueous electrolyte solutions. *Langmuir* **32**, 11245–11255. https://doi.org/10.1021/acs.langmuir.6b01644.

**Zhang H**, **Wang YE**, **Fan Q and Zuo YY** (2011) On the low surface tension of lung surfactant. *Langmuir* **27**, 8351–8358. https://doi.org/10.1021/la201482n.

**Zhang Q**, **Bastard P**, **Bolze A**, **Jouanguy E**, **Zhang S-Y**, **COVID Human Genetic Effort**; **Cobat A**, **Notarangelo LD**, **Su HC**, **Abel L and Casanova J-L** (2020) Life-threatening COVID-19: Defective interferons unleash excessive inflammation. *Med (N Y)* **1**, 14–20. https://doi.org/10.1016/j.medj.2020.12.001.

## Chapter 2

### The Background

**Lo Nostro P and Ninham BW** (2020) After DLVO: Hans Lyklema and the keepers of the faith. *Advances in Colloid and Interface Science* **276**, 102082.

**Ninham BW** (2006) The present state of molecular forces. *Progress in Colloid and Polymer Science* **133**, 65–73. https://doi.org/10.1007/2882_051.

**Ninham BW** (2017) The biological/physical sciences divide and the age of unreason. *Substantia* **1**(1), 7–24. https://doi.org/10.13128/Substantia-6.

**Ninham BW** (2019) B. V. Derjaguin and J. Theo. G. Overbeek. Their times, and ours. *Substantia* **3**(2), 65–72. https://doi.org/10.13128/Substantia-637.

**Ninham BW and Lo Nostro P** (2010) *Molecular Forces and Self-Assembly: In Colloid, Nano Sciences and Biology*. Cambridge: Cambridge University Press.

**Ninham BW and Lo Nostro P** (2020) Unexpected properties of degassed solutions. *The Journal of Physical Chemistry* **124**, 7872–7878. https://doi.org/10.1021/acs.jpcb.0c05001.

**Ninham BW and Yaminsky V** (1997) Ion binding and ion specificity: The Hofmeister effect Onsager and Lifschitz theories. *Langmuir* **13**(7), 2097–2108. https://doi.org/10.1021/la960974y.

### Bubble–bubble fusion in electrolytes

**Craig V and Henry C** (2010) Inhibition of bubble coalescence by salts and sugars. In *XXV International Mineral Processing Congress: IMPC 2010 Proceedings, Brisbane, Australia*, ed. Conference Program Committee, CSIRO Publishing pp. 1814–1826.

**Henry CL and Craig VSG** (2009) Inhibition of bubble coalescence by osmolytes: Sucrose, other sugars, and urea. *Langmuir* **25**, 11406–11412.

**Kékicheff P and Ninham BW** (1990) The double-layer interaction of asymmetric electrolytes. *Europhysics Letters* **12**(5), 471–477. https://doi.org/10.1209/0295-5075/12/5/016.

**Nylander T**, **Kékicheff P and Ninham BW** (1994) The effect of solution behavior of insulin on interactions between adsorbed layers of insulin. *Journal of Colloid and Interface Science* **164**(1), 136–150. https://doi.org/10.1006/jcis.1994.1152.

### Classical electrolyte theory and colloid science

Also, see previous references.

**Boström M**, **Craig VJS**, **Albion R**, **Williams DRM and Ninham BW** (2003) Hofmeister effects in pH measurements: Role of added salt and co-ions. *The Journal of Physical Chemistry. B* **107**(13), 2875–2878. https://doi.org/10.1021/jp026804d.

**Ninham BW**, **Kurihara K and Vinogradova OI** (1997) Hydrophobicity, specific ion adsorption and reactivity. *Colloids Surfaces A: Physicochemical and Engineering Aspects* **123–124**, 7–12. https://doi.org/10.1016/S0927-7757(96)03794-6.

**Salis A**, **Cappai L**, **Carucci C**, **Parsons DF and Monduzzi M** (2020) Specific buffer effects on the intermolecular interactions among protein molecules at physiological pH. *Journal of Physical Chemistry Letters* **11**, 6805–6811. https://doi.org/10.1021/acs.jpclett.0c01900.

**Salis A**, **Pinna MC**, **Bilaničova D**, **Monduzzi M**, **Lo Nostro P and Ninham BW** (2006) Specific anion effects on glass electrode pH measurements of buffer solutions: Bulk and surface phenomena. *The Journal of Physical Chemistry. B* **110**(6), 2949–2956. https://doi.org/10.1021/jp0546296.

### Dissolved atmospheric gas

See references (Yurchenko et al., 2016; Reines and Ninham, 2019; Zhang et al., 2011) especially and references in section 'Nanobubbles'.

Bubble and nanobubble technologies open up a plethora new efficient and cheap technologies for desalination, sterilisation, heavy metal harvesting and removal, control of cavitation in shipping, a huge worldwide economic problem and much more.

**Ninham BW and Pashley RM** (2020) 'Outline', 'Postscript', & 'cavitation', in 'about water: Novel technologies for the new millennium'. special issue of *Substantia* **4**(2 Suppl.), 5–8. https://doi.org/10.36253/Substantia-1155.

**Taseidifar M**, **Antony J and Pashley RM** (2020) Prevention of cavitation in propellers. *Substantia* **4**(2 Suppl.), 109–117. https://doi.org/10.36253/Substantia-821.

### Nanobubbles

Also, see references in section 'Dissolved atmospheric gas'.

**Alheshibri M and Craig VSJ** (2018) Differentiating between nanoparticles and nanobubbles by evaluation of the compressibility and density of nanoparticles. *The Journal of Physical Chemistry C* **122**, 21998–22007. https://doi.org/10.1021/acs.jpcc.8b07174.

**Alheshibri M and Craig VSJ** (2019a) Armoured nanobubbles; ultrasound contrast agents under pressure. *Journal of Colloid and Interface Science* **537**, 123–131. https://doi.org/10.1016/j.jcis.2018.10.108.

**Alheshibri M and Craig VSJ** (2019b) Generation of nanoparticles upon mixing ethanol and water: Nanobubbles or not? *Journal of Colloid and Interface Science* **542**, 136–143. https://doi.org/10.1016/j.jcis.2019.01.134.

**Alheshibri M, Jehannin M, Coleman VA and Craig VSJ** (2019) Does gas supersaturation by a chemical reaction produce bulk nanobubbles? *Journal of Colloid and Interface Science* **554**, 388–395. https://doi.org/10.1016/j.jcis.2019.07.016.

**Bunkin NF, Kiseleva OA, Lobeyev AV, Movchan TG, Ninham BW and Vinogradova OI** (1997) Effect of salts and dissolved gas on optical cavitation near hydrophobic and hydrophilic surfaces. *Langmuir* **13**(11), 3024–3028. https://doi.org/10.1021/la960265k.

**Bunkin NF, Kochergin AV, Lobeyev AV, Ninham BW and Vinogradova OI** (1996) Existence of charged submicrobubble clusters in polar liquids as revealed by correlation between optical cavitation and electrical conductivity. *Colloids and Surfaces A: Physicochemical & Engineering Aspects* **110**(2), 207–212. https://doi.org/10.1016/0927-7757(95)03422-6.

**Bunkin NF, Lobeyev AV, Lyakhov GA and Ninham BW** (1999) Mechanism of low-threshold hypersonic cavitation stimulated by broadband laser pump. *Physical Review E* **60**(2), 1681–1690. https://doi.org/10.1103/PhysRevE.60.1681.

**Bunkin NF, Ninham BW, Babenko VA, Suyazov NV and Sychev AA** (2010) Role of dissolved gas in optical breakdown of water: Differences between effects due to helium and other gases. *Journal of Physical Chemistry B* **114**(23), 7743–7752. https://doi.org/10.1021/jp101657f.

**Bunkin NF, Ninham BW, Ignatiev PS, Kozlov VA, Shkirin AV and Starosvetskiy AV** (2011) Long-lived nanobubbles of dissolved gas in aqueous solutions of salts and eurythocyte suspensions. *Journal of Biophotonics* **4**(3), 150–164. https://doi.org/10.1002/jbio.201000093.

**Cooper JS, Phuyal P and Shah N** (2021) Oxygen Toxicity. [Updated 2021 July 18]. In: StatPearls [Internet]. Treasure Island (FL): StatPearls Publishing. Available from: https://www.ncbi.nlm.nih.gov/books/NBK430743/.

**Horn RG, Del Castillo LA and Ohnishi S** (2011) Coalescence map for bubbles in surfactant-free aqueous electrolyte solutions. *Advances in Colloid and Interface Science* **168**(1–2), 85–92. https://doi.org/10.1016/j.cis.2011.05.006.

**Rao Y** (2010) Nanofluids: Stability, phase diagram, rheology and applications. *Particuology* **8**(6), 549–555. https://doi.org/10.1016/j.partic.2010.08.004.

**Taylor MJ, Bailes JE, Elrifai AM, Shih SR, Teeple E, Leavitt ML, Baust JG and Maroon JC** (1995) A new solution for life without blood. Asanguineous low-flow perfusion of a whole-body perfusate during 3 hours of cardiac arrest and profound hypothermia. *Circulation* **91**(2), 431–444. https://doi.org/10.1161/01.cir.91.2.431.

**Vinogradova OI, Bunkin NV, Churaev NV, Kiseleva OA, Lobeyev AV and Ninham BW** (1995) Submicrocavity structure of water between hydrophobic and hydrophilic walls as revealed by optical cavitation. *Journal of Colloid and Interface Science* **173**(2), 443–447. https://doi.org/10.1006/jcis.1995.1345.

## Carbon dioxide bubbles destroy viruses and bacteria: $CO_2$ a saint not sinner

**Adams MH** (1948) Surface inactivation of bacterial viruses and of proteins. *The Journal of General Physiology* **31**(5), 417–431.

**Arkill KP, Knupp C, Michel CC, Neal CR, Qvortrup K, Rostgaard J and Squire JM** (2011) Similar endothelial glycocalyx structures in microvessels from a range of mammalian tissues: Evidence for a common filtering mechanism? *Biophysical Journal* **101**(5), 1046–1056. https://doi.org/10.1016/j.bpj.2011.07.036.

**Bunkin NF, Shkirin AV, Ninham BW, Chirikov SN, Chaikov LL, Penkov NV, Kozlov VA and Gudkov SV** (2020) Shaking-induced aggregation and flotation in immunoglobulin. Dispersions: Differences between water and water−ethanol mixtures. *ACS Omega* **5**, 14689–14701.

**El-Betany AMM, Behiry E, Gumbleton M and Harding K** (2020) Humidified warmed $CO_2$ treatment therapy strategies can save lives with mitigation and suppression of SARS-CoV-2 infection: An evidence review. *Frontiers in Medicine* **7**, 594295. https://doi.org/10.3389/fmed.2020.594295.

**Fang Z, Wang X, Zhou L, Zhang L and Hu J** (2020) Formation and stability of bulk nanobubbles by vibration. *Langmuir* **36**, 2264–2270. https://doi.org/10.1021/acs.langmuir.0c00036.

**Garrido A, Pashley RM and Ninham BW** (2016) Low temperature MS2 (ATCC15597-B1) virus inactivation using a hot bubble column evaporator (HBCE). *Colloids and Surfaces. B, Biointerfaces* **151**, 1–10. https://doi.org/10.1016/j.colsurfb.2016.11.026.

**Garrido A, Pashley RM and Ninham BW** (2018) Water sterilisation using different hot gases in a bubble column reactor. *Journal of Environmental Chemical Engineering* **6**, 2651–2659. https://doi.org/10.1016/j.jece.2018.04.004.

**Garrido AS and Jin L** (2020) Evaluation of the new energy-efficient hot bubble pilot plant (HBPP) for water sterilization from the livestock farming industry. *Water Resources and Industry* **24**, 100135.

**Gudkov SV, Lyakhov GA, Pustovoy VI and Shcherbakov IA** (2019) Influence of mechanical effects on the hydrogen peroxide concentration in aqueous solutions. *Physics of Wave Phenomena* **27**, 141–144.

**Karaman ME, Ninham BW and Pashley RM** (1996) Effects of dissolved gas on emulsions, emulsion polymerization, and surfactant aggregation. *Journal of Physical Chemistry* **100**(38), 15503–15507. https://doi.org/10.1021/jp960758y.

**Kim H-K, Tuite E, Nordén B and Ninham BW** (2001) Co-ion dependence of DNA nuclease activity suggests hydrophobic cavitation as a potential source of activation energy. *European Physical Journal E: Soft Matter* **4**(4), 411–417. https://doi.org/10.1007/s101890170096.

**Lee JK, Walker KL, Han HS, Kang J, Prinz FB, Waymouth RM, Nam HG and Zare RN** (2019) Spontaneous generation of hydrogen peroxide from aqueous microdroplets. *Proceedings of the National Academy of Sciences of the United States of America* **116**, 19294–19298.

**Queisser KA, Mellema RA, Middleton EA, Portier I, Manne BK, Denorme F, Beswick EJ, Rondina MT, Campbell RA and Petrey AC** (2021) COVID-19 generates hyaluronan fragments that directly induce endothelial barrier dysfunction. *JCI Insight* **6**, e147472. https://doi.org/10.1172/jci.insight.147472.

## Chemical reactions, enzyme action, nanobubbles

**Allen M, Evans DF, Mitchell DJ and Ninham BW** (1987) Interfacial tension of ionic microemulsions. *Journal of Physical Chemistry* **91**(9), 2320–2324. https://doi.org/10.1021/j100293a022.

**Evans DF, Mitchell DJ and Ninham BW** (1986) Oil, water, and surfactant: Properties and conjectured structure of simple microemulsions. *The Journal of Physical Chemistry* **90**(13), 2817–2825. https://doi.org/10.1021/j100404a009.

**Feng B, Sosa RP, Mårtensson AKF, Jiang K, Tong A, Dorfman KD, Takahashi M, Lincoln P, Bustamante CJ, Westerlund F and Nordén B** (2019) Hydrophobic catalysis and a potential biological role of DNA unstacking induced by environment effects. *Proceedings of the National Academy of Sciences of the United States of America* **116**, 17169–17174.

**Knackstedt MA and Ninham BW** (1994) Model disorder media for ternary microemulsions. *Physical Review E* **50**, 2839–2843. https://doi.org/10.1103/PhysRevE.50.2839.

**Knackstedt MA and Ninham BW** (1995) Ternary microemulsions as model disordered media. *AICHE Journal* **41**(5), 1295–1305. https://doi.org/10.1002/aic.690410524.

**Mitchell DJ and Ninham BW** (1981) Micelles, vesicles and microemulsions. *Journal of the Chemical Society, Faraday Transactions II* **77**(4), 601–629. https://doi.org/10.1039/F29817700601.

**Ninham BW, Barnes IS, Hyde ST, Derian P-J and Zemb TN** (1987) Random connected cylinders: A new structure in three component microemulsions. *Europhysics Letters* **4**(5), 561–568. https://doi.org/10.1209/0295-5075/4/5/009.

**Pingoud A and Jeltsch A** (2001) Structure and function of type II restriction endonucleases. *Nucleic Acids Research* **29**(18), 3705–3727. https://doi.org/10.1093/nar/29.18.3705.

**Zemb TN**, **Hyde ST**, **Derian P-J**, **Barnes IS and Ninham BW** (1987) Microstructure from X-ray scattering: The disordered open connected model of microemulsions. *Journal of Physical Chemistry* **91**(14), 3814–3820. https://doi.org/10.1021/j100298a018.

## Lung surfactant structure: Self-assembly of lipids

**Alfredsson V**, **Lo Nostro P**, **Ninham BW and Nylander T** (2021) Morphologies and structure of brain lipid membrane dispersions. *Frontiers in Cell and Developmental Biology* **9**, 675140. https://doi.org/10.3389/fcell.2021.675140.

**Bangham AD** (1995) Surface tension in the lungs. *Biophysical Journal* **68**, 1630–1633. https://www.ncbi.nlm.nih.gov/pmc/articles/PMC1282060/pdf/biophysj00063-0429.pdf.

**Bangham D**, **Morley CJ and Phillips MC** (1979) The physical properties of an effective lung surfactant. *Biochimica et Biophysica Acta (BBA)* **573**(3), 552–556. https://doi.org/10.1016/0005-2760(79)90229-7.

**Follows D**, **Tiberg F**, **Thomas RK and Larsson M** (2007) Multilayers at the surface of solutions of exogenous lung surfactant: Direct observation by neutron reflection. *Biochimica et Biophysica Acta* **1768**, 228–235.

**Gershfeld NL** (1989) The critical unilamellar lipid state: A perspective for membrane bilayer assembly. *Biochimica et Biophysica Acta* **988**, 335. https://doi.org/10.1016/0304-4157(89)90009-9.

**Gershfeld NL** (1989) Thermodynamics of phospholipid bilayer assembly. *Biochemistry* **28**(10), 4229–4232.

**Greene KE**, **King Jr TE**, **Kuroki Y**, **Bucher-Bartelson B**, **Hunninghake GW**, **Newman LS**, **Nagae H and Mason RJ** Serum surfactant proteins-A and -D as biomarkers in idiopathic pulmonary fibrosis. *European Respiratory Journal* **19**, 439–446. https://doi.org/10.1183/09031936.02.00081102.

**Ninham BW and Evans DF** (1986) The Rideal lecture: Vesicles and molecular forces. *Faraday Discussion Chemical Society* **81**(1), 1–17. https://doi.org/10.1039/DC9868100001.

**Sorensen GL** (2018) Surfactant protein D in respiratory and non-respiratory diseases. *Frontiers in Medicine* **5**, 18. https://doi.org/10.3389/fmed.2018.00018.

## The real lung surfactant – Structure and machinery

**Almsherqi ZA** (2021) Potential role of plasmalogens in the modulation of biomembrane morphology. *Frontiers in Cell and Developmental Biology* **9**, 673917. https://doi.org/10.3389/fcell.2021.673917.

**Angelova A**, **Angelov B**, **Drechsler M**, **Bizien T**, **Gorshkova YE and Deng Y** (2021) Plasmalogen-based liquid crystalline multiphase structures involving Docosapentaenoyl derivatives inspired by biological cubic membranes. *Frontiers in Cell and Developmental Biology* **9**, 617984. https://doi.org/10.3389/fcell.2021.617984.

**Ashman RB**, **Blanden RV**, **Ninham BW and Evans DF** (1986) Interaction of amphiphilic aggregates with cells of the immune system. *Immunology Today* **7**(9), 278–283. https://doi.org/10.1016/0167-5699(86)90010-1.

**Ashman RB and Ninham BW** (1985) Immunosuppressive effects of cationic vesicles. *Molecular Immunology* **22**(5), 609–612. https://doi.org/10.1016/0161-5890(85)90185-3.

**Ball JM**, **Chen S and Li W** (2022) Mitochondria in cone photoreceptors act as microlenses to enhance photon delivery and confer directional sensitivity to light. *Science Advances* **8**(9), eabn2070. https://doi.org/10.1126/sciadv.abn2070.

**Hopfer H**, **Herzig MC**, **Gosert R**, **Menter T**, **Hench J**, **Tzankov A**, **Hirsch HH and Miller SE** (2021) Hunting coronavirus by transmission electron microscopy—A guide to SARS-CoV-2-associated ultrastructural pathology in COVID-19 tissues. *Histopathology* **78**(3), 358–370. https://doi.org/10.1111/his.14264.

**Johnson BA**, **Xie X**, **Kalveram B**, **Lokugamage KG**, **Muruato A**, **Zou J**, **Zhang X**, **Juelich T**, **Smith JK**, **Zhang L**, **Bopp N**, **Schindewolf C**, **Vu M**, **Vanderheiden A**, **Swetnam D**, **Plante JA**, **Aguilar P**, **Plante KS**, **Lee B**, **Weaver SC and Menachery VD** (2020) *Furin cleavage site is key to SARS-CoV-2 pathogenesis. bioRxiv.* https://doi.org/10.1101/2020.08.26.268854.

**Leu A** (2020) https://www.organicconsumers.org/blog/covid-19-spike-and-furin-cleavage.

**Parker JL**, **Christenson HK and Ninham BW** (1988) Forces between bilayers of a cationic surfactant with hydroxylated headgroups: Effects of interbilayer adhesion on the interactions. *Journal of Physical Chemistry* **92**(14), 4155–4159. https://doi.org/10.1021/j100325a032.

**Pashley RM**, **McGuiggan PM**, **Ninham BW**, **Brady J and Evans DF** (1986) Direct measurements of surface forces between bilayers of double-chained quaternary ammonium acetate and bromide surfactants. *Journal of Physical Chemistry* **90**(8), 1637–1642. https://doi.org/10.1021/j100399a037.

**Peacock TP**, **Goldhill DH**, **Zhou J**, **Baillon L**, **Frise R**, **Swann OC**, **Kugathasan R**, **Penn R**, **Brown JC**, **Sanchez-David RY**, **Braga L**, **Williamson MK**, **Hassard JA**, **Staller E**, **Hanley B**, **Osborn M**, **Giacca M**, **Davidson AD**, **Matthews DA and Barclay WS** (2021) The furin cleavage site in the SARS-CoV-2 spike protein is required for transmission in ferrets. *Nature Microbiology* **6**, 899–909. https://doi.org/10.1038/s41564-021-00908-w.

**Reuben AD and Harris AR** (2004) Heliox for asthma in the emergency department: A review of the literature. *Emergency Medicine Journal (EMJ)* **21**(2), 131–135. https://doi.org/10.1136/emj.2002.003483.

**Safaee Fakhr B**, **Di Fenza R**, **Gianni S**, **Wiegand SB**, **Miyazaki Y**, **Araujo Morais CC**, **Gibson LE**, **Chang MG**, **Mueller AL**, **Rodriguez-Lopez JM**, **Ackman JB**, **Arora P**, **Scott LK**, **Bloch DB**, **Zapol WM**, **Carroll RW**, **Ichinose F**, **Berra L and Investigators NOS** (2021) Inhaled high dose nitric oxide is a safe and effective respiratory treatment in spontaneous breathing hospitalized patients with COVID-19 pneumonia. *Nitric Oxide: Biology and Chemistry* **116**, 7–13. https://doi.org/10.1016/j.niox.2021.08.003.

**Sanderson RJ and Vatter AE** (1977) A mode of formation of tubular myelin from lamellar bodies in the lung. *The Journal of Cell Biology* **74**, 1027–1031.

**Scarpelli EM** (1998) The alveolar surface network: A new anatomy and its physiological significance. *The Anatomical Record* **251**(4), 491–527. https://doi.org/10.1002/(SICI)1097-0185(199808)251:4<491:AID-AR8>3.0.CO;2-V.

**Wei R**, **Pashley R and The M** (2020) "Molecule diode" effect caused by adsorbed surfactant layers improves evaporation efficiency in the bubble column evaporator (BCE) for seawater desalination. *Journal of Environmental Chemical Engineering* **8**(5), 104303. https://doi.org/10.1016/j.jece.2020.104303.

**Wu Y and Zhao S** (2021) Furin cleavage sites naturally occur in coronaviruses. *Stem Cell Research* **50**, 102115. https://doi.org/10.1016/j.scr.2020.102115.

**Young SL**, **Fram EK and Larsson EW** (1992) Three-dimensional reconstruction of tubular myelin. *Experimental Lung Research* **18**, 497.

**Zhuo R**, **Rong P**, **Wang J**, **Parvin R and Deng Y** (2021) The potential role of bioactive plasmalogens in lung surfactant. *Frontiers in Cell and Developmental Biology* **9**, 618102. https://doi.org/10.3389/fcell.2021.618102.

**Zou W**, **Xiong M**, **Hao S**, **Zhang EY**, **Baumlin N**, **Kim MD**, **Salathe M**, **Yan Z and Qiu J** (2021) The SARS-CoV-2 transcriptome and the dynamics of the S Gene Furin cleavage site in primary human airway epithelia. *mBio* **12**(3), e01006–e01021. https://doi.org/10.1128/mBio.01006-21.
