## [Reviewer Report]

*Comments to Author*: Reines, Ninham, and Thomas

“COVID 19 as a natural experiment in physiology”

This 2-part manuscript proposes a new hypothesis, supported by extensive experimental evidence, that provides plausible explanations for numerous "COVID anomalies".The work also suggests potential pathways for COVID treatments based on the physical model.The work challenges previous orthodoxy in the field of physiology related to the surface tension model for the action of pulmonary surfactant.The work ties in multiple threads from work of the Australian school of colloid science, accumulated over recent decades, including the formation of the surfactant cubic phases.The suggestion that these phases are an essential element of the COVID story is a thread that deserves further investigation.

Overall, the 2 papers provide a provocative and compelling overview of this new paradigm illustrating the importance of the endothelial surface layer for gas absorption.I think the work should be published, but I strongly recommend 2 additions/changes prior to publication.

I would recommend a specific section near the end of the 2nd article that outlines further directions of experimental research that can test the hypotheses put forward in the 2 articles.Where do we go next in testing and fleshing out this story?Some concrete suggestions would be very helpful.

The second article is written in a provocative style.This is fine up to a point but I think that article could benefit from a serious revision that smooths out and organizes the narrative.The thread of ideas is clearly there but it needs an organizing revision.Also removal of some of the more provocative comments would not hurt the scientific content.

---

## [Reviewer Report]

*Comments to Author*: The manuscript by Ninham et al. includes two related essays related to Covid 19, and the role of pulmonary surfactants (PS) and gas exchange. In this context the authors raise the question if new physical chemistry is necessary to explain the Covid anomalies with respect to surface tension and hypoxemia. They conclude that PS and drugs enhancing PS production should immediately be applied to treat advanced Covid-19 (Essay I), and that gas exchange involves a phase change of PS and water channels (Essay II). The two essays are divided into subchapters and written in an unusual style, including many short paragraphs consisting only of 1-2 sentences that remind of philosophy. In several instances, the authors state that previous scientific findings have been “ignored” (rather than, e.g., “escaped attention”).

It is possible, and even likely that the use of PS for treating Covid has its merits. This is in fact intensely discussed in the current literature (see e.g., Tayeb et al., Nanoemulsions: Formulation, characterization, biological fate, and potential role against COVID-19 and other viral outbreaks. Colloid and Interface Science Communications, Volume 45, November 2021, 100533, https://www.sciencedirect.com/science/article/pii/S2215038221001734; Rahaman et al. Surfactant-based therapy against COVID-19: A review. Published by De Gruyter November 30, 2021, From the journal Tenside Surfactants Detergents. https://doi.org/10.1515/tsd-2021-2382; Smith et al., Biosurfactants: A Covid-19 Perspective. Front. Microbiol., 09 June 2020, https://doi.org/10.3389/fmicb.2020.01341). Since surfactants can also dissolve other membranes that then ones from the virus, the risks should not be taken lightly.

Regarding the “new physical chemistry”: extraordinary claims require thorough presentation and analysis of data to convince the audience. The broad span of fields further requires a thorough introduction to set the stage. Both are not convincing to me. Moreover, it is striking that the authors mainly cite their own work, and important statements are not always supported with references.

If the authors wish to improve their manuscript, I recommend that they start with describing the relevant anatomy, i.e. where gas exchange happens and what is known about it (including a description of the relevant glycocalyx). The authors should further give a good picture about what is known about the molecular mechanisms of virus uptake - there are several very good cryo-EM studies about this, both related to enveloped viruses in general and for SARS-CoV 2 in particular (rather than simply stating “A virus may cause molecular damage simply by adsorbing to a molecular surface”, without reference). Also of relevance to this manuscript is the role and evolution of globins (with hemoglobin serving oxygen uptake and transport to the tissues, where myglobin accepts oxygen from hemoglobin with an intricate allosteric mechanism). Words like “hypoxemia”, “dyspnea” and “pulmonary surfactants" deserve an explanation, if the readers shall follow - not everyone has a broad background from medicine to physical chemistry. Finally, I recommend that the authors publish their essays on a preprint server to get a wide range of expert comments that will help to improve the manuscript for publication in a journal.

Also the topic of nanobubbles would benefit from a proper review of the literature, including treatment of Covid (see e.g., Review of Oxygenation with Nanobubbles: Possible Treatment for Hypoxic COVID-19 Patients, by Afshari et al. ACS Appl. Nano Mater. 2021, 4, 11, 11386-11412).

I was surprised to read that elephant seals are the deepest-diving mammals, since I thought these were whales (which in fact they are). - And I do not quite understand why restriction enzymes were brought into the discussion? (a section that is not very convincing in the face of decades of advances in enzymology; quote: “No one has any idea how any enzymes catalyse reactions and where the required substantial energy to cut a phosphate bond comes from; except mumble the obligatory word ATP.”)

For the second assay, the structure of pulmonary surfactants is essential. Given the many flaws in other parts of the manuscript, I would like an EM expert to make sure that these data are reliable, and not artefacts of poorly conducted experiments. Finally, if there is “new physical chemistry”, this should be described scientifically, preferably with relevant theoretical models.

If the authors on the other hand are mainly interested in writing an essay in their unusual style, irrespective of the scientific merits, maybe they should find a publisher, who is interested in this? - I am not entirely sure what the intention of the authors is with this manuscript? Maybe they just want to boost their H-index?

---

## [Reviewer Report]

*Comments to Author*: I enjoyed very much reading this lengthy review of the importance of nanobubbles as the possible cause for covert viral damage to three-dimensional structure of pulminary surfactants in lungs.I do have a suggestion that might improve the presentation.It might help to offer an explanation why nanobubbles cause these reactions to occur.I suggest the answer might lie in the existence of an electric double layer (EDL) at the hydrophilic-hydrophobic interface.For example, this would help the reader understand why N2 nanobubbles, which might seem to be chemically inert, can play an important role, as this manuscript points out.

One a different matter, I am uncomfortable with how your manuscript ends by making a deep dive into philosophy, and I suggest that this is not the place to do that.I am referring to the following:

“D’Arcy Thompson in his famous book On Growth and Form, Cambridge University Press 1916-1968

tells us that of the chemistry his day and generation, Kant declared that it was a science, but not Science

- “eine wissenschaft, aber nicht Wissenschaft” - for that the criterion of true science lay in its relation

to mathematics.

It as an old story that dates to Roger Bacon and Leonardo da Vinci. D’Arcy tells us too, as do the

founders of the cell theory of biology, that physiology is vastly strengthened and enlarged by making

use of the chemistry, and of the physics of the age.

Both statements are true, but not true.

For if the mathematics we use is based on false gods as happened with too simplistic employment of

computer simulation it is worse than misleading. And the same for chemistry.

We have attempted to explain how the foundations became deeply flawed. They are missing dissolved

gas and its capacity to organise with the aid of lipids into nanobubbles, and their reactivity.

It is a startling jump to realise that gas exchange is via nanobubbles and the consequences of that.

The foundations are missing in their treatment of molecular forces and the extraordinary behaviour

and specificity of salts and bubble fusion; of the role of CO2; and even the existence of the endothelial

surface layer integral to the exchange of gas, and lethal to Covid."

You are certainly entitled to express your own opinions, but I think this approach detracts from the points you are wishing to make.I suggest you can convey the same information in a less argumentative manner.I think that it will be better accepted if you change this approach.

On some minor matters, I prefer a space to be put between a number and its units.The present manuscript is inconsistent in doing that.I also think I should point out that in the first part of the manuscript whoever typed this manuscript confused O with 0.Please change 02 to O2, C02 to CO2, H20 to H2O, etc. throughout.

---

## [Reviewer Report]

*Comments to Author*: This is a highly insightful article that should be published as it deals with a topic that is seldom dealt with in depth.

I have just a couple of comments.

1) The correct notation is: COVID-19. The authors had used several notations inconsistently: Covid, Covid 19 and Covid19.

2) I am wondering if the authors had looked comparatively at both COVID-19 and the 2003 SARS. A comparative study could

allow them to gain further insights into the physiological effects of both SARS-CoV-2 and SARS-CoV-1 infections. SARS-CoV-1 is more virulent than SARS-CoV-2

as the case fatality rates are 0.5-2% and 10% respectively.I suspect that while SARS1 is more virulent, SARS2 causes greater morbidity as clinical studies have

shown that there are many cases of long COVID-19 with damage to the lungs. Goh et al has proposed that SARS1 replicates more rapidly in vital organs such

as the lungs even if SARS2 tend to release more infectious particles because of the greater disorder at the inner shell protein (N). A good comparative clinical

study of SARS1 and SARS2 can be found at:

Wolfel, R.; Corman, V.M.; Guggemos, W.; et al. Virological asscessmant of hospitalizedpatients with COVID-19. Nature. 2020, 581, 465-469.

Further details about the differences between the virion physiology of SARS1 and SARS2 can be found at:

Goh, G.; Foster, J.; Dunker, A.K.; Uversky, V. Computational, Experimental, and Clinical Evidence for a Specific but Peculiar Evolutionary Nature of (COVID-19) SARS-CoV-2. Preprints 2021, 2021050025

https://www.preprints.org/manuscript/202105.0025/v1

It is also possible that comparison between SARS1 and SARS2 is beyond the scope of the present article. If so, this is thensomething to look into in for your next article.

---

## [Reviewer Report]

*Comments to Author*: Review of QRD-D-21-00026

“COVID-19 as a natural experiment…” Parts 1 and 2

Summary:

Paper 1 contain essentially one imaginative and controversial idea regarding the physiology of Covid infection in humans. That idea is surely worthy of dissemination, since it is backed up with clinical findings and other references, although it is certain to be resisted. It should be published, with some revisions.

Paper 2 presents the fundamental background to the idea in Paper 1. Again, I find that an excellent idea. In my view, the paper can be strengthened significantly. It is a useful one, provided it is confined to its intent as set out in the initial paragraphs - to provide background to the idea in Paper 1. I therefore suspect it would be substantially improved with some editing. I can understand why the authors have included so many comments en passant in that Paper 2, but they serve only to weaken to foundations of the argument.

General comments:

The authors write that the 3D box-like structure of the pulmonary surfactant-proteins complex, for which there is good evidence from EM data, acts as a host for compartmentalisation of inhaled gases as nanobubbles (whose existence is also being confirmed more and more in recent times).

*I ask that they clarify one issue which is not currently explained, namely that the complex CLP structure is likely to be maintained in the presence of those nanobubbles. Indeed, my understanding is that the authors suggest that the CLP structure essentially stabilises the nanobubbles, since the ‘collapse’ of CLP leads to freeing of the inhaled gases, which are then used for respiration. This is a really attractive idea, but I don’t know why they should be trapped in the CLP membrane array, unless they are too large to escape (which is the assumption I guess). What then sets the nanobubble dimensions?

* Although it is intuitively appealing, I don’t like the analogy of the collapsed CLP membrane with unravelling a knotted sweater. The analogy seems to be be assumed to be very strong, since the authors write of “unravelling” of bilayer “strings”. Its that reasonable, given the absence (to my knowledge) of string-like assemblies of bilayers, which tend to form 2D not 1D structures?

* A few physiological terms are likely to be unknown to the typical reader (incl me). Can the authors please define lung “compliance” and “functional residual capacity”? The compliance and surface tension relaxation times on injection of surfactant are independent, an interesting observation. I suggest the authors spell out the implications of the CLP structure for lung compliance and surface tension. What is going on to slow the compliance so much? And is the 15 minute time-frame consistent with surfactant absorption and assembly? Can they also define “dyspnea”?

* One final general query: Are the authors saying that the CLP structure collapses and reforms on each respiratory cycle? Surely, that is accommodated by changes in lattice dimensions of the CLP structure, as proposed in the original 1999 J Dispersion Sci paper by Larsson et al?

* There are a number of clear and illuminating images and descriptions of the CLP structure currently “hidden” in Paper 2. I suggest that those materials can be moved to Paper 1, since it would help the reader when first confronted with the model - likely to be new for many.

Specific comments:

Paper 1.

p. 5 (Table 1) row 5: I dot understand why SARS is removed from blood but not other fluids. “Our premise” and “Our conclusion” columns don’t seem to explain that. (Note also the text in “Our confusion” in that row seems garbled.)

p. 7: Please define the “V/Q mismatch”, and “pulmonary shunting”.

p. 8: typo: …. near bottom “…who had been briefly intubateD”

p. 11. Fig 1. A-C are surely sections of surface models rather than actual images? Also, the lattice dimensions of the CLP structure are very free, so I don’t see why “it had precisely the dimensions of the lattice-like structure seen… in TM”. From my reading of a later Larsson and Larsson paper (Advances in Colloid and Interf Sci, vol 205, 68-73 (2014)), the dimensions were tuned (as was the c/a ratio) to get agreement and to give quasi-circular holes.

p. 11 text: The CLP structure “is what pulmonary surfactant looks like when in its most ordered and contracted state”. Are the authors comparing that to the liposomal bodies also seen? Or does the structure collapse each breathing cycle, since the nanobubbles are then released… although on that issue too the authors write “it is possible that the entire sheet of PS is frictionless enough to slide through the capillary wall…” Is that a collapsed sheet, or a fragment ofd the CLP structure? The Box 2 note on the following page suggest that CLP structure swells ands shrinks without topologically changing on breathing, which makes sense. Maybe clarify and tighten this point.

p.12: The discussion of surfactant strings and threads is unnecessarily unconventional in my view. (See my comment above.)

p. 13: I know of no surfactant which raises surface tension. Indeed, if a molecule does that, does it still qualify as a surfactant? I am confused by this claim. It seems to me that it requires some amplification of the concept of measurement of surface tension. Is it a surface monolayer measurement in a trough? If so, does that have any relevance to the situation in vivo, which is very unlikely to be a monolayer? If something else, how is it done and how relevant is it to the physiological situation? I find the subsequent discussion of diving seals and their physiology really intriguing and interesting. The final sentence of that paragraph “In routine dives…” includes some very messy units and measures, which can be edited to read better.

p. 15, bottom para. Capillary dimensions are listed as µM, whichis confusing (metres of moles?)‥ change to µm?

p. 17: The protein SP-A is a dodecamer, or a hexameter of trimers. There is some confusing mentions of it as a hexagonal protein, and later a trimeric protein.

p. 19 (Table 2): Some typos “Very Much‥” and “Just Pulmonary‥” and “Effects” (affects?). More broadly, it is unclear to me what this Table offers the reader apart from off-centre speculations?

Paper 2.

The paper is riddled with one sentence paragraphs, which are in place too terse and abrupt to this reader.

In places, it is simply too stuffed with suggestions and ideas to remain digestible. I applaud the breadth of the authors, but fear that broadening focus so much loses brightness here and the story fades in and out of the readers mind. I wold suggest that the side comments on asbestosis, tobacco smoke, Permian extinctions, Egyptian mummies, pot cleaning, Nafion, electric eels and acupuncture, arginine, trace metals… weaken the intent of the papers. Focus on the matters at hand: nanobubbles, self-assembly and the model of the ESL. No more.

I was also sideswiped by section 7.3, which - even if only peripherally relevant and hence removable - I do not understand.

Paper 1 would profit from incorporation of Section 4.1 - which is very interesting - early on, demonstrating the serious issues with current understanding of respiration physiology.

Section 4.2 is confusing to me. The intro repeats what is already written. But there is an important claim made early on: “The plug is pulled, by removing SP-A and the gas is released to another medium.” It is unclear when this binding to ACE11 occurs? (A reference would be useful here.) Is that happening each respiratory cycle? In any cases, that material can be either incorporated once into Paper 1, or removed if it is already stated.

Later on, it deals with N2, but I simply do not understand the focus on N2. Much of it strikes as peripheral (and disposable?)

Section 4.4 is confusing to me. I don’t get it. Is it essential to the issue?

Section 5 is relevant and interesting (except the distracting introductory terse dot points.) The sterilisation function of CO2 in nanobubbles is intriguing.

5.2 and 5.3 (surely one subsection rather than two) is very brief, but relevant. And the “implications” section is also simultaneously terse, yet sweeping. Children have smaller lung capacity than adults (I suspect), so is their “greater physical capacity” sure to induce increased CO2 nanobubble production and hence sterilisation? After all, children are more prone to other viruses, which should also be sterilised in the same way?

The para breaks between the first mention of Oberleithner and the following 2 paragraphs make it very tough to read the argument.

5.4 makes some very strong claims, which are peripheral to the paper in my view.

Section 6 offers a tenuous link to the main thread. They d support the whole notion of nanobubble activity in biology, they too dilute the main case.

p. 16: The authors write “Note that a curved bilayer is necessarily asymmetric.” That is untrue without qualification - for example the CLP structure is one of zero mean curvature and hence symmetric. A vesicle has nonzero mean curvature, and is necessarily asymmetric.

p. 17: The caption to Fig 6 describes “water channels between the red and blue sides”- this should read “‥channels within the red and blue sub-volumes‥>” (or something similar). In those part of the paper too, there are many white spaces, suggesting new paragraphs, making reading difficult.

Section 7 discusses giant vesicles, which are possibly relevant to the picture, since similar structures are seen adjacent to the CLP membrane regions. However these structures are more reminiscent of dense liposomes, admittedly swollen. I believe the Gershfeld vesicles are single-walled?

p. 20: First para. This is a very interesting statement, despite it standing in total isolation, making identification of the “unsustainable myth” difficult! Here a second discussion of surface-tension-elevating surfactants appears - I assume the “myth” is the belief that surfactants lower surface tension? See my previous discussion of this issue in Paper 1 comments.

Section 7.3 is weak in my view, adding little to the broader picture. I would remove.

Section 8 intro and 8.1 are excellent. They can be moved wholesale into Paper 1 - assisting the reader as to details of the complex CLP lipid-protein assembly.

p. 23: I note the image in Fig 9 is from a metal lung. My understanding is that lung surfactants are only fond in vivo after birther - indeed, they are produced at speed immediately on birth. So I don’t understand how this CLP-like image can be found pre-birth?

I wold remove the sentences at the bottom of p. 23 starting with “Several points” I find them peripheral and distracting.

Finally, I find 8.2 and 8.3 more speculative and on shakier ground than the rest of the papers- I would remove them.

In closing, I note that the suggested edits and deletions given above are not indicative of a poor paper(s). These papers are challenging. The best chance of engaging the reader, who is likely to be coming at this with a sceptical eye, is to keep the arguments tightly focussed. And the ideas here merit serious reflection.

---

## [Reviewer Report]

*Comments to Author*: Reviewer #1: The manuscript by Ninham et al. includes two related essays related to Covid 19, and the role of pulmonary surfactants (PS) and gas exchange. In this context the authors raise the question if new physical chemistry is necessary to explain the Covid anomalies with respect to surface tension and hypoxemia. They conclude that PS and drugs enhancing PS production should immediately be applied to treat advanced Covid-19 (Essay I), and that gas exchange involves a phase change of PS and water channels (Essay II). The two essays are divided into subchapters and written in an unusual style, including many short paragraphs consisting only of 1-2 sentences that remind of philosophy. In several instances, the authors state that previous scientific findings have been “ignored” (rather than, e.g., “escaped attention”).

It is possible, and even likely that the use of PS for treating Covid has its merits. This is in fact intensely discussed in the current literature (see e.g., Tayeb et al., Nanoemulsions: Formulation, characterization, biological fate, and potential role against COVID-19 and other viral outbreaks. Colloid and Interface Science Communications, Volume 45, November 2021, 100533, https://www.sciencedirect.com/science/article/pii/S2215038221001734; Rahaman et al. Surfactant-based therapy against COVID-19: A review. Published by De Gruyter November 30, 2021, From the journal Tenside Surfactants Detergents. https://doi.org/10.1515/tsd-2021-2382; Smith et al., Biosurfactants: A Covid-19 Perspective. Front. Microbiol., 09 June 2020, https://doi.org/10.3389/fmicb.2020.01341). Since surfactants can also dissolve other membranes that then ones from the virus, the risks should not be taken lightly.

Regarding the “new physical chemistry”: extraordinary claims require thorough presentation and analysis of data to convince the audience. The broad span of fields further requires a thorough introduction to set the stage. Both are not convincing to me. Moreover, it is striking that the authors mainly cite their own work, and important statements are not always supported with references.

If the authors wish to improve their manuscript, I recommend that they start with describing the relevant anatomy, i.e. where gas exchange happens and what is known about it (including a description of the relevant glycocalyx). The authors should further give a good picture about what is known about the molecular mechanisms of virus uptake - there are several very good cryo-EM studies about this, both related to enveloped viruses in general and for SARS-CoV 2 in particular (rather than simply stating “A virus may cause molecular damage simply by adsorbing to a molecular surface”, without reference). Also of relevance to this manuscript is the role and evolution of globins (with hemoglobin serving oxygen uptake and transport to the tissues, where myglobin accepts oxygen from hemoglobin with an intricate allosteric mechanism). Words like “hypoxemia”, “dyspnea” and “pulmonary surfactants” deserve an explanation, if the readers shall follow - not everyone has a broad background from medicine to physical chemistry. Finally, I recommend that the authors publish their essays on a preprint server to get a wide range of expert comments that will help to improve the manuscript for publication in a journal.

Also the topic of nanobubbles would benefit from a proper review of the literature, including treatment of Covid (see e.g., Review of Oxygenation with Nanobubbles: Possible Treatment for Hypoxic COVID-19 Patients, by Afshari et al. ACS Appl. Nano Mater. 2021, 4, 11, 11386-11412).

I was surprised to read that elephant seals are the deepest-diving mammals, since I thought these were whales (which in fact they are). - And I do not quite understand why restriction enzymes were brought into the discussion? (a section that is not very convincing in the face of decades of advances in enzymology; quote: “No one has any idea how any enzymes catalyse reactions and where the required substantial energy to cut a phosphate bond comes from; except mumble the obligatory word ATP.”)

For the second assay, the structure of pulmonary surfactants is essential. Given the many flaws in other parts of the manuscript, I would like an EM expert to make sure that these data are reliable, and not artefacts of poorly conducted experiments. Finally, if there is “new physical chemistry”, this should be described scientifically, preferably with relevant theoretical models.

If the authors on the other hand are mainly interested in writing an essay in their unusual style, irrespective of the scientific merits, maybe they should find a publisher, who is interested in this? - I am not entirely sure what the intention of the authors is with this manuscript? Maybe they just want to boost their H-index?

Reviewer #3: Reines, Ninham, and Thomas

“COVID 19 as a natural experiment in physiology”

This 2-part manuscript proposes a new hypothesis, supported by extensive experimental evidence, that provides plausible explanations for numerous “COVID anomalies”.The work also suggests potential pathways for COVID treatments based on the physical model.The work challenges previous orthodoxy in the field of physiology related to the surface tension model for the action of pulmonary surfactant.The work ties in multiple threads from work of the Australian school of colloid science, accumulated over recent decades, including the formation of the surfactant cubic phases.The suggestion that these phases are an essential element of the COVID story is a thread that deserves further investigation.

Overall, the 2 papers provide a provocative and compelling overview of this new paradigm illustrating the importance of the endothelial surface layer for gas absorption.I think the work should be published, but I strongly recommend 2 additions/changes prior to publication.

I would recommend a specific section near the end of the 2nd article that outlines further directions of experimental research that can test the hypotheses put forward in the 2 articles.Where do we go next in testing and fleshing out this story?Some concrete suggestions would be very helpful.

The second article is written in a provocative style.This is fine up to a point but I think that article could benefit from a serious revision that smooths out and organizes the narrative.The thread of ideas is clearly there but it needs an organizing revision.Also removal of some of the more provocative comments would not hurt the scientific content.

Reviewer #4: This is a highly insightful article that should be published as it deals with a topic that is seldom dealt with in depth.

I have just a couple of comments.

1) The correct notation is: COVID-19. The authors had used several notations inconsistently: Covid, Covid 19 and Covid19.

2) I am wondering if the authors had looked comparatively at both COVID-19 and the 2003 SARS. A comparative study could

allow them to gain further insights into the physiological effects of both SARS-CoV-2 and SARS-CoV-1 infections. SARS-CoV-1 is more virulent than SARS-CoV-2

as the case fatality rates are 0.5-2% and 10% respectively.I suspect that while SARS1 is more virulent, SARS2 causes greater morbidity as clinical studies have

shown that there are many cases of long COVID-19 with damage to the lungs. Goh et al has proposed that SARS1 replicates more rapidly in vital organs such

as the lungs even if SARS2 tend to release more infectious particles because of the greater disorder at the inner shell protein (N). A good comparative clinical

study of SARS1 and SARS2 can be found at:

Wolfel, R.; Corman, V.M.; Guggemos, W.; et al. Virological asscessmant of hospitalizedpatients with COVID-19. Nature. 2020, 581, 465-469.

Further details about the differences between the virion physiology of SARS1 and SARS2 can be found at:

Goh, G.; Foster, J.; Dunker, A.K.; Uversky, V. Computational, Experimental, and Clinical Evidence for a Specific but Peculiar Evolutionary Nature of (COVID-19) SARS-CoV-2. Preprints 2021, 2021050025

https://www.preprints.org/manuscript/202105.0025/v1

It is also possible that comparison between SARS1 and SARS2 is beyond the scope of the present article. If so, this is thensomething to look into in for your next article.

Reviewer #5: I enjoyed very much reading this lengthy review of the importance of nanobubbles as the possible cause for covert viral damage to three-dimensional structure of pulminary surfactants in lungs.I do have a suggestion that might improve the presentation.It might help to offer an explanation why nanobubbles cause these reactions to occur.I suggest the answer might lie in the existence of an electric double layer (EDL) at the hydrophilic-hydrophobic interface.For example, this would help the reader understand why N2 nanobubbles, which might seem to be chemically inert, can play an important role, as this manuscript points out.

One a different matter, I am uncomfortable with how your manuscript ends by making a deep dive into philosophy, and I suggest that this is not the place to do that.I am referring to the following:

“D’Arcy Thompson in his famous book On Growth and Form, Cambridge University Press 1916-1968

tells us that of the chemistry his day and generation, Kant declared that it was a science, but not Science

- “eine wissenschaft, aber nicht Wissenschaft” - for that the criterion of true science lay in its relation

to mathematics.

It as an old story that dates to Roger Bacon and Leonardo da Vinci. D’Arcy tells us too, as do the

founders of the cell theory of biology, that physiology is vastly strengthened and enlarged by making

use of the chemistry, and of the physics of the age.

Both statements are true, but not true.

For if the mathematics we use is based on false gods as happened with too simplistic employment of

computer simulation it is worse than misleading. And the same for chemistry.

We have attempted to explain how the foundations became deeply flawed. They are missing dissolved

gas and its capacity to organise with the aid of lipids into nanobubbles, and their reactivity.

It is a startling jump to realise that gas exchange is via nanobubbles and the consequences of that.

The foundations are missing in their treatment of molecular forces and the extraordinary behaviour

and specificity of salts and bubble fusion; of the role of CO2; and even the existence of the endothelial

surface layer integral to the exchange of gas, and lethal to Covid."

You are certainly entitled to express your own opinions, but I think this approach detracts from the points you are wishing to make.I suggest you can convey the same information in a less argumentative manner.I think that it will be better accepted if you change this approach.

On some minor matters, I prefer a space to be put between a number and its units.The present manuscript is inconsistent in doing that.I also think I should point out that in the first part of the manuscript whoever typed this manuscript confused O with 0.Please change 02 to O2, C02 to CO2, H20 to H2O, etc. throughout.

Reviewer #6: Review of QRD-D-21-00026

“COVID-19 as a natural experiment…” Parts 1 and 2

Summary:

Paper 1 contain essentially one imaginative and controversial idea regarding the physiology of Covid infection in humans. That idea is surely worthy of dissemination, since it is backed up with clinical findings and other references, although it is certain to be resisted. It should be published, with some revisions.

Paper 2 presents the fundamental background to the idea in Paper 1. Again, I find that an excellent idea. In my view, the paper can be strengthened significantly. It is a useful one, provided it is confined to its intent as set out in the initial paragraphs - to provide background to the idea in Paper 1. I therefore suspect it would be substantially improved with some editing. I can understand why the authors have included so many comments en passant in that Paper 2, but they serve only to weaken to foundations of the argument.

General comments:

The authors write that the 3D box-like structure of the pulmonary surfactant-proteins complex, for which there is good evidence from EM data, acts as a host for compartmentalisation of inhaled gases as nanobubbles (whose existence is also being confirmed more and more in recent times).

*I ask that they clarify one issue which is not currently explained, namely that the complex CLP structure is likely to be maintained in the presence of those nanobubbles. Indeed, my understanding is that the authors suggest that the CLP structure essentially stabilises the nanobubbles, since the ‘collapse’ of CLP leads to freeing of the inhaled gases, which are then used for respiration. This is a really attractive idea, but I don’t know why they should be trapped in the CLP membrane array, unless they are too large to escape (which is the assumption I guess). What then sets the nanobubble dimensions?

* Although it is intuitively appealing, I don’t like the analogy of the collapsed CLP membrane with unravelling a knotted sweater. The analogy seems to be be assumed to be very strong, since the authors write of “unravelling” of bilayer “strings”. Its that reasonable, given the absence (to my knowledge) of string-like assemblies of bilayers, which tend to form 2D not 1D structures?

* A few physiological terms are likely to be unknown to the typical reader (incl me). Can the authors please define lung “compliance” and “functional residual capacity”? The compliance and surface tension relaxation times on injection of surfactant are independent, an interesting observation. I suggest the authors spell out the implications of the CLP structure for lung compliance and surface tension. What is going on to slow the compliance so much? And is the 15 minute time-frame consistent with surfactant absorption and assembly? Can they also define “dyspnea”?

* One final general query: Are the authors saying that the CLP structure collapses and reforms on each respiratory cycle? Surely, that is accommodated by changes in lattice dimensions of the CLP structure, as proposed in the original 1999 J Dispersion Sci paper by Larsson et al?

* There are a number of clear and illuminating images and descriptions of the CLP structure currently “hidden” in Paper 2. I suggest that those materials can be moved to Paper 1, since it would help the reader when first confronted with the model - likely to be new for many.

Specific comments:

Paper 1.

p. 5 (Table 1) row 5: I dot understand why SARS is removed from blood but not other fluids. “Our premise” and “Our conclusion” columns don’t seem to explain that. (Note also the text in “Our confusion” in that row seems garbled.)

p. 7: Please define the “V/Q mismatch”, and “pulmonary shunting”.

p. 8: typo: …. near bottom “…who had been briefly intubateD”

p. 11. Fig 1. A-C are surely sections of surface models rather than actual images? Also, the lattice dimensions of the CLP structure are very free, so I don’t see why “it had precisely the dimensions of the lattice-like structure seen… in TM”. From my reading of a later Larsson and Larsson paper (Advances in Colloid and Interf Sci, vol 205, 68-73 (2014)), the dimensions were tuned (as was the c/a ratio) to get agreement and to give quasi-circular holes.

p. 11 text: The CLP structure “is what pulmonary surfactant looks like when in its most ordered and contracted state”. Are the authors comparing that to the liposomal bodies also seen? Or does the structure collapse each breathing cycle, since the nanobubbles are then released… although on that issue too the authors write “it is possible that the entire sheet of PS is frictionless enough to slide through the capillary wall…” Is that a collapsed sheet, or a fragment ofd the CLP structure? The Box 2 note on the following page suggest that CLP structure swells ands shrinks without topologically changing on breathing, which makes sense. Maybe clarify and tighten this point.

p.12: The discussion of surfactant strings and threads is unnecessarily unconventional in my view. (See my comment above.)

p. 13: I know of no surfactant which raises surface tension. Indeed, if a molecule does that, does it still qualify as a surfactant? I am confused by this claim. It seems to me that it requires some amplification of the concept of measurement of surface tension. Is it a surface monolayer measurement in a trough? If so, does that have any relevance to the situation in vivo, which is very unlikely to be a monolayer? If something else, how is it done and how relevant is it to the physiological situation? I find the subsequent discussion of diving seals and their physiology really intriguing and interesting. The final sentence of that paragraph “In routine dives…” includes some very messy units and measures, which can be edited to read better.

p. 15, bottom para. Capillary dimensions are listed as µM, whichis confusing (metres of moles?)‥ change to µm?

p. 17: The protein SP-A is a dodecamer, or a hexameter of trimers. There is some confusing mentions of it as a hexagonal protein, and later a trimeric protein.

p. 19 (Table 2): Some typos “Very Much‥” and “Just Pulmonary‥” and “Effects” (affects?). More broadly, it is unclear to me what this Table offers the reader apart from off-centre speculations?

Paper 2.

The paper is riddled with one sentence paragraphs, which are in place too terse and abrupt to this reader.

In places, it is simply too stuffed with suggestions and ideas to remain digestible. I applaud the breadth of the authors, but fear that broadening focus so much loses brightness here and the story fades in and out of the readers mind. I wold suggest that the side comments on asbestosis, tobacco smoke, Permian extinctions, Egyptian mummies, pot cleaning, Nafion, electric eels and acupuncture, arginine, trace metals… weaken the intent of the papers. Focus on the matters at hand: nanobubbles, self-assembly and the model of the ESL. No more.

I was also sideswiped by section 7.3, which - even if only peripherally relevant and hence removable - I do not understand.

Paper 1 would profit from incorporation of Section 4.1 - which is very interesting - early on, demonstrating the serious issues with current understanding of respiration physiology.

Section 4.2 is confusing to me. The intro repeats what is already written. But there is an important claim made early on: “The plug is pulled, by removing SP-A and the gas is released to another medium.” It is unclear when this binding to ACE11 occurs? (A reference would be useful here.) Is that happening each respiratory cycle? In any cases, that material can be either incorporated once into Paper 1, or removed if it is already stated.

Later on, it deals with N2, but I simply do not understand the focus on N2. Much of it strikes as peripheral (and disposable?)

Section 4.4 is confusing to me. I don’t get it. Is it essential to the issue?

Section 5 is relevant and interesting (except the distracting introductory terse dot points.) The sterilisation function of CO2 in nanobubbles is intriguing.

5.2 and 5.3 (surely one subsection rather than two) is very brief, but relevant. And the “implications” section is also simultaneously terse, yet sweeping. Children have smaller lung capacity than adults (I suspect), so is their “greater physical capacity” sure to induce increased CO2 nanobubble production and hence sterilisation? After all, children are more prone to other viruses, which should also be sterilised in the same way?

The para breaks between the first mention of Oberleithner and the following 2 paragraphs make it very tough to read the argument.

5.4 makes some very strong claims, which are peripheral to the paper in my view.

Section 6 offers a tenuous link to the main thread. They d support the whole notion of nanobubble activity in biology, they too dilute the main case.

p. 16: The authors write “Note that a curved bilayer is necessarily asymmetric.” That is untrue without qualification - for example the CLP structure is one of zero mean curvature and hence symmetric. A vesicle has nonzero mean curvature, and is necessarily asymmetric.

p. 17: The caption to Fig 6 describes “water channels between the red and blue sides”- this should read “‥channels within the red and blue sub-volumes ‥>” (or something similar). In those part of the paper too, there are many white spaces, suggesting new paragraphs, making reading difficult.

Section 7 discusses giant vesicles, which are possibly relevant to the picture, since similar structures are seen adjacent to the CLP membrane regions. However these structures are more reminiscent of dense liposomes, admittedly swollen. I believe the Gershfeld vesicles are single-walled?

p. 20: First para. This is a very interesting statement, despite it standing in total isolation, making identification of the “unsustainable myth” difficult! Here a second discussion of surface-tension-elevating surfactants appears - I assume the “myth” is the belief that surfactants lower surface tension? See my previous discussion of this issue in Paper 1 comments.

Section 7.3 is weak in my view, adding little to the broader picture. I would remove.

Section 8 intro and 8.1 are excellent. They can be moved wholesale into Paper 1 - assisting the reader as to details of the complex CLP lipid-protein assembly.

p. 23: I note the image in Fig 9 is from a metal lung. My understanding is that lung surfactants are only fond in vivo after birther - indeed, they are produced at speed immediately on birth. So I don’t understand how this CLP-like image can be found pre-birth?

I wold remove the sentences at the bottom of p. 23 starting with “Several points” I find them peripheral and distracting.

Finally, I find 8.2 and 8.3 more speculative and on shakier ground than the rest of the papers- I would remove them.

In closing, I note that the suggested edits and deletions given above are not indicative of a poor paper(s). These papers are challenging. The best chance of engaging the reader, who is likely to be coming at this with a sceptical eye, is to keep the arguments tightly focussed. And the ideas here merit serious reflection.

---

## [Reviewer Report]

*Comments to Author*: Paper I

Page 5, end of 3rd paragraph.in vivo.should be in italic.Same thing twice on page 13

Paper IIAbstract: physiological 0.17 Molar salt…Read: physiological 0.17 molar salt…

Page 3#4. CO2 nanobubbles are potent in… #5. Nanobubbles are integral to enzymatic and other chemical reactions.I think these statements are too strong: say instead CO2 nanobubbles appear to bepotent in… Nanobubbles are most probably integral to…

Page 4. Some pairs, e.g. (Na Cl)…Read:Some pairs, e.g. (Na+ Cl-)…Same with Others (Na ClO4) do notRead: …(Na+ ClO4 -)…

Page 7: This has not just stabilising salt above 0.17 M,‥Read:This has not just the role of stabilising salt above 0.17 M,‥

Three lines below: With the O2/N2 nanobubbles, oxygen is twice as soluble nitrogen.Read: With the O2/N2 nanobubbles, oxygen is twice as soluble as nitrogen.

Pag 8above the critical 0.17 Mol saltRead: above the critical 0.17 M salt

Page 12. Not shown is further work. It was found that on addition of vitamin C, a known free radical scavenger, the enzyme ceases cutting.– do you have a reference?

Page 18Ca++ concentration…Read: Ca2+ oncentration…